# Hydroclimatic vulnerability of peat carbon in the central Congo Basin

Yannick Garcin[1,2,21✉], Enno Schefuß[3,21✉], Greta C. Dargie[4,21✉], Donna Hawthorne[5], Ian T. Lawson[5], David Sebag[6,7], George E. Biddulph[5], Bart Crezee[4], Yannick E. Bocko[8], Suspense A. Ifo[9], Y. Emmanuel Mampouya Wenina[8], Mackline Mbemba[10], Corneille E. N. Ewango[11,12], Ovide Emba[13], Pierre Bola[13], Joseph Kanyama Tabu[11], Genevieve Tyrrell[14], Dylan M. Young[4], Ghislain Gassier[1], Nicholas T. Girkin[15], Christopher H. Vane[16], Thierry Adatte[17], Andy J. Baird[4], Arnoud Boom[14], Pauline Gulliver[18], Paul J. Morris[4], Susan E. Page[14], Sofie Sjögersten[19] & Simon L. Lewis[4,20,21✉]

The forested swamps of the central Congo Basin store approximately 30 billion metric tonnes of carbon in peat[1,2]. Little is known about the vulnerability of these carbon stocks. Here we investigate this vulnerability using peat cores from a large interfluvial basin in the Republic of the Congo and palaeoenvironmental methods. We find that peat accumulation began at least at 17,500 calibrated years before present (cal. yr BP; taken as AD 1950). Our data show that the peat that accumulated between around 7,500 to around 2,000 cal. yr BP is much more decomposed compared with older and younger peat. Hydrogen isotopes of plant waxes indicate a drying trend, starting at approximately 5,000 cal. yr BP and culminating at approximately 2,000 cal. yr BP, coeval with a decline in dominant swamp forest taxa. The data imply that the drying climate probably resulted in a regional drop in the water table, which triggered peat decomposition, including the loss of peat carbon accumulated prior to the onset of the drier conditions. After approximately 2,000 cal. yr BP, our data show that the drying trend ceased, hydrologic conditions stabilized and peat accumulation resumed. This reversible accumulation–loss–accumulation pattern is consistent with other peat cores across the region, indicating that the carbon stocks of the central Congo peatlands may lie close to a climatically driven drought threshold. Further research should quantify the combination of peatland threshold behaviour and droughts driven by anthropogenic carbon emissions that may trigger this positive carbon cycle feedback in the Earth system.

The Congo Basin is the second-largest river basin on Earth, draining a 3.7 million km² catchment[3]. Although dominated by terra firme tropical forest, extensive swamp forests occupy much of the Cuvette Centrale, or 'central depression' region (Fig. 1). The recent mapping of 167,600 km² of peat shows that the central Congo peatlands are the world's largest tropical peatland complex, storing 28% of Earth's tropical peat carbon stock[2]. Existing limited radiocarbon dating shows that these peatlands began forming approximately 11,000 years ago[1]. The areas of interfluvial peatland appear to be rain-fed[1] and form shallow domes[4], largely found in the western part of the region, with river-influenced peatlands also occurring in its eastern part[2]. Overall,

little is known about the history of the vegetation, water, peat or carbon dynamics over the lifespan of the peatland complex. Understanding this history will help determine how vulnerable the ecosystem is to climate change and inform policies to assess logging, oil exploration and agricultural impacts, which all threaten these peatlands[2,5].

Here we evaluate the response of the central Congo peatlands to past hydrologic changes focusing primarily on a high-resolution record of changes in peat preservation, peat decomposition, vegetation and climate. We provide a detailed analysis of a 6.29-m-long core from the centre of an approximately 40-km-wide interfluvial peat-filled basin[4] which is a domed peatland covered by tropical swamp forest[1,4],

[1]Aix Marseille University, CNRS, IRD, INRAE, CEREGE, Aix-en-Provence, France. [2]Institute of Geosciences, University of Potsdam, Potsdam, Germany. [3]MARUM—Center for Marine Environmental Sciences, University of Bremen, Bremen, Germany. [4]School of Geography, University of Leeds, Leeds, UK. [5]School of Geography and Sustainable Development, University of St Andrews, St Andrews, UK. [6]IFP Energies Nouvelles, Earth Sciences and Environmental Technologies Division, Rueil-Malmaison, France. [7]Institute of Earth Surface Dynamics, Geopolis, University of Lausanne, Lausanne, Switzerland. [8]Faculté des Sciences et Techniques, Université Marien Ngouabi, Brazzaville, Republic of the Congo. [9]École Normale Supérieure, Université Marien Ngouabi, Brazzaville, Republic of the Congo. [10]École Normale Supérieure d'Agronomie et de Foresterie, Université Marien Ngouabi, Brazzaville, Republic of the Congo. [11]Faculté de Gestion des Ressources Naturelles Renouvelables, Université de Kisangani, Kisangani, Democratic Republic of the Congo. [12]Faculté des Sciences, Université de Kisangani, Kisangani, Democratic Republic of the Congo. [13]Institut Supérieur Pédagogique de Mbandaka, Mbandaka, Democratic Republic of the Congo. [14]School of Geography, Geology and the Environment, University of Leicester, Leicester, UK. [15]School of Water, Energy and Environment, Cranfield University, Bedford, UK. [16]British Geological Survey, Centre for Environmental Geochemistry, Keyworth, UK. [17]Institute of Earth Sciences, University of Lausanne, Lausanne, Switzerland. [18]NEIF Radiocarbon Laboratory, Scottish Universities Environmental Research Centre (SUERC), Glasgow, UK. [19]School of Biosciences, University of Nottingham, Nottingham, UK. [20]Department of Geography, University College London, London, UK. [21]These authors contributed equally: Yannick Garcin, Enno Schefuß, Greta C. Dargie, Simon L. Lewis. ✉e-mail: garcin@cerege.fr; eschefuss@marum.de; G.C.Dargie@leeds.ac.uk; S.L.Lewis@leeds.ac.uk

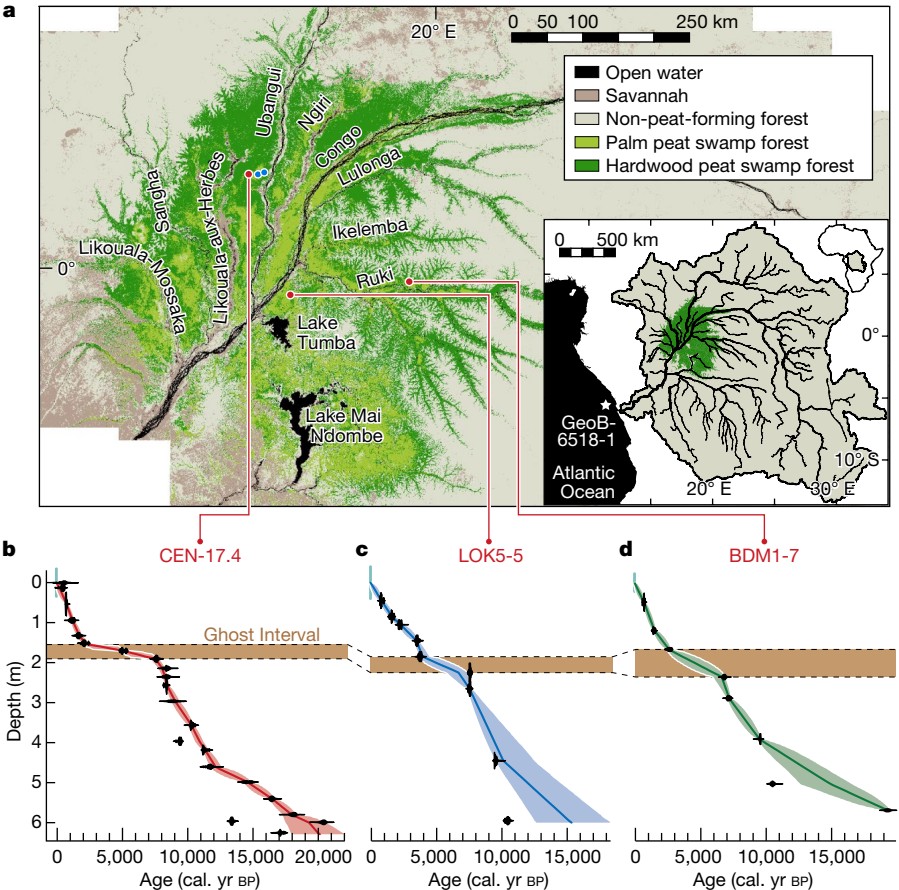

**Fig. 1 | Central Congo Basin peatlands region, peat core locations and radiocarbon chronologies. a**, Map of the Cuvette Centrale showing the spatial distribution of palm-dominated peat swamp forest (light green) and hardwood-dominated peat swamp forest (dark green) derived from ref. [2]. Red dots show the location of cores CEN-17.4, LOK5-5 and BDM1-7, blue dots show the location of cores EKGKM7-2019 (left) and EKG03 (right). Inset map shows the location of the central Congo peatlands, in green, the perimeter of the Congo Basin (black line) and the location of marine core GeoB6518-1[19,38] (white star). **b–d**, Age–depth models of cores CEN-17.4 (**b**), LOK5-5 (**c**) and BDM1-7 (**d**). Median age for each depth (line), 95% confidence intervals (filled envelopes) and calibrated [14]C dates (black markers). A break in the modelled age–depth profiles highlighted by a horizontal brown band indicates the Ghost Interval. Dashed lines show stratigraphic correlations.

informally named Ekolongouma, in the Likouala Department, Republic of the Congo, containing some of the deepest peat yet observed in the region[1,2]. The core, named CEN-17.4 (for the centre of the interfluvial basin), was taken from 1° 11′ 0.49″ N, 17° 38′ 23.7″ E, 327 m above sea level (Fig. 1).

## A Mid- to Late Holocene 'Ghost Interval'

The chronology of CEN-17.4 (see Methods and Fig. 1b) is established by 22 [14]C AMS (accelerator mass spectrometry) dates on the fine fraction less than 150 μm (Extended Data Table 1). Peat accumulation at this location started at 599 cm, that is, 17,500–20,400 cal. yr BP (95% confidence intervals), older than previously dated and less central cores from the same peat-filled basin[1]. In the depth interval 190 to 150 cm, dated from 7,520 to 2,090 cal. yr BP, the gradient of the modelled age–depth profile is five to eight times shallower than in the peat immediately below and above (Fig. 1b). Given the lack of expected peat accumulation, we term this the 'Ghost Interval'.

We assess if this Ghost Interval is a widespread feature of the central Congo peatlands, by [14]C dating two other approximately 6-m-long peat cores from (1) LOK5-5, from a basin close to the Congo River in the Democratic Republic of the Congo, 177 km from CEN-17.4, and (2) BDM1-7, from a river-influenced valley-floor peatland close to the Ruki River, a tributary of the Congo River in Democratic Republic of the Congo, 274 km from CEN-17.4 (Fig. 1, see Methods). The three cores

have a comparable age–depth profile pattern, indicating a common large-scale driver of the Ghost Interval (Fig. 1b–d). However, the precise timing of the onset and termination of the Ghost Interval at each location will be impacted by differing AMS dating resolution and site-specific differences.

The shallower age–depth relationship during the Ghost Interval may represent either slow peat accumulation during the approximate period 7,500 to 2,000 cal. yr BP, caused by changes in peat accumulation or decomposition, or may represent the post-accumulation removal of peat[6,7]. Specifically, four scenarios (see Extended Data Table 2), individually or in combination, could explain the Ghost Interval: (1) Reduced inputs–reduced peat accumulation between approximately 7,500 and 2,000 cal. yr BP, owing to low litter inputs and reduced formation of new peat; (2) Increased contemporaneous decomposition–reduced peat accumulation due to increased contemporaneous decomposition between approximately 7,500 and 2,000 cal. yr BP, owing to increasingly dry conditions; (3) Secondary decomposition–post-accumulation removal of old peat through a subsequent increase in decomposition at or close to approximately 2,000 cal. yr BP, caused by more severe drying that deepened water tables; (4) Secondary removal–physical removal of a section of previously accumulated peat at or close to approximately 2,000 cal. yr BP, by an event such as deep burning, fluvial erosion or anthropogenic disturbance.

We analyse palaeoenvironmental proxies (organic matter properties), palaeovegetation proxies (preserved pollen) and palaeohydrological

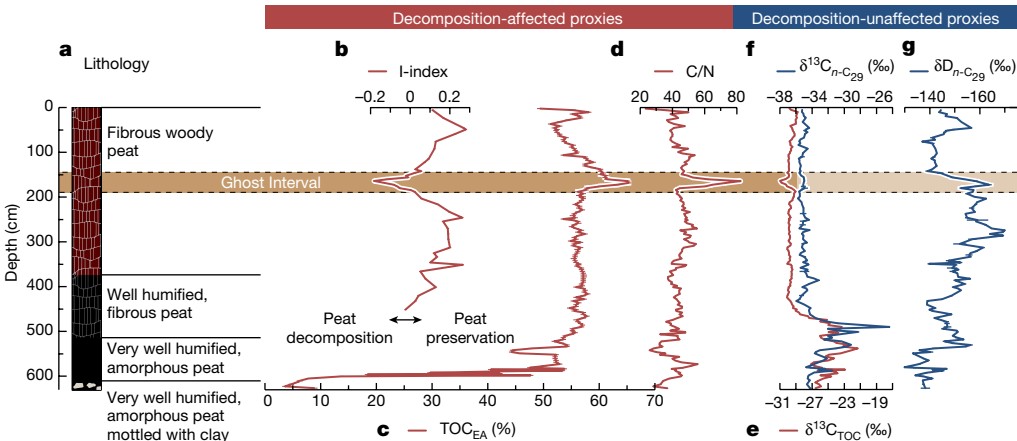

**Fig. 2 | Stratigraphy, decomposition-affected and decomposition-unaffected geochemical properties for peat core CEN-17.4. a**, Lithology. **b**, I-index (I for immature), indicating the preservation of thermally labile immature organic matter[9]. Note that I-index values of fresh organic matter ranges from 0.2 to 0.5 (ref. [9]). **c**, Total organic carbon ($TOC_{EA}$). **d**, Carbon-to-nitrogen ratio (C/N). **e**, Carbon isotopes of total organic carbon (TOC) ($\delta^{13}C_{TOC}$). **f**, Carbon isotopes of $n$-$C_{29}$ alkanes from plant waxes ($\delta^{13}C_{n-C_{29}}$). **g**, Hydrogen isotopes of $n$-$C_{29}$ alkanes from plant waxes ($\delta D_{n-C_{29}}$). Proxy indicators are classified as decomposition-affected and decomposition-unaffected (red and blue lines, respectively). Error bars based on replicate analyses represent the $1\sigma$ uncertainty. The horizontal brown band bounded by dashed lines denotes the Ghost Interval. Note that peat colour and texture in the Ghost Interval is similar to that of the peat immediately below and above. The basal 17 cm of the core, which predates peat formation, consists of very well humified peat mottled with clay representing mineral deposits of unknown age and unknown total depth.

proxies (hydrogen isotopes of plant waxes) from the CEN-17.4 peat core to distinguish between the four scenarios.

## Palaeoenvironmental proxies

Four palaeoenvironmental proxies in core CEN-17.4 indicate that the Ghost Interval consists of highly decomposed peat; that is, more decomposed than during earlier or later periods, consistent with contemporaneous (scenario 2) and/or secondary decomposition (scenario 3).

First, I-index values from Rock-Eval pyrolysis, which describe the balance between thermally labile and resistant pools of organic matter[8–10], indicate intense decomposition of the peat in the Ghost Interval and predominantly better preservation elsewhere in the peat column (Fig. 2b, Methods and Extended Data Fig. 1).

Second, within the Ghost Interval peat, the lowest I-index values are associated with the highest total organic carbon (TOC) values (of up to approximately 65%, Fig. 2c). These values are similar to those measured in lignite[9] (Extended Data Fig. 1a) suggesting that the peat within the Ghost Interval consists of highly condensed, refractory organic matter.

Third, the carbon-to-nitrogen ratio (C/N) indicates increased decomposition (Fig. 2d and Extended Data Fig. 2b), as we observe a prominent increase in C/N to approximately 80 in this part of the record. At our low-nutrient site[11] this may imply a preferential loss of nitrogen through aerobic decomposition, triggered by a lowering of the water table, although other processes might also influence C/N (see Methods).

Fourth, the C-isotope composition of total organic carbon ($\delta^{13}C_{TOC}$) displays an approximate 2‰ shift towards more negative values in the Ghost Interval peat (Fig. 2e). The stable C-isotope composition of recalcitrant plant waxes ($\delta^{13}C_{n-C_{29}}$; where $n$-$C_{29}$ denotes an $n$-alkane containing 29 C atoms) remains relatively invariant across this interval (Fig. 2f), which suggests that the shift in $\delta^{13}C_{TOC}$ was caused by selective loss of a labile, isotopically [13]C-enriched organic fraction[12], again supporting decomposition during the Ghost Interval.

Overall, the Ghost Interval is defined as a section of highly decomposed peat, assessed by negative I-index values and corresponding increases in TOC values and C/N ratios. We extended this analysis of palaeoenvironmental proxies to two cores closer to the edge of the Ekolongouma interfluvial basin peatland, EKGKM7-2019 and EKG03, 18 and 21 km east of CEN-17.4, and 8 and 6 km from the edge of the peatland.

They show a Ghost Interval with similar trends in the I-index, TOC values and C/N ratios, consistent with contemporaneous or secondary decomposition (Extended Data Fig. 3).

## Palaeovegetation proxies

Reconstructions of past vegetation using pollen assemblages (see Methods) can help distinguish among the four scenarios, particularly whether reduced litter inputs resulted in reduced peat accumulation (scenario 1). Arboreal pollen fraction was greater than 85% before, during, and after the Ghost Interval, indicating continued forest cover at CEN-17.4, and therefore suggesting no large change in litter input, counter to scenario 1 (Fig. 3c). Pollen concentration increased during the Ghost Interval, consistent with preservation of relatively recalcitrant pollen grains alongside contemporaneous and secondary decomposition of more labile components of the peat (scenarios 2 and 3; Extended Data Fig. 4f).

The two most abundant swamp forest-associated taxa, in terms of pollen fraction at CEN-17.4, are *Pandanus* and *Pycnanthus*. Both decline in abundance (with fluctuations) throughout the Ghost Interval, from approximately 50% to less than 5% (both taxa combined) of all pollen by 2,000 cal. yr BP (Fig. 3d,e). This decline in swamp forest taxa is accompanied by an increase in light demanding and pioneer tree taxa (Extended Data Fig. 4). Thus, over the Ghost Interval the hydrophilic swamp forest taxa were gradually replaced by tree taxa tolerant of drier conditions, including more light-demanding taxa. Given that flooded and terra firme forests do not differ in overall litter production[13], and annual litter production is not related to total precipitation in tropical forests[13], the pollen data, although indicative of drier local conditions, are not consistent with reduced litter inputs reducing peat accumulation (scenario 1).

Immediately following the Ghost Interval, swamp forest taxa gradually increased, reaching their maximum abundance in the record by approximately 800 cal. yr BP (Fig. 3d,e), indicating that the peat swamp forest ecosystem recovered from the prior disturbance, as net peat accumulation also resumed (Fig. 1b).

Evidence for human impact is absent from the CEN-17.4 record: pollen from cultivated plants is absent and there is no increase in charcoal during the Ghost Interval, suggesting no increase in fire extent or frequency

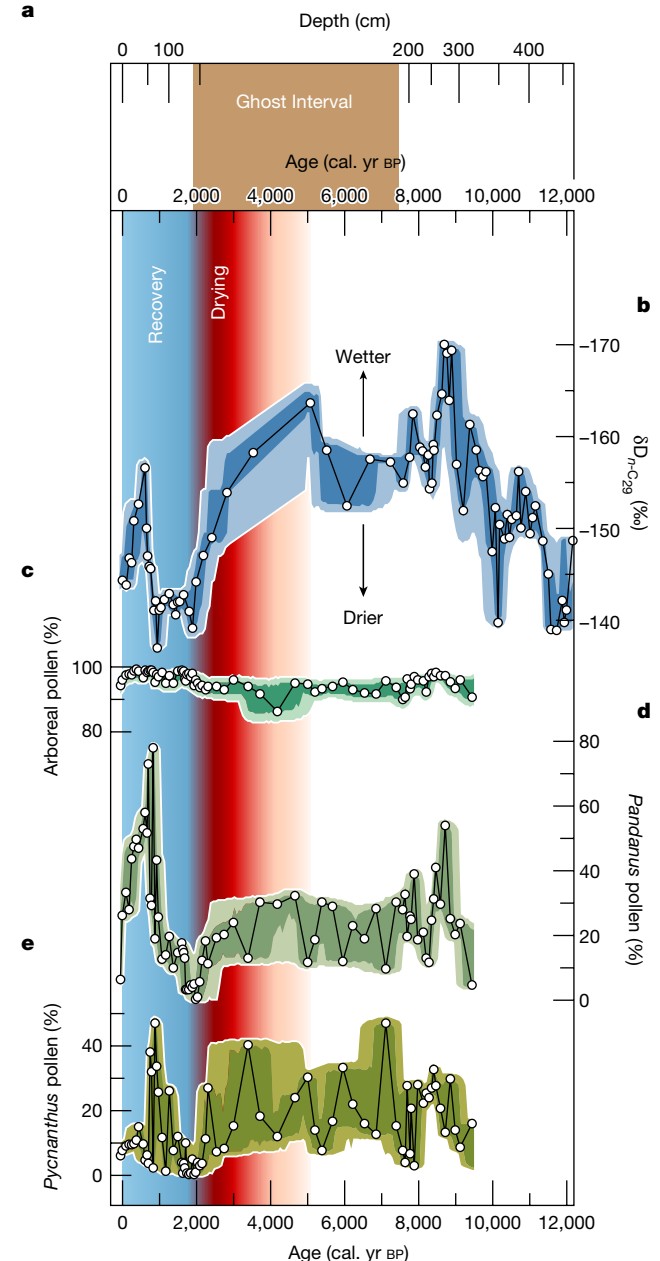

**Fig. 3 | Hydroclimate and vegetation proxy records from peat core CEN-17.4.**
**a**, The Ghost Interval (vertical brown band) shown in depth (top horizontal axis) and age (bottom horizontal axis); the depth axis is plotted to match the age axis. **b**, Hydrogen isotopes of $n$-$C_{29}$ alkanes from plant waxes, $\delta D_{n\text{-}C_{29}}$; note the reversed vertical axis; more negative values are at the top of the graph and indicate wetter conditions. **c**, Arboreal pollen. **d**, *Pandanus* pollen. **e**, *Pycnanthus* pollen. Pollen records are reported as a percentage of total pollen counts. In **b**–**e**, white dots are data points, black lines are interpolated, envelopes reflect 68% (dark) and 95% (light) confidence intervals in the reconstructions, based on analytical and age model errors. The vertical reddish band denotes the drying trend that resulted in the Ghost Interval, with darker colours indicating drier conditions. The vertical blue band denotes the subsequent peat recovery.

(Extended Data Fig. 4g). A [14]C sample near the middle of the Ghost Interval (Fig. 1b) produced an intermediate age between its top and bottom boundaries, suggesting that secondary removal of peat (scenario 4) is unlikely at the CEN-17.4 site (not a hyperlocal event, for example, tree tip-up pool formation[14], or a larger-scale event, for example, natural or anthropogenic deep burning or erosion). Although Iron Age human

settlements are reported in the central Congo from approximately 2,500 cal. yr BP onward[15,16], we see no evidence of humans affecting this very remote swamp forest.

## Climate-mediated peat decomposition

The palaeoenvironmental proxies and vegetation reconstruction suggest that either contemporaneous (scenario 2) and/or secondary (scenario 3) decomposition are the most probable causes of the Ghost Interval. To discriminate between these scenarios, we estimate past changes in precipitation regimes (amount and seasonality of precipitation) using plant wax $\delta D_{n\text{-}C_{29}}$ (Fig. 3b) that reflect the isotopic composition of precipitation at the time the plant was produced[17] ($\delta D_{precip}$), which in turn is negatively correlated with precipitation amount in central Africa[18] (Extended Data Fig. 5).

Over the last 12,000 years, $\delta D_{n\text{-}C_{29}}$ values (Fig. 3b) range from −170 to −137‰. From approximately 12,000 to 9,000 cal. yr BP, decreasing $\delta D_{n\text{-}C_{29}}$ values indicate a wetting trend, followed by generally wet conditions until around 5,000 cal. yr BP. From approximately 5,000 to 2,000 cal. yr BP, increasingly D-enriched $\delta D_{n\text{-}C_{29}}$ values indicate a gradual drying. The overall 29‰ increase in $\delta D_{n\text{-}C_{29}}$ values observed from approximately 5,000 to 2,000 cal. yr BP reflects a drying that strengthened through time. This coincides with the drying trend detected in an offshore marine archive, which was attributed to the increasing meridional South Atlantic sea-surface temperature gradient from the Mid- to Late Holocene causing intensified trade-winds which reduced moisture transport from the Atlantic Ocean onto central Africa[19].

The $\delta D_{n\text{-}C_{29}}$-derived drying is consistent with the very shallow peat age–depth relationship (Fig. 1b), decomposed peat (Fig. 2), and decline in pollen from swamp forest-associated taxa (Fig. 3d,e) at CEN-17.4 across the Ghost Interval. This pattern of drying and shallower age–depth relationship is seen in other tropical peatlands in Amazonia[20,21] and Southeast Asia[22]. Yet, the basal age of the highly decomposed peat in the Ghost Interval at CEN-17.4 is around 7,500 cal. yr BP, which is older than the beginning of the climatic drying at approximately 5,000 cal. yr BP. However, the end of the decomposed Ghost Interval section and the end of the climatic drying trend are coincident at approximately 2,000 cal. yr BP. This pattern is consistent with secondary decomposition (scenario 3), probably resulting from drier conditions lowering the water table and exposing older peat layers to oxidation, including peat deposited prior to the beginning of the drying, that is, decomposing peat older than 5,000 cal. yr BP (compare the 'Ghost Interval' and 'Drying' periods in Fig. 3a,b).

To provide an estimate of the level of drying we compared the relative changes in peat $\delta D_{n\text{-}C_{29}}$ values at CEN-17.4 with $\delta D_{precip}$ values computed using modern climate data (see Methods and Extended Data Fig. 6) to provide a climate space that includes all climate solutions consistent with peat $\delta D_{n\text{-}C_{29}}$ values for both the pre-drying period, that is, at 5,000 cal. yr BP (dashed lines in Fig. 4) and the end of the drying trend, that is, at 2,000 cal. yr BP (solid lines in Fig. 4). First, the climate space data indicate that the present-day peatlands of central Congo exist under considerably drier conditions than other tropical peatlands in America and Asia/Oceania. Second, the reconstructions of past precipitation regimes suggest that the precipitation at approximately 2,000 cal. yr BP was at least 800 mm yr⁻¹ lower than that at approximately 5,000 cal. yr BP, and potentially as much as 1,500 mm yr⁻¹ lower (Fig. 4). This precipitation reduction may have also increased the seasonality, which is often detrimental to peat accumulation[23].

Our hydroclimate record also reveals drier conditions at approximately 12,000 cal. yr BP (Fig. 3b), but this is not associated with as shallow an age–depth profile as we find in the Ghost Interval (Fig. 1b). This probably relates to the differing environment at the time, which was dominated by $C_4$ grasses that are more typical of wetter floodplain and marshy habitats[24], rather than the forest we find in the Ghost Interval

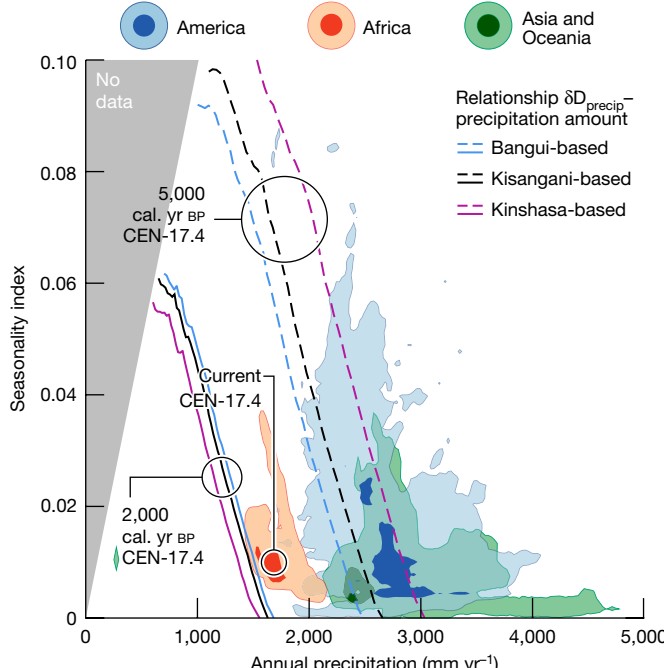

**Fig. 4 | Past changes in precipitation regimes for peat core CEN-17.4 relative to the current climate space of tropical African, American and Asian/Oceanian peatlands.** Climate space (precipitation amount, seasonality index[48]) of tropical peatland areas derived from the CHELSA dataset[49] with spatial extent derived from the PEATMAP dataset[50], plotted as bivariate kernel density estimates (KDEs; see Methods). Dark-coloured and light-coloured surfaces indicate 68% and 95% contours, respectively, from the peak density of the KDEs. Solutions for amount and seasonality of precipitation are derived from peat $\delta D_{n\text{-}C_{29}}$ values at the CEN-17.4 site, for both the pre-drying period (at 5,000 cal. yr BP, dashed lines) and the end of the drying trend (at 2,000 cal. yr BP, solid lines), and are shown as inferred using the modern relationships $\delta D_{precip}$–precipitation amount from Bangui, Kisangani and Kinshasa (Extended Data Figs. 5 and 6). Central Congo hosts about 77% of Africa's current tropical peatland area. The seasonality index is dimensionless. The grey shaded region marked 'No data' is outside the solution space.

(inferred from the shift from higher to lower $\delta^{13}C_{n\text{-}C_{29}}$ values at approximately 460 cm, that is, at around 12,000 cal. yr BP; Fig. 2f). Furthermore, the peat was thinner at approximately 12,000 cal. yr BP, and so probably had a lower elevational gradient between its centre and its margins, likely resulting in slower surface-water flow, and so potentially lessening the impact of the drying[25].

The recovery of hydrophilic swamp forest taxa and renewed high rates of new peat addition following the Ghost Interval ('Recovery' in Fig. 3) contrasts with the high and stable $\delta D_{precip}$, suggesting that the drier conditions were relatively stable and lasted from about 2,000 until approximately 900 cal. yr BP. This may indicate that the peatland ecosystem was more sensitive to the change in the precipitation regime, than to the absolute level of precipitation. Research shows that net peat accumulation can recover, even at permanently lower levels of water inputs, via negative feedback mechanisms such as changes in peat permeability in response to drying, that lead to shallower water tables[26–28]. Alternatively, the recovery may have resulted from an increase in the amount of precipitation and a decrease in seasonality, as this would result in invariant $\delta D_{precip}$ values (Fig. 4) alongside renewed net peat accumulation.

The drying trend recorded at CEN-17.4 is broadly consistent with the drying trend seen in independent hydroclimate records in western central Africa[19,29–31] (Extended Data Fig. 7). Additionally, lake sediment records from or near the Congo coastal region also show drier

conditions between roughly 3,000 and 1,500 cal. yr BP, including changing forest composition and forest being replaced by savannah[32,33]. Our findings of changes in forest taxa are temporally consistent with this so-called 'Late Holocene rainforest crisis', a vegetation disturbance between approximately 3,000 and 2,000 cal. yr BP documented in western central Africa[34,35]. Although the timing, extent, impact and causes of this 'crisis' are debated[31,35–37], our CEN-17.4 site is far from direct human influence, which suggests that the broader vegetation disturbance at the time of the rainforest crisis had an important climatic component.

We found a drying trend at CEN-17.4 that began impacting the peatlands at approximately 5,000 cal. yr BP and culminated at about 2,000 cal. yr BP. This appears to have caused the Ghost Interval, principally via contemporaneous and secondary decomposition (scenarios 2 and 3). This broad Ghost Interval pattern was also seen across our three regionally spaced peat cores, suggesting that the drying impact was large-scale (Fig. 1). Marine sediments in the Congo River deep-sea fan provide an even larger-scale perspective. The presence of pre-aged terrestrial organic matter at site GeoB6518-1 in the Congo fan (Fig. 1a) suggests that the drying conditions may have reduced the area of inundated wetland in the central Congo Basin from the Mid-Holocene onwards, alongside the erosion and release of previously anoxic deposits into the river system[38]. The fan record is consistent with some secondary removal (scenario 4) at the regional scale.

Collectively, the evidence suggests that the hydroclimate change from the Mid- to Late Holocene resulted in a regional drop in the water table and led to both contemporaneous (scenario 2) and secondary (scenario 3) decomposition of peat, coupled with secondary removal via peat erosion at sites close to river courses (scenario 4). The latter result implies that prior to the drying trend beginning at approximately 5,000 cal. yr BP the surface area occupied by the central Congo peatlands may have been larger than today's 167,600 km² (ref. [2]).

## Impacts on carbon cycling

Changes in the slope of the age–depth profile at CEN-17.4 (Fig. 1b) before ($0.50 \pm 0.32$ mm yr$^{-1}$), during ($0.09 \pm 0.05$ mm yr$^{-1}$) and after ($0.74 \pm 0.17$ mm yr$^{-1}$) the Ghost Interval, imply that a substantial amount of peat—and hence carbon—may have been lost from the profile, potentially 2–4 m, compared to the current peat depth of 6 m—that is, 0.5 or 0.74 mm peat accumulation × 5,500 years, the length of the Ghost Interval, minus 390 mm peat remaining in the Ghost Interval, equals 2.36 or 3.68 m peat depth lost. The timing of the decomposition event, driven by drying conditions beginning approximately 5,000 cal. yr BP, suggests that it could have lasted for a maximum period of approximately 3,000 years. However, more rapid responses are possible, because modern-day data from anthropogenic disturbances that lower the water tables of southeast Asian tropical peat swamp forests show rapid carbon losses, of several centimetres of peat per year[39,40]. On the other hand, the lack of a sharp increase in atmospheric $CO_2$ from high-resolution ice core records at approximately 2,000 cal. yr BP implies that carbon release was over a period of at least decades, if not hundreds of years[41].

Our results demonstrate that hydroclimate variability within the Holocene altered the central Congo peatland ecosystems, most probably causing a shift from a carbon sink over millennia[1], to a carbon source for up to 3,000 years. This source reverted back to a carbon sink during a recovery phase over the past 2,000 years. Closer to the present day, our data indicate that a short-term excursion to D-depleted $\delta D_{n\text{-}C_{29}}$ values (wetter conditions) at around 600 cal. yr BP is broadly coeval with an increase in the swamp forest taxa *Pandanus* (Fig. 3b,d). At present, the CEN-17.4 site is dominated by swamp forest taxa and shows no signs of direct human disturbance[11].

Our analyses suggest that the climate space occupied by the central Congo peatlands prior to the period of drying at approximately 5,000 cal. yr BP overlapped the climate space occupied by modern

tropical peatlands in Asia and America, but is much drier today (Fig. 4). Nonetheless, today's central Congo peatlands represent an ecosystem that has recovered from the period of substantial peat loss marked by the Ghost Interval and are less dry than they were at around 2,000 cal. yr BP (Fig. 4). However, given that the boreal summer dry season in the Congo Basin may be lengthening[42–44], our finding of a major peat decomposition event climaxing 2,000 years ago suggests that these carbon dense ecosystems may be more vulnerable to future climate change than most other tropical peatlands.

Our results indicate a positive feedback in the global carbon cycle, if carbon dioxide emissions led to climate-induced drying in the central Congo Basin, which would trigger the release of further carbon from peat to the atmosphere. Rising air temperatures may amplify this feedback by either decreasing forest productivity and therefore litter inputs[45] and/or increasing microbially mediated soil organic matter decomposition rates[46]. However, as our results show, once climatic conditions stabilize, undisturbed peatlands may recover, accumulating peat and sequestering carbon once more. Investments in palaeoenvironmental research at other central Congo peatland sites, monitoring of both regional climate and contemporary peatland condition, and predictive models of peatland carbon storage and release[7,47] are needed to identify future peatland threshold behaviour and fully assess the susceptibility of these carbon dense ecosystems to twenty-first-century climate change.

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

## Methods

### Setting and coring

CEN-17.4 was collected near the centre of a peat-filled interfluvial basin, named Ekolongouma, situated between the Likouala-aux-Herbes and Ubangui Rivers, in the Likouala Department, Republic of the Congo. The overlying swamp forest at CEN-17.4 is formed of a mix of hardwood species including *Uapaca mole* Pax, *Xylopia rubescens* Oliv., *Xylopia aethiopica* (Dunal) A. Rich., *Carapa procera* DC. and the large palm *Raphia laurentii* De Wild., whilst the mid-canopy and understorey is dominated by *Pandanus candelabrum* P. Beauv., with *Afromomum* sp. and *Palisota mannii* C.B. Clarke also common in the understorey. At the time of sampling, the water table was above the peatland's surface by a few centimetres. Peat core CEN-17.4 was collected using a 50-cm-long D-shaped Russian type corer (Eijkelkamp) alternating coring in two boreholes about 30 cm apart in March 2014. The deposits were transferred into PVC half pipe and wrapped in plastic film. As some peat sections compacted during transportation, showing a depth reduction of up to 5 cm per 50 cm, the depth of each sample taken was restored on an undistorted depth scale using AnalySeries 2.0.8 (ref. [51]).

Cores EKGKM7-2019 (1° 10′ 57.29″ N, 17° 48′ 18.4″ E) and EKG03 (1° 11′ 17.7″ N, 17° 49′ 53.29″ E), both also located in the Ekolongouma peat complex (Extended Data Fig. 3), and cores LOK5-5 (at Lokolama; 0° 19′ 36.62″ S, 18° 10′ 24.37″ E) and BDM1-7 (at Bondamba; 0° 10′ 24.38″ S, 19° 41′ 45.56″ E), both located in Équateur province, Democratic Republic of the Congo (Fig. 1), were collected using the same method, with the exception of LOK5-5, which was identical except samples were transported in 10 cm slices wrapped in plastic film.

### Peat chronology

[14]C dates were obtained via accelerator mass spectrometry dating at AWI MICADAS Laboratory (Bremerhaven, Germany) and at NEIF Radiocarbon Laboratory, East Kilbride, Scotland (Extended Data Table 1). As roots from the trees above usually intrude into the peat as it accumulates, we dated two particle-size fractions from the same depth (at 296.5 cm) to evaluate a potential age offset. The coarse fraction >150 μm (including roots) is $670 \pm 150$ [14]C yr younger than the fine fraction <150 μm, suggesting a substantial age offset and a necessity to use the fine fraction for accurate dating. Peat chronology was based on [14]C dates on fine fraction <150 μm for core CEN-17.4 and on fine fraction <180 μm for cores EKGKM7-2019, EKG03, LOK5-5 and BDM1-7. Although different fractions of organic matter may provide different dates, the difference between the sieving meshes used (<150 μm and <180 μm) is probably too small to cause substantial age offsets.

On the basis of 22, nine and eight [14]C dates and assigning the top boundary of the cores to year 2014, 2020 and 2019, we generated age/depth models for cores CEN-17.4, LOK5-5 and BDM1-7, respectively (Fig. 1b–d) using Bayesian approaches, as implemented in the freely available rbacon package 2.5.7 (refs. [52,53]) using R statistical computing language (version 3.6.3, R Core Team). Following the recommendation on the use of [14]C calibration curves according to the local position of the Inter-Tropical Convergence Zone[54], [14]C dates were calibrated using a mixed curve 50%:50% of north–south air-mass mixing derived from both the Northern Hemisphere curve (IntCal20)[55] and the Southern Hemisphere curve (SHCal20)[54]. The age–depth profiles for cores CEN-17.4, LOK5-5 and BDM1-7 cover the last 20,400, 15,200 and 19,300 cal. yr BP and have median 95% age confidences of ~610 yr (range: 6–3,990 yr), 1,470 yr (range: 6–5,670 yr) and 840 yr (range: 6–3,940 yr), respectively.

In addition, [14]C dates were obtained at or near the top and bottom boundaries of the Ghost Interval in cores EKGKM7-2019 and EKG03 (Extended Data Fig. 3).

### Bulk organic analyses

Bulk organic analyses of total organic carbon ($TOC_{EA}$) and total nitrogen of cores CEN-17.4, EKGKM7-2019 and EKG03 were performed at the University of Leicester, UK on 166, 32 and 18 samples, respectively, using an elemental analyser (SerCon ANCA GSL) interfaced to a continuous flow isotope ratio mass spectrometer (Hydra 20–20). $\delta^{13}C_{TOC}$ was further measured on the samples of core CEN-17.4. Precision on $\delta^{13}C$ was <0.1‰. C/N ratios are expressed in terms of mass.

Higher $TOC_{EA}$ values in cores CEN-17.4, EKGKM7-2019 and EKG03 are associated with enhanced decomposition because the peat from this study area is made almost entirely of organic matter (~97%)[11], and peat decomposition over time results in a continuous increase in C content owing to preferential mineralization of the more labile organic fractions, notably the H-bond-rich compounds (hydrocarbon compounds), leading to a relative enrichment in more refractory forms of carbon, such as C-bond rich compounds (Fig. 2 and Extended Data Figs. 2 and 3).

C/N ratios in cores CEN-17.4, EKGKM7-2019 and EKG03 are also related to decomposition processes. The response of C/N ratios to decomposition depends, in part, on peat nutrient status with low-nutrient peats being characterized by preferential protein loss[56]. The observed topmost increase in the C/N ratios with depth (Fig. 2 and Extended Data Figs. 2 and 3) is consistent with a preferential loss of nitrogen in the surface peat owing to aerobic respiration of organic matter. The C/N ratios stabilize at ~40–50 and do not show any further downward decrease as typically observed in boreal *Sphagnum* peat where the loss of more labile organic C is related to anaerobic decomposition of organic matter in the deep waterlogged peat[57]. Fluctuating wet and dry conditions, varying N deposition and changing vegetation during peat formation can overprint the expected trends of C/N with depth[58–61]. Peat and plant chemistry are different in boreal and tropical regions, resulting in higher recalcitrance for tropical peat[62]. The organic matter of the Ekolongouma peat is highly recalcitrant, dominated by roots and woody material, and anaerobic decomposition of this type of peat is typically very slow under constant high water tables[63]. Because there are no obvious changes in the source of organic matter and/or vegetation cover during the Holocene in the peat cores from the Ekolongouma basin, higher C/N ratios during the Ghost Interval are most probably associated with enhanced aerobic decomposition.

### Rock-Eval analyses

Rock-Eval analyses of core CEN-17.4 (Fig. 2b and Extended Data Fig. 1) were performed at the Institute of Earth Sciences of the University of Lausanne, Switzerland using a Rock-Eval 6 (Vinci Technologies). Rock-Eval is a trademark registered by IFP Energies Nouvelles. 40 peat samples encompassing the Ghost Interval were freeze-dried (50 mg) and finely crushed. To assess the spatial representativeness of the Rock-Eval parameters of core CEN-17.4, cores EKGKM7-2019 and EKG03, which are located to the east of core CEN-17.4 (Extended Data Fig. 3), were further analysed using a Rock-Eval 6 both at the University of Lausanne and at the British Geological Survey, Centre for Environmental Geochemistry, Keyworth, UK.

Rock-Eval analysis is based on continuous measurement of effluents (hydrocarbon (HC), CO and $CO_2$) released during the thermal cracking of organic compounds (and thermal decomposition of carbonates) in pyrolytic conditions (up to 650 °C in an inert atmosphere), then during the combustion of residual organic and inorganic carbon (up to 850 °C in an oxidative atmosphere). A flame ionization detector identifies the release of HC during the pyrolytic stage, and an infrared cell detects the release of CO and $CO_2$ during both stages. The resulting thermograms are used to calculate standard parameters by integrating the amounts of HC, CO and $CO_2$ between defined temperature limits[64,65]. These standard parameters are either quantitative, such as TOC and MINC, which measure the organic and mineral C content, respectively, or qualitative, such as hydrogen index (HI) and oxygen index (OI), which measure the HC or $CO_2$ content in relation to TOC, respectively.

In the absence of carbonates, the organic C content corresponds to the sum of all the carbon moieties (TOC and MINC) released during pyrolysis and oxidation. As Rock-Eval organic C content is standardized for mature sedimentary organic matter (that is, kerogens), we further

applied a correction factor of 1.166256 (determined for immature organic samples)[66] to calculate Rock-Eval TOC ($TOC_{RE}$) values, which are comparable to $TOC_{EA}$ values (see above). As highly organic samples such as peat may saturate the signal during the oxidation phase, which may lead to underestimating $TOC_{RE}$ values, the shape of S4 thermograms was carefully monitored.

The shape of S2 thermograms (HC released during pyrolysis) provides additional information about organic matter quality[66]. Two indices (R-index and I-index) represent the thermal status of organic matter. They are calculated from five subdivided areas of the S2 thermograms (Extended Data Fig. 1c) between the following bounds[8,9]: 200–340 °C for highly labile (A1), 340–400 °C for labile (A2), 400–460 °C for resistant (A3), 460–520 °C for refractory (A4) and 520–650 °C for highly refractory pool (A5). The R-index ((A3 + A4 + A5)/100) relates to the thermally resistant and refractory pools of organic matter, and the I-index ($\log_{10}[(A1 + A2)/A3]$) relates to the ratio between the thermally labile and resistant pools[9] (Extended Data Fig. 1b). R-index, I-index and $TOC_{RE}$ in peat core CEN-17.4 were further compared to the same variables in various organic layers and ferrasols from Gabon and lignites from India[9] (Extended Data Fig. 1).

## Plant wax analyses

Plant wax analyses were performed at MARUM, University of Bremen. 116 samples from core CEN-17.4 were used for plant wax isotope analyses (Fig. 2f,g and Extended Data Fig. 2c–e). Between 240 and 1,470 mg of freeze-dried and finely ground samples were extracted in an ASE200 accelerated solvent extractor using a dichloromethane (DCM):methanol (MeOH) 9:1 solution at 1,000 psi and 100 °C for three cycles lasting 5 min each. Known amounts of squalane were added prior to extraction as internal standard. Total lipid extracts (TLEs) were dried in a Heidolph ROTOVAP system. Residual water was removed over columns of $Na_2SO_4$ using hexane as eluent. Total lipid extracts were saponified in 0.5 ml of 0.1 M KOH in MeOH solution. After adding bi-distilled water, the neutral fractions were obtained by liquid–liquid extraction using hexane. The neutral fractions were separated over pipette columns of deactivated silica (1% $H_2O$) using hexane, DCM and DCM:MeOH 1:1 as eluents to obtain the hydrocarbon, ketone and polar fractions, respectively. The hydrocarbon fractions were further purified by cleaning over columns of $AgNO_3$-coated $SiO_2$ using hexane as solvent to remove unsaturated compounds.

n-Alkanes were quantified using a gas chromatograph (GC; Scientific Focus, Thermo Fisher) equipped with a 30-m Rxi-5ms column (30 m, 0.25 mm, 0.25 μm) and a flame ionization detector. Quantification was achieved by comparing the integrated peak areas to those from external standard solutions consisting of n-alkanes of varying chain length. Repeated analyses of standard solutions indicate a quantification precision of 10%. All samples are dominated by odd-numbered long-chain n-alkanes with $n$-$C_{29}$ and $n$-$C_{31}$ alkanes being the most abundant homologues in all samples. The Carbon Preference Index[67] values $\left(\text{CPI} = \frac{(C_{25}+C_{27}+C_{29}+C_{31}+C_{33}) + (C_{27}+C_{29}+C_{31}+C_{33}+C_{35})}{2(C_{26}+C_{28}+C_{30}+C_{32}+C_{34})}\right)$ of long-chain n-alkanes were 7.3 on average (5–10) indicating their origin from epicuticular waxes of terrestrial higher plants[68].

Hydrogen atoms in plant wax n-alkanes are not affected by isotopic changes caused by selective decomposition of labile compounds, as they are non-exchangeable[69]. δD analyses of n-alkanes were conducted on a MAT 253 isotope ratio mass spectrometer (Thermo Fisher Scientific) coupled via a GC IsoLink operated at 1,420 °C to a GC (TRACE, Thermo Fisher Scientific) equipped with a HP-5ms column (30 m, 0.25 mm, 1 μm). Each sample was measured at least in duplicate. δD values were calibrated against $H_2$ reference gas of known isotopic composition and are given in ‰ VSMOW (Vienna Standard Mean Ocean Water). Accuracy and precision were controlled by a lab internal n-alkane standard calibrated against the A4-Mix isotope standard (provided by A. Schimmelmann, University of Indiana) every six measurements and by the daily determination of the $H_3^+$ factor. Measurement precision was determined by calculating the difference between the analysed values of each standard measurement and the long-term mean of standard measurements, which yielded a 1σ error of <3‰. $H_3^+$ factors varied between 4.9 and 5.2 (mean ± s.d., 5.1 ± 0.1). Accuracy and precision of the squalane internal standard were both 2‰ ($n = 238$). Precision of the replicate analyses of the $n$-$C_{27}$, $n$-$C_{29}$ and $n$-$C_{31}$ alkanes was 1‰ on average.

$\delta^{13}C$ analyses of n-alkanes were conducted on a MAT 252 isotope ratio mass spectrometer (Thermo Fisher Scientific) coupled via a gas chromatograph-combustion (GC-C) interface with a nickel catalyser operated at 1,000 °C to a GC (Trace, Thermo Fisher Scientific) equipped with a HP-5ms column (30 m, 0.25 mm, 0.25 μm). Each sample was measured at least in duplicate. $\delta^{13}C$ values were calibrated against $CO_2$ reference gas of known isotopic composition and are given in ‰ VPDB (Vienna Pee Dee Belemnite). Accuracy and precision were determined by measuring n-alkane standards calibrated against the A4-Mix isotope standard every six measurements. The difference between the long-term means and the measured standard values yielded a 1σ error of <0.3‰. Accuracy and precision of the squalane internal standard were both 0.2‰ ($n = 264$). Precision of the replicate analyses of the $n$-$C_{27}$, $n$-$C_{29}$ and $n$-$C_{31}$ alkanes was 0.1‰ on average.

We note that peat n-alkanes in the coarse fraction >150 μm (plant debris including roots) and in the fine fraction <150 μm have similar chain-length distributions and identical (within errors) δD and $\delta^{13}C$ values, suggesting that n-alkanes in both grain-size fractions derive from the same source organisms (Extended Data Fig. 8). Different vegetation types (for example, $C_3$ versus $C_4$) can cause offsets in δD values of plant wax, owing to different hydrogen isotope fractionation factors[17]. However, as the measured n-alkane $\delta^{13}C$ values are stable during the Holocene (Extended Data Fig. 2d), no correction for vegetation type changes was applied. Global ice volume changes can further affect isotopes in the hydrological cycle[70], but, as this effect is insignificant for the Holocene, no ice volume correction was applied. The $\delta D_{n\text{-}C_{27}}$, $\delta D_{n\text{-}C_{29}}$ and $\delta D_{n\text{-}C_{31}}$ roughly co-vary (Extended Data Fig. 2e). However, the $\delta D_{n\text{-}C_{27}}$ values (range of −199 to −145‰) are much lighter than the $\delta D_{n\text{-}C_{29}}$ and $\delta D_{n\text{-}C_{31}}$ values (range of −170 to −130‰ and −161 to −131‰, respectively). Conversely, the $\delta^{13}C_{n\text{-}C_{27}}$ values are systematically higher (approximately +1.5‰) than the $\delta^{13}C_{n\text{-}C_{29}}$ and $\delta^{13}C_{n\text{-}C_{31}}$ values during the Holocene, suggesting that the $n$-$C_{27}$ alkanes were overprinted by contributions of other source organisms such as bacteria[71]. As the $\delta^{13}C_{n\text{-}C_{29}}$ and $\delta^{13}C_{n\text{-}C_{31}}$ values are similar in magnitude and strongly co-vary, $n$-$C_{29}$ and $n$-$C_{31}$ alkanes probably derive from the same source organisms. Isotope interpretations were based on the $n$-$C_{29}$ alkane, which was the most abundant long-chain homologue and which we infer to originate from terrestrial higher plants on the basis of the above observations.

## Pollen and charcoal analyses

Seventy five peat sub-samples were extracted from the first 315 cm of core CEN-17.4 for palynological analysis. Laboratory preparations followed standard palynological methods adapted from ref. [72], substituting density separation for the HF treatment. Two *Lycopodium* tablets (batch no. 201,890 with 11,267 ± 370 spores per tablet) were used as an exotic marker against 1 cm$^3$ of peat. Analytical resolution varied from a minimum 8 cm to higher resolution (concentrated across the Ghost Interval). Pollen grains were counted and identified at ×1,000 magnification using a Zeiss Axioskop microscope. Identifications were made using a personal pollen key, including reference material compiled from the literature and personal observations and/or photographs from reference collections at the University of Oxford, Goethe University, University of Montpellier II, Pierre and Marie Curie University Paris and CEREGE. Pollen grains were counted to a minimum 300 pollen grain sum; percentage calculations were made in Tilia (v.2.0.41). Pollen preservation was largely observed to be excellent throughout the core, with a low proportion of indeterminate grains and no notable decomposition of grains. *Pandanus* and *Pycnanthus* pollen were selected to describe indicative changes in swamp forest vegetation. Both taxa are well-known swamp forest taxa, occurring across central Africa. More specifically, *Pandanus candelabrum*[73],

which we find at the CEN-17.4 site today, is a relatively heliophilic species often found on hydromorphic soils and can tolerate standing water up to about 1.2 m deep[74]. It is seen in the wider peat swamp forests of the Ekolongouma region (G.C.D., Y.E.B. and S.L.L., personal observations) and has been documented in swamp forests around Lake Télé, Lake Djéké, Lake Déké, Lake Manmagoye, in the flooded forests and on the banks of the watercourses in the Cuvette Centrale[24]. Two species of *Pycnanthus* are documented in the Cuvette Centrale, *Pycnanthus angolensis* and *Pycnanthus marchalianus* Ghesq. They are both relatively heliophilic species and are often found on hydromorphic or clay–sandy soils. *Pycnanthus angolensis* is common in swamp forests as well as riverine and valley forests, open woodland and secondary bushland[75]. *Pycnanthus marchalianus* Ghesq. is a more obligate swamp specialist, documented in the inundated forests of the Cuvette Centrale on marshy soils[24].

Macroscopic charcoal was analysed on the same samples of core CEN-17.4 analysed for pollen, following standard methods[76]. Charcoal samples (1 cm$^3$) were sieved using a mesh size of 150 μm. The total area, individual particle size and total number of particles were quantified for each subsample using digital image analysis[77]. The records of charcoal area versus number were comparable, confirming that particle breakage during processing was not an issue. Here we present the macro-charcoal record as total particles cm$^{-3}$ (Extended Data Fig. 4g).

### Estimation of precipitation regimes using peat $\delta D_{n\text{-}C_{29}}$

Plant wax $\delta D$ is a robust tracer for $\delta D_{precip}$ and is extensively used in tropical Africa to reconstruct palaeohydrology[19,30,31,78,79]. Because there are no notable changes in both plant life forms and photosynthetic pathways in the CEN-17.4 record during the Holocene (see above) we assume an invariant hydrogen isotope fractionation between $\delta D_{n\text{-}C_{29}}$ and $\delta D_{precip}$. Furthermore, a marine record off the Congo River (core GeoB 6518-1) going back to 20,000 cal. yr BP shows a notable correlation between $\delta D_{n\text{-}C_{29}}$ and the $\delta^{18}O$ of planktonic foraminifera[19]. As the latter is determined by discharge amount of the Congo River, this suggests that both, notably independent, climatic proxies record large-scale central African precipitation changes[19].

We estimated past changes in precipitation amount and/or seasonality using peat $\delta D_{n\text{-}C_{29}}$ values compared with a wide range of $\delta D_{precip}$ values at the CEN-17.4 site computed using a three-step empirical approach based on modern climate data (Extended Data Fig. 6). Peat $\delta D_{n\text{-}C_{29}}$ values were translated into mean annual $\delta D_{precip}$ values using a global surface soil calibration relationship[80], which includes 11 peat samples, and which has a slope of approximately 1.

In the first step, we evaluated the relationship between $\delta D_{precip}$ and precipitation amount at the CEN-17.4 site (Extended Data Fig. 5). In the tropics, stable isotope ratios in precipitation are often negatively correlated with precipitation amounts on a monthly scale[81]. However, the relationship between $\delta D$ (or $\delta^{18}O$) and precipitation amount can be very variable and other convection-related processes such as cloud type (convective versus stratiform), moisture source and transport and cloud microphysics may overprint it[82–84]. The Congo Basin, which has a concentric structure with a large central depression (the 'Cuvette Centrale') surrounded by topographic highs, is one of the most convective regions on Earth[85]. Local recycling processes are very important and more than 80% of the total moisture contribution to precipitation over the basin originates from land sources, with approximately 60% of the total originating from the Congo Basin itself, and with relatively stable contributions throughout the climatological year[86]. Three stations from the Global Network of Isotopes in Precipitation (GNIP)[87] (Bangui, Kinshasa and Kisangani) bordering the central Congo Basin show, on a monthly scale, significant negative linear relationship ($R^2$ range of 0.65–0.81 and $P$ values ≤ 0.001) between $\delta D_{precip}$ and precipitation amount. Different slopes (−2.5, −4.6 and −3.2 for Bangui, Kinshasa and Kisangani, respectively) probably reflect different moisture trajectories and rates of moisture recycling. We selected the empirical inverse relationships from these stations to estimate past precipitation regimes at the CEN-17.4 site.

In the second step, to account for the temporal integration time of a given plant wax compound in the swamp forest, which may integrate environmental information over a large portion of the annual growth[88,89], we computed weighted-mean annual $\delta D_{precip}$ values combining monthly precipitation data encompassing the continental tropics (see below) with the Kinshasa, Bangui and Kisangani relationships between $\delta D_{precip}$ and precipitation amount. We note that because moisture trajectories and recycling are spatially variable, the weighted-mean annual $\delta D_{precip}$ values that we obtained are only indicative for the central Congo peatlands. Weighted-mean annual $\delta D_{precip}$ values were compared to mean annual precipitation and seasonality index (product of relative entropy D of monthly precipitation, with respect to the uniform distribution, and mean annual precipitation normalized with respect to observed maximum within the gridded data)[48] in a climate space plot (Fig. 4).

In the third step, the differences in $\delta D_{precip}$ (derived from $\delta D_{n\text{-}C_{29}}$ values) at 5,000 and 2,000 cal. yr BP relative to current $\delta D_{precip}$ (derived from $\delta D_{n\text{-}C_{29}}$ values) (that is, −20.3‰ and +5.6‰, respectively) were added to the computed weighted-mean annual $\delta D_{precip}$ value at the CEN-17.4 site and resulted in a climate space including all climate solutions for both the amount and the seasonality of precipitation for each time frame. We note that similar climate solutions were obtained using different regional relationships between $\delta D_{precip}$ and annual precipitation (Fig. 4 and Extended Data Fig. 6).

### Constraining climate spaces

We derived climatic values, that is, amount (mm yr$^{-1}$) and seasonality index[48] (dimensionless) of precipitation for the continental tropics based on monthly precipitation amount (mm month$^{-1}$) of the high-spatial-resolution (30 arcsec) CHELSA dataset[49]. The current geographical extents of the main peat-bearing tropical regions, including Africa, Asia/Oceania and America, which are compiled in the PEATMAP dataset[50], were used to extract the climate spaces (precipitation amount, seasonality index) for each of these regions (Extended Data Fig. 6).

The obtained climate spaces were plotted as bivariate kernel density estimates (KDEs) where each data point is converted into a Gaussian curve along the $x$ axis (precipitation amount) and $y$ axis (seasonality index) and with a kernel bandwidth (the width of each individual Gaussian) as determined by optimization[90]. The bivariate KDEs are three-dimensional (3D) plots, which can be contoured on the basis of a specified density contour interval; all contours presented here are 68% and 95% from the peak density.

The geospatial analyses and mapping were performed using open source Jupyterhub notebooks (5.7.8; https://jupyter.org/) running Python 3.7.3 on server (16 Intel Xeon Gold 52Go RAM calculation core; 18R CPU (2.10 GHz)) with fiona (1.8.20), geocube (0.1.0), geopandas (0.10.1), ipykernel (6.4.1), ipython (7.28.0), jupyter (1.0.0), KDE-diffusion (1.0.3), matplotlib (3.4.3), notebook (6.4.4), numpy (1.20.3), pandas (1.3.3), rioxarray (0.7.1), scipy (1.7.1) and shapely (1.7.1) packages.

### Reporting summary

Further information on research design is available in the Nature Research Reporting Summary linked to this article.

### Data availability

Data that support the findings of this study are available in the PANGAEA repository: https://doi.org/10.1594/PANGAEA.938019.

### Code availability

Codes for age–depth models and for processing and analysis of geospatial data (climate spaces, tropical peatland distribution and precipitation reconstruction) are available in the IRD Dataverse repository: https://doi.org/10.23708/FO2HGM.

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

**Acknowledgements** We thank the government of the Republic of the Congo, particularly the Minister of Environment, Sustainable Development and the Congo Basin, A. Soudan-Nonault, the Ministry for Research, and the governor of Likouala Department for permission to sample in the peatlands and for support. We thank the government of the Democratic Republic of the Congo, the governor of Équateur Province, and the Ministry of Environment and Sustainable Development for permission to sample in the peatlands and for support. We thank the villages of Itanga and Ekolongouma who hosted our field campaigns in Republic of the Congo; R. Mobongo, F. Twagirashyaka, P. Telfer and B. Evans, and the Wildlife Conservation Society Congo Programme for logistical support; and Marien Ngouabi University for ongoing advice and support, particularly J. Loumeto and F. Kouabouana in Republic of the Congo. We thank P. Abia, T. Angohouni, I. Mokondo, M. Iwango, A. Mobembe, G. N'Gongo, R. Molayei and B. Elongo (Bolembe and Ekolongouma villages), J.-B. Bobetolo, C. Fatty, Chancel, E. B. Moniobo Belen Ekous, L. Mandomba Landry, and F. Mouapeta Fulgence (Itanga village) for field assistance in collecting the peat cores in Republic of the Congo. We thank the villages of Lokolama and Bondamba for hosting our field campaigns in Democratic Republic of the Congo; R. Monsembula and Greenpeace Africa and J. Mathe, J.P. Lokila, I. Sando, P. Bosange, F. Mongonga, R. Kendewa, B. Bongwemisa and Groupe d'Action pour Sauver l'Homme et son Environnement (GASHE) for logistical support; M. Bosulu, E. Bolalangu, J. Eyombe, N. Ekoko, L. Lombojo, I. Ikalinga, B. Bakki, I. Bonjomba, E. Bokongo, B. Nkombo, B. Boka, B. Mbokalondjo, B. Bokana, N. Bondjoloko, M. Enge, I. Bonkana, L. Lokoso, B. Bonicile, B. Mboyo, E. Bonicile, B. Mongu, Y. Ntange, I. Ikiyo and Y. Madole (Lokolama village); D. Isimba Bombolo, T. Nkuma Lyandja, L. Mbayo, B. Lifoko Mboyo, N. Efoloko, B. Bofifa, Mamale Isimba, E. Ngwayo Wakada, E. Bongwala, J. Efoloko, B. Bofifa, M. Empange, B. Bosela, Mboyo Isimba, Balimembo, P. Lokumba, Z. Bongoli Bofifa, F. Isanyongo Boka, P. Efoloko, E. Isola, B. Isimba, E. Nyangi, I. Bonjanda and B. Iloko (Bondamba village) for field assistance in collecting the peat cores. We thank R. Kreutz at MARUM and W. Hiles in St. Andrews for laboratory support. We are grateful to R. Dommain, M.R. Strecker, A. Licht, B. Hamelin, R. Betts, E. Mitchard, L. Miles, J. Sancho, S. Georgiou, I. Davenport, E. Burke and S. Chadburn, for discussions. H. Plante provided UK logistical support. This work was funded by CongoPeat—a NERC large grant (NE/R016860/1) to S.L.L., I.T.L., S.E.P., A.B., A.J.B., P.J.M., P.G. and S.S., Agence Nationale de la Recherche (ANR) grant ANR-19-CE01-0022 to Y.G. and Deutsche Forschungsgemeinschaft (DFG) grant SCHE903/19-1 to E.S. (joint project 'ORACLE'), Natural Environment Research Council (CASE award to S.L.L. and G.C.D.), Leeds–York NERC Doctoral Training Partnership ('SPHERES') award to B.C. (NE/L002574/1), NERC Radiocarbon Facility NRCF010001 (alloc. no. 1688.0313, 1797.0414, 2222.1119, 14.108 and 2329.0920 to I.T.L., S.L.L., G.E.B., B.C., P.G. and G.C.D.), Wildlife Conservation Society – Congo (to G.C.D.), the Royal Society (to S.L.L.), Philip Leverhulme Prize (to S.L.L.), and a Greenpeace Fund award to (S.L.L.). E.S. was supported by the DFG–Cluster of Excellence 'The Ocean in the Earth System' at MARUM. C.H.V. publishes with permission of the Executive Director of the British Geological Survey, UKRI. This study is a contribution to the International Joint Laboratory 'Dynamics of land ecosystems in central Africa in a context of global changes' (LMI DYCOFAC).

**Author contributions** S.L.L., I.T.L., S.E.P. and G.C.D. initially conceived the study. E.S., Y.G., I.T.L., G.C.D. and S.L.L. dzw.E.N.E. conducted fieldwork and peat core recovery. D.H., D.S., G.E.B., G.T., A.B., N.T.G., C.H.V., S.S., G.C.D., P.G., T.A., Y.G. and E.S. performed sample preparation and analyses. Y.G. and G.G. conducted geospatial analyses and coding. Y.G., E.S. and D.H. compiled and analysed the results. Y.G., E.S. and S.L.L. wrote the paper, with important contributions from I.T.L., G.C.D., D.H., D.S., A.J.B., P.J.M., D.M.Y., A.B. and S.E.P. The figures were designed by Y.G. All co-authors contributed to the final manuscript.

**Funding** Open access funding provided by Staats- und Universitätsbibliothek Bremen.

**Competing interests** The authors declare no competing interests.

**Additional information**
**Correspondence and requests for materials** should be addressed to Yannick Garcin, Enno Schefuß, Greta C. Dargie or Simon L. Lewis.

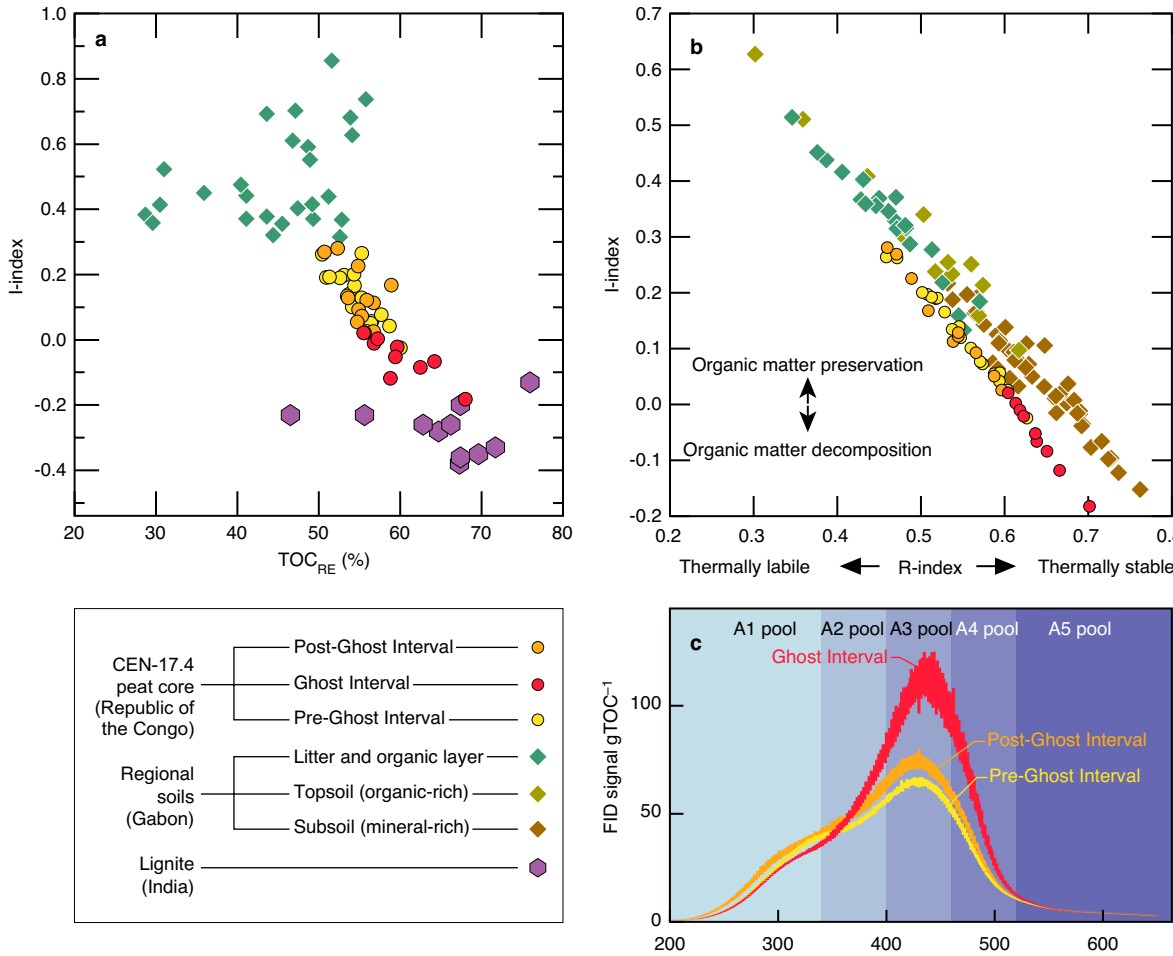

**Extended Data Fig. 1 | Rock-Eval data as a proxy for the preservation versus decomposition status of peat core CEN-17.4 compared to soils of the region and lignite. a**, Rock-Eval I/TOC diagram of considered mineral-free samples including peat core CEN-17.4, regional organic layers, soils and lignite[9] (key in bottom left box). $TOC_{RE}$ is Rock-Eval TOC (see Methods). **b**, Rock-Eval I/R diagram of peat core CEN-17.4 compared to regional soil data[9] and its interpretation. As the I-index is inversely correlated with the R-index, which relates to the thermally resistant and refractory pools of organic matter[9], this suggests a stabilization of the organic matter through the progressive decomposition of labile organic compounds and the relative enrichment in resistant and refractory ones in CEN-17.4. As observed in compost[91], organic layers and soils[9,92–94], a decrease in thermally labile pools results in a concomitant increase in more thermally stable pools. **c**, Thermograms (mean values and standard deviation) of core CEN-17.4 for the Pre-Ghost Interval, the Ghost Interval and the Post-Ghost Interval, overlying the thermal bounds used to characterize the highly labile (A1) to the strongly refractory (A5) pools of organic matter.

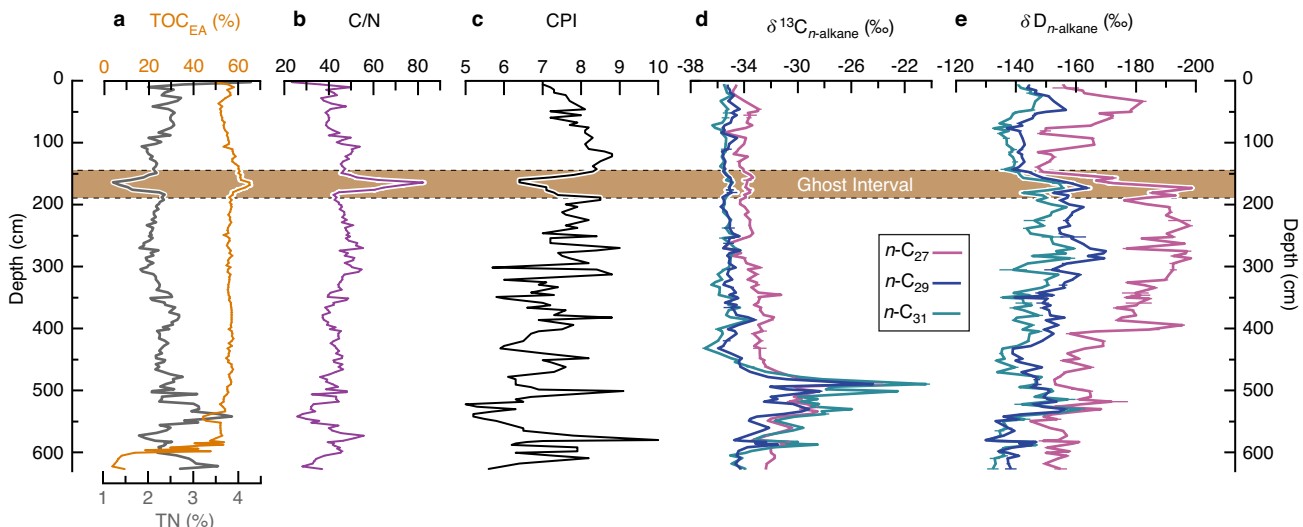

**Extended Data Fig. 2 | Peat core CEN-17.4 bulk organic and plant wax data.**
**a**, Total organic carbon ($TOC_{EA}$) and total nitrogen (TN). **b**, Carbon-to-nitrogen ratios (C/N). **c**, Carbon Preference Index (CPI). Note that the marked decrease in CPI during the Ghost Interval supports the increase in decomposition[95,96]. **d**, Carbon isotopes of long-chain $n$-alkanes ($\delta^{13}C_{n\text{-alkane}}$). **e**, Hydrogen isotopes of long-chain $n$-alkanes ($\delta D_{n\text{-alkane}}$). Error bars based on replicate analyses represent the $1\sigma$ uncertainty and are often too small to distinguish from the lines. Horizontal brown band bounded by dashed lines denotes the Ghost Interval.

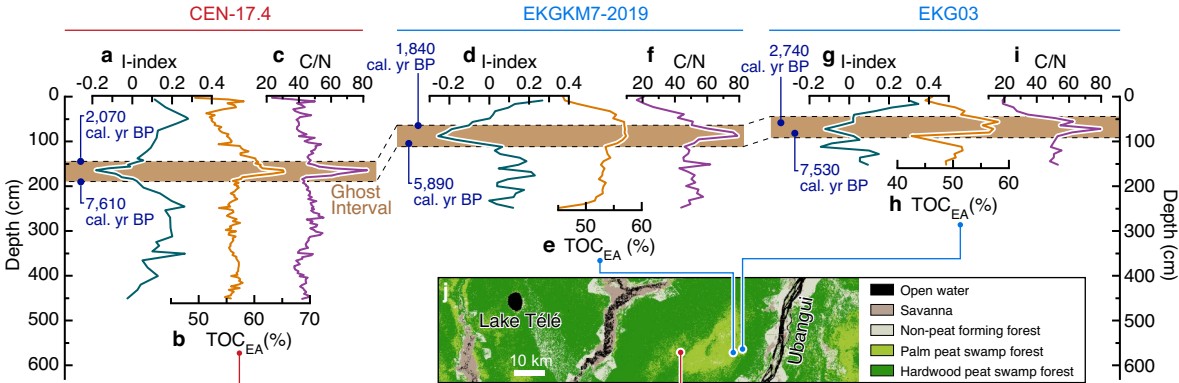

**Extended Data Fig. 3 | Spatial representativeness of the Ghost Interval within the peat-filled Ekolongouma basin.** Geochemical properties of core CEN-17.4 (**a**–**c**), core EKGKM7-2019 (**d**–**f**) and core EKG03 (**g**–**i**). **a**,**d**,**g**, I-index. **b**,**e**,**h**, Total organic carbon (TOC$_{EA}$). **c**,**f**,**i**, Carbon-to-nitrogen (C/N) ratios. Horizontal brown band denotes the Ghost Interval, defined from coeval changes in geochemical proxies (see text); dashed lines show proposed stratigraphic correlations. Dark blue dots show calibrated radiocarbon dates available at or near the boundaries of the Ghost Interval. **j**, Location map derived from ref. [2].

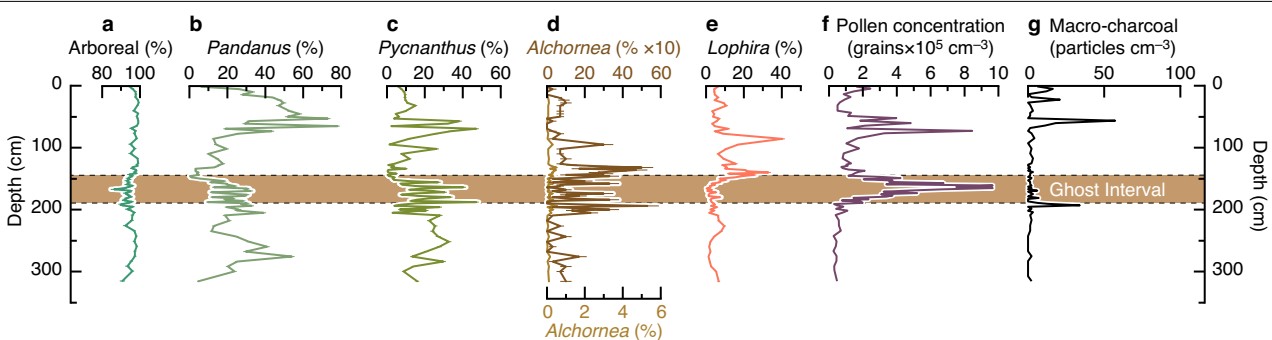

**Extended Data Fig. 4 | Core CEN-17.4 pollen and macro-charcoal data.**
a–e, Pollen taxa, arboreal (a), swamp forest-associated taxa: *Pandanus* (b) and *Pycnanthus* (c), and light-demanding taxa: *Alchornea* (d) and *Lophira* (e) contributing to the sum of pollen counted. f, Pollen concentration. g, Macro-charcoal (>150 µm) concentration. Horizontal brown band bounded by dashed lines denotes the Ghost Interval.

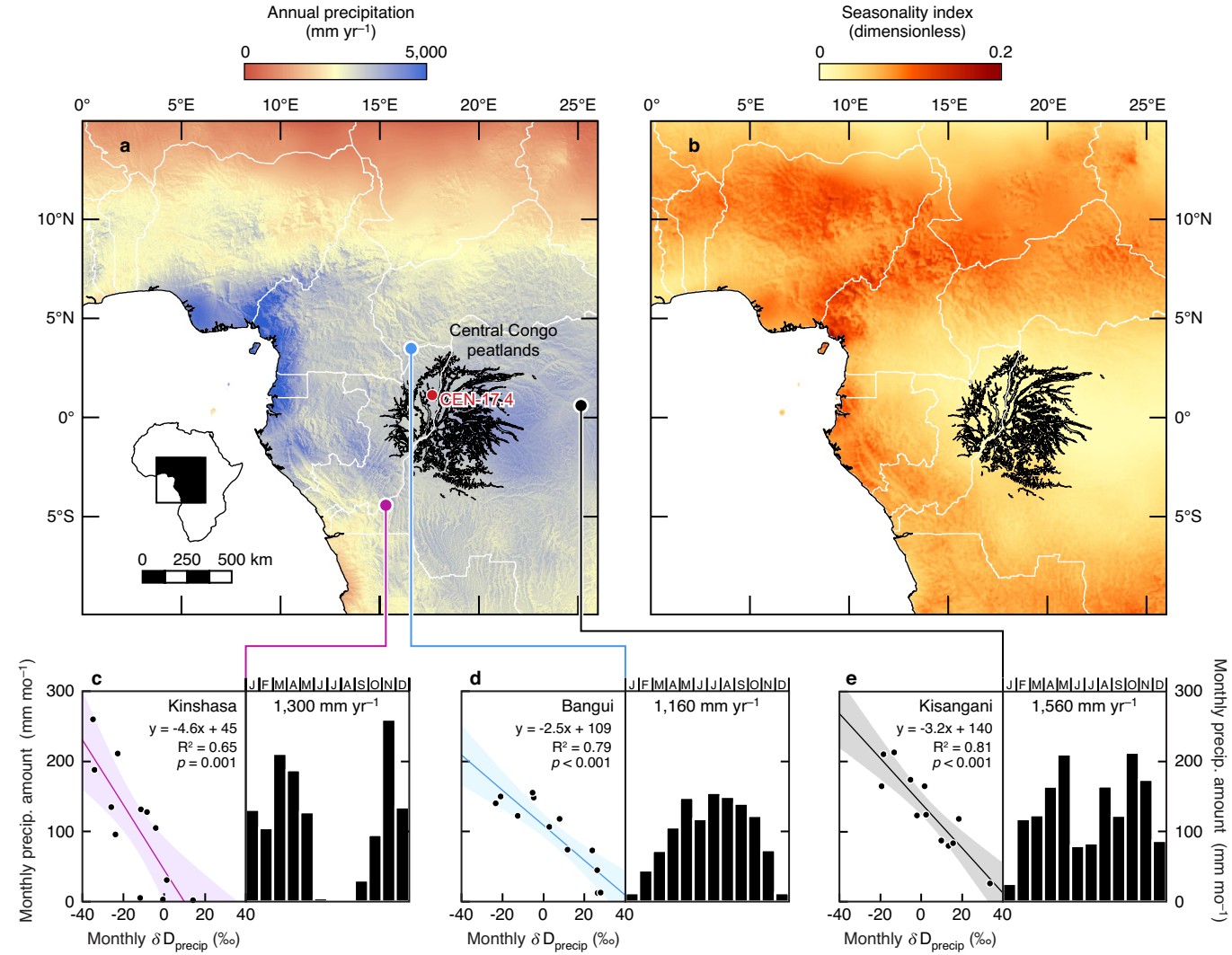

**Extended Data Fig. 5 | Climatology of central Africa including the central Congo Basin peatlands. a,b,** Amount of precipitation and seasonality index[48], respectively, derived from the CHELSA data[49]. Black line shows central Congo peatlands extent[2]. Red dot shows core CEN-17.4 location. **c–e,** Relationship δD$_{precip}$–precipitation amount (at monthly scale) for Kinshasa, Bangui and Kisangani (all three showing significant correlations, colour-coded with the sample location dots) based on GNIP data[87]. Filled envelopes represent the 95% confidence intervals of regression lines.

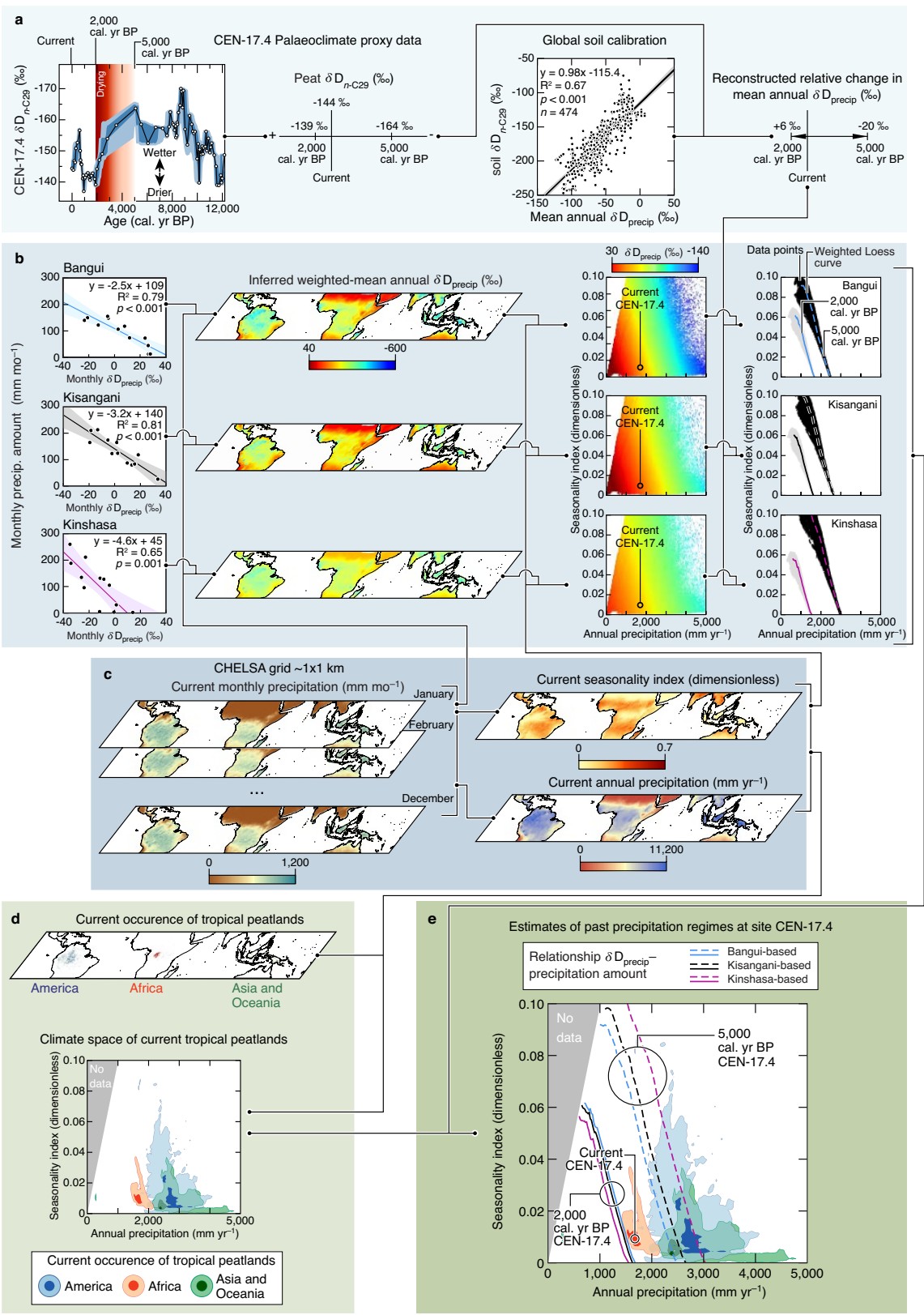

**Extended Data Fig. 6** | See next page for caption.

**Extended Data Fig. 6 | Schematic of the workflow used to compute estimates of past changes in precipitation regimes at the CEN-17.4 site using different relationships between $\delta D_{precip}$ and precipitation amount.** The code, performed using open source Jupyterhub notebooks, follows three main steps: 1) computation of weighted-mean annual $\delta D_{precip}$ values for the continental tropics based on local relationships between $\delta D_{precip}$ and precipitation amount, 2) computation of past changes in precipitation regimes using inferred $\delta D_{precip}$ from palaeoclimate proxy data, and 3) comparison between modern data and palaeoclimate reconstruction in a climate space. **a**, Reconstruction of mean annual $\delta D_{precip}$ derived from the peat plant wax $\delta D_{n\text{-}C29}$ data for both the pre-drying period (at 5,000 cal. yr BP) and the end of the drying trend (at 2,000 cal. yr BP) using a global surface soil calibration relationship[80]. **b**, Computation of weighted-mean annual $\delta D_{precip}$ for the continental tropics inferred using the modern relationships $\delta D_{precip}$– precipitation amount from Kinshasa, Bangui and Kisangani[87], plotted in climate space (precipitation amount, seasonality index), and with extracted $\delta D_{precip}$ values at 5,000 cal. yr BP (smoothed using a Loess function, dashed lines) and at 2,000 cal. yr BP (smoothed using a Loess function, solid lines). **c**, Computation of precipitation amount and seasonality index[48] for the continental tropics derived from the CHELSA dataset[49]. **d**, Spatial extent of tropical peatland areas derived from the PEATMAP dataset[50] and associated climate space. **e**, Same as **d**, including the solutions for amount and seasonality of precipitation derived from peat $\delta D_{n\text{-}C29}$ values at 5,000 cal. yr BP (dashed lines) and at 2,000 cal. yr BP (solid lines). Dark-coloured and light-coloured surfaces in **d** and **e** indicate 68% and 95% contours, respectively, from the KDEs peak density.

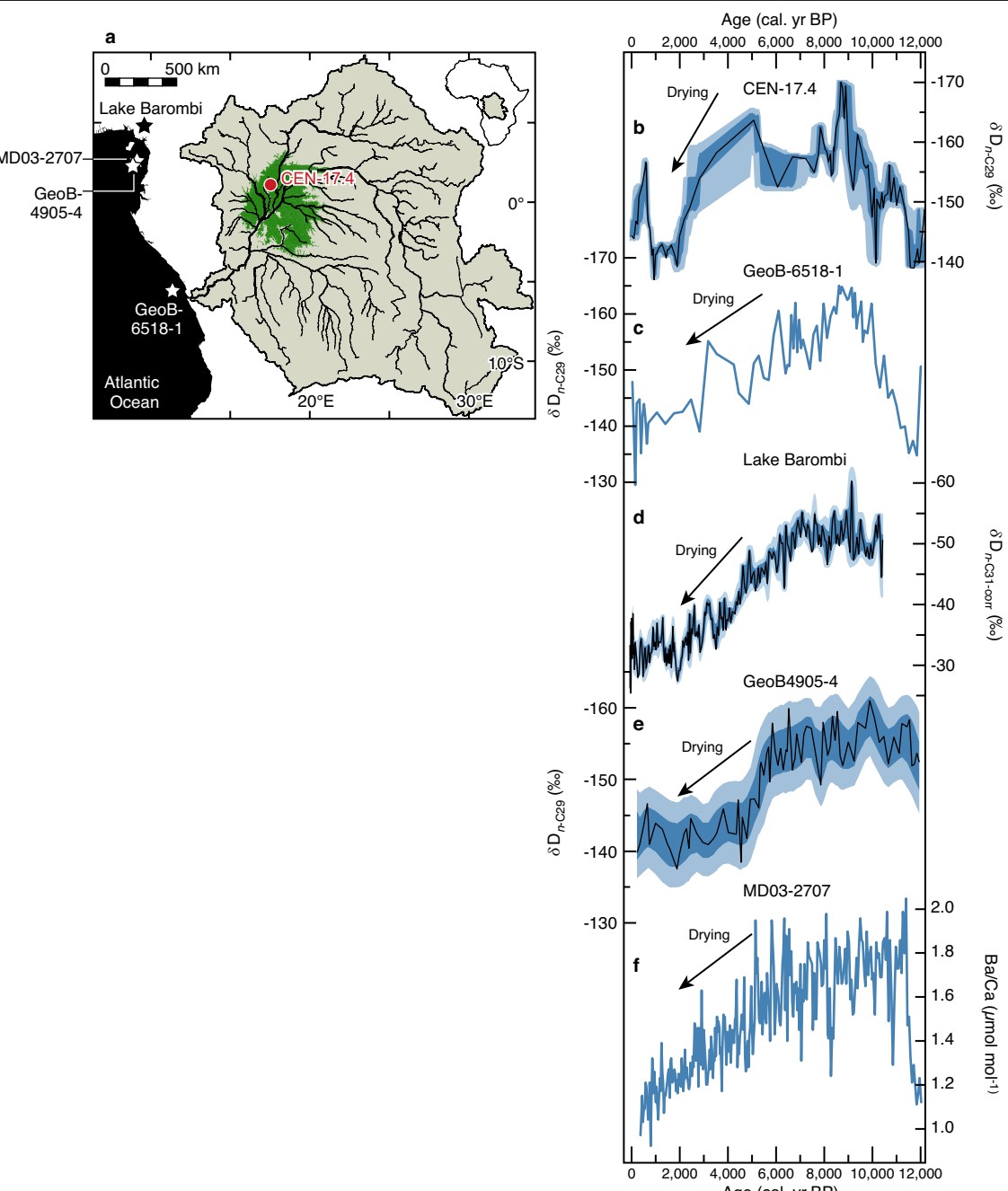

**Extended Data Fig. 7 | Records of hydrological change in western central Africa during the Holocene. a**, Location map including the central Congo peatlands[2] (green), the perimeter of the Congo Basin (black line), peat core CEN-17.4 (red dot; this study), Lake Barombi core[31] (black star), and marine cores GeoB6518-1[19,38], GeoB4905-4[30], and MD03-2707[29] (white stars). **b**, Core CEN-17.4 $\delta D_{n\text{-}C29}$ (this study). **c**, Core GeoB6518-1 $\delta D_{n\text{-}C29}$ (ref. [19]). **d**, Lake Barombi $\delta D_{n\text{-}C31}$ (vegetation adjusted)[31]. **e**, Core GeoB4905-4 $\delta D_{n\text{-}C29}$ (ref. [30]). **f**, Core MD03-2707 planktonic foraminiferal Ba/Ca (a proxy for runoff)[29]. Lines are interpolated data; envelopes reflect 68% (dark) and 95% (light) confidence intervals in the reconstructions, based on analytical and age model errors.

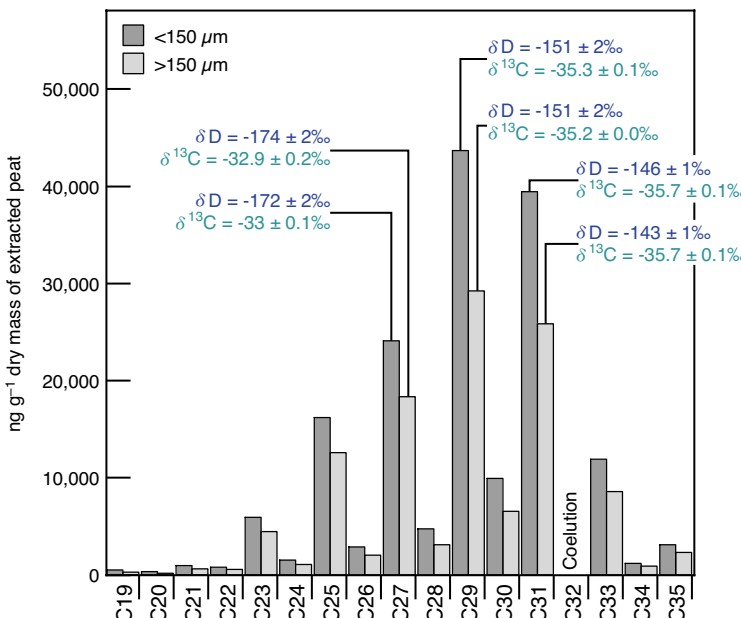

**Extended Data Fig. 8 | Concentration, distribution, δD and δ¹³C of *n*-alkanes in different size fractions.** Coarse fraction >150 μm (including roots) and fine fraction <150 μm were extracted from sample depth 296.5 cm. *n*-Alkanes in both fractions show similar distribution and have the same δD and δ¹³C values.

**Extended Data Table 1 | ¹⁴C dataset of peat cores CEN-17.4, LOK5-5 and BDM1-7 used for the age–depth models, and start and end ¹⁴C dates of the Ghost Interval from peat cores EKGKM7-2019 and EKG03**

| Core name | Depth (cm) | Lab code | Material | ¹⁴C age (yr BP) | ¹⁴C age error (yr) | Median age cal. yr BP (IntCal20/SHCal20 curve)[b] | Age range (cal. yr BP; 95.5%)[b] |
|---|---|---|---|---|---|---|---|
| CEN-17.4 | 0.5 | AWI-2567.1.1 | Fine fraction (<150 $\mu m$) | 565 | 170 | 540 | 0-899 |
| CEN-17.4 | 12.8 | AWI-1820.1.1 | Fine fraction (<150 $\mu m$) | 345 | 60 | 390 | 158-500 |
| CEN-17.4 | 53.6 | AWI-2255.2.1 | Fine fraction (<150 $\mu m$) | 835 | 25 | 713 | 676-768 |
| CEN-17.4 | 94.4 | AWI-2568.1.1 | Fine fraction (<150 $\mu m$) | 1,240 | 75 | 1,134 | 974-1,282 |
| CEN-17.4 | 132.5 | AWI-2802.1.2 | Fine fraction (<150 $\mu m$) | 1,760 | 70 | 1,641 | 1,427-1,826 |
| CEN-17.4 | 151.5 | AWI-1821.1.1 | Fine fraction (<150 $\mu m$) | 2,120 | 60 | 2,069 | 1,893-2,306 |
| CEN-17.4 | 170.5 | AWI-2256.2.1 | Fine fraction (<150 $\mu m$) | 4,450 | 30 | 5,031 | 4,872-5,279 |
| CEN-17.4 | 190.5 | AWI-2569.1.1 | Fine fraction (<150 $\mu m$) | 6,770 | 85 | 7,610 | 7,431-7,776 |
| CEN-17.4 | 214.4 | AWI-2570.1.1 | Fine fraction (<150 $\mu m$) | 7,630 | 95 | 8,413 | 8,197-8,591 |
| CEN-17.4 | 235.6 | AWI-2571.1.1 | Fine fraction (<150 $\mu m$) | 7,640 | 110 | 8,424 | 8,183-8,635 |
| CEN-17.4 | 256.5 | AWI-2257.2.1 | Fine fraction (<150 $\mu m$) | 7,570 | 35 | 8,370 | 8,213-8,419 |
| CEN-17.4 | 296.5 | AWI-1677.2.1 | Fine fraction (<150 $\mu m$) | 8,020 | 140 | 8,864 | 8,482-9,286 |
| CEN-17.4 | 296.5 | AWI-1677.1.1 | Coarse fraction (>150 $\mu m$)[a] | 7,350 | 50 | 8,120 | 8,017-8,315 |
| CEN-17.4 | 356.5 | AWI-2803.1.1 | Fine fraction (<150 $\mu m$) | 9,140 | 50 | 10,285 | 10,196-10,485 |
| CEN-17.4 | 396.5 | AWI-2258.2.1 | Fine fraction (<150 $\mu m$) | 8,360 | 35 | 9,361 | 9,148-9,473 |
| CEN-17.4 | 418.5 | AWI-2572.1.1 | Fine fraction (<150 $\mu m$) | 9,930 | 40 | 11,301 | 11,215-11,608 |
| CEN-17.4 | 460.5 | AWI-1822.1.1 | Fine fraction (<150 $\mu m$) | 10,150 | 50 | 11,751 | 11,357-11,936 |
| CEN-17.4 | 498.5 | AWI-2573.1.1 | Fine fraction (<150 $\mu m$) | 12,450 | 40 | 14,569 | 14,293-14,913 |
| CEN-17.4 | 540.5 | AWI-2259.1.1 | Fine fraction (<150 $\mu m$) | 13,650 | 45 | 16,463 | 16,299-16,637 |
| CEN-17.4 | 579.7 | AWI-2574.1.1 | Fine fraction (<150 $\mu m$) | 14,750 | 85 | 18,051 | 17,824-18,233 |
| CEN-17.4 | 596.4 | AWI-1823.1.1 | Fine fraction (<150 $\mu m$) | 11,550 | 55 | 13,401 | 13,303-13,567 |
| CEN-17.4 | 599.1 | AWI-2575.1.1 | Fine fraction (<150 $\mu m$) | 16,850 | 85 | 20,350 | 20,116-20,534 |
| CEN-17.4 | 625.2 | AWI-1824.1.2 | Fine fraction (<150 $\mu m$) | 14,050 | 55 | 17,071 | 16,931-17,330 |
| LOK5-5 | 45 | SUERC-101947 | Fine fraction (<180 $\mu m$) | 870 | 35 | 751 | 680-898 |
| LOK5-5 | 84 | SUERC-101948 | Fine fraction (<180 $\mu m$) | 1,680 | 35 | 1,556 | 1,421-1,693 |
| LOK5-5 | 105 | SUERC-99681 | Fine fraction (<180 $\mu m$) | 2,270 | 35 | 2,227 | 2,145-2,343 |
| LOK5-5 | 145 | SUERC-101949 | Fine fraction (<180 $\mu m$) | 3,320 | 35 | 3,521 | 3,411-3,633 |
| LOK5-5 | 185 | SUERC-101950 | Fine fraction (<180 $\mu m$) | 3,500 | 35 | 3,757 | 3,640-3,869 |
| LOK5-5 | 225 | SUERC-101951 | Fine fraction (<180 $\mu m$) | 6,710 | 35 | 7,560 | 7,477-7,659 |
| LOK5-5 | 265 | SUERC-99682 | Fine fraction (<180 $\mu m$) | 6,670 | 40 | 7,527 | 7,431-7,584 |
| LOK5-5 | 445 | SUERC-99683 | Fine fraction (<180 $\mu m$) | 8,650 | 40 | 9,594 | 9,533-9,689 |
| LOK5-5 | 595 | SUERC-99687 | Fine fraction (<180 $\mu m$) | 9,290 | 40 | 10,448 | 10,290-10,574 |
| BDM1-7 | 49.5 | SUERC-94360 | Fine fraction (<180 $\mu m$) | 750 | 35 | 671 | 568-726 |
| BDM1-7 | 120.5 | SUERC-99647 | Fine fraction (<180 $\mu m$) | 1,560 | 35 | 1,419 | 1,352-1,522 |
| BDM1-7 | 167.5 | SUERC-99648 | Fine fraction (<180 $\mu m$) | 2,510 | 35 | 2,581 | 2,371-2,726 |
| BDM1-7 | 236.5 | SUERC-99649 | Fine fraction (<180 $\mu m$) | 5,990 | 40 | 6,807 | 6,675-6,936 |
| BDM1-7 | 289.5 | SUERC-94361 | Fine fraction (<180 $\mu m$) | 6,230 | 35 | 7,093 | 6,992-7,251 |
| BDM1-7 | 391.5 | SUERC-99650 | Fine fraction (<180 $\mu m$) | 8,500 | 40 | 9,499 | 9,439-9,538 |
| BDM1-7 | 503.5 | SUERC-99651 | Fine fraction (<180 $\mu m$) | 9,350 | 40 | 10,539 | 10,404-10,687 |
| BDM1-7 | 569.5 | SUERC-94362 | Fine fraction (<180 $\mu m$) | 16,000 | 65 | 19,296 | 19,113-19,485 |
| EKGKM7-2019 | 64.5 | SUERC-101850 | Fine fraction (<180 $\mu m$) | 1,930 | 35 | 1,838 | 1,740-1,926 |
| EKGKM7-2019 | 104.5 | SUERC-101851 | Fine fraction (<180 $\mu m$) | 5,150 | 35 | 5,886 | 5,749-5,989 |
| EKG03 | 58.5 | AWI-8639.1.1 | Fine fraction (<180 $\mu m$) | 2,620 | 30 | 2,741 | 2,541-2,778 |
| EKG03 | 80.5 | SUERC-101849 | Fine fraction (<180 $\mu m$) | 6,670 | 35 | 7,528 | 7,431-7,583 |

[a]Not used for the age/depth model.
[b]Calibrated using Calib online Version 8.20 (http://calib.org/calib/calib.html)[97].

**Extended Data Table 2 | Summary of four hypothesized scenarios, which, individually or in combination, could explain the Ghost Interval**

| Scenarios | Potential drivers | Hydrosystem responses | Ecosystem responses | Peat responses | Observed relevant palaeo-proxy evidences | Likeliness |
|---|---|---|---|---|---|---|
| 1) Reduced inputs. Reduced peat accumulation between ~7,500 and ~2,000 cal. yr BP, due to low litter inputs and reduced formation of new peat. | Gradual hydro-climate drying. | Gradual water table lowering. | Gradual reduction of swamp forest taxa replaced with taxa adapted to or tolerant of drier conditions. | Slow peat accumulation. | Gradual decrease in swamp forest pollen taxa and increase in $D_{n-C29}$ values (hydroclimate drying). But conditions were still wet until ~5,000 cal. yr BP and flooded and terra firme forests do not differ in litter production. | Little evidence at site CEN-17.4, given the relative enrichment of pollen and TOC concentration during Ghost Interval. |
| 2) Increased contemporaneous decomposition. Reduced peat accumulation due to increased contemporaneous decomposition between ~7,500 and ~2,000 cal. yr BP due to increasingly dry conditions. | Gradual hydro-climate drying. | Gradual water table lowering. | Gradual reduction of swamp forest taxa replaced with taxa adapted to or tolerant of drier conditions. | Oxic conditions drive contemporaneous peat decomposition. | Change in decomposition-affected proxies (I-index, C/N ratio, TOC and $^{13}C_{TOC}$) well before 2,000 cal. yr BP and absence of erosional unconformity. Swamp forest pollen taxa decreased and $D_{n-C29}$ values increased (hydroclimatic drying) from ~5,000 cal. yr BP onwards. | Good evidence that this occurred at site CEN-17.4, but this process cannot explain all trends (i.e., decomposition of peat accumulated during wet conditions before 5,000 cal. yr BP). |
| 3) Secondary decomposition. Post-accumulation removal of old peat through a subsequent increase in decomposition at ~2,000 cal. yr BP, caused by more severe drying that deepened water tables. | Marked hydro-climate drying. | Marked water table lowering. | Marked reduction of swamp forest taxa replaced with taxa adapted to or tolerant of drier conditions. | Oxic conditions drive decomposition of previously accumulated peat. | Change in decomposition-affected proxies (I-index, C/N ratio, TOC and $^{13}C_{TOC}$) extending into peat deposited under wet conditions (7,500 to 5,000 cal. yr BP). Swamp forest pollen taxa at their lowest level at ~2,000 cal yr. BP, and increase in $D_{n-C29}$ values (hydroclimatic drying) show greatest drying at ~2,000 cal. yr BP. | Strongest evidence that this occurred at site CEN-17.4 explaining many features of the data. |
| 4) Secondary removal. Physical removal of a section of previously accumulated peat at ~2,000 cal. yr BP, by an event such as deep burning, fluvial erosion, or anthropogenic disturbance. | Marked peat drying and fluvial erosion in next wet season. | Marked dry season. | Marked disappearance of swamp forest taxa replaced with taxa adapted to or tolerant of drier conditions. | Physical erosion of previously accumulated peat. | No presence of erosional unconformity at site CEN-17.4. | No evidence of this having occurred at CEN-17.4 (but may have occurred at peatland margins resulting in pre-aged carbon in marine sediments after 5,000 cal. yr BP). |
| | Deep natural burning or anthropogenic deforestation and/or drainage. | Marked water table lowering. | Marked disappearance of swamp forest taxa replaced with taxa adapted to or tolerant of drier conditions. | Physical erosion of previously accumulated peat including peat burning. | No presence of erosional unconformity nor presence of any charcoal, ash layer, and charred peat at site CEN-17.4. | Not occurring at site CEN-17.4. |

# Reporting Summary

## Statistics

For all statistical analyses, confirm that the following items are present in the figure legend, table legend, main text, or Methods section.

| n/a | Confirmed | |
|-----|-----------|---|
| ☐ | ☒ | The exact sample size (*n*) for each experimental group/condition, given as a discrete number and unit of measurement |
| ☐ | ☒ | A statement on whether measurements were taken from distinct samples or whether the same sample was measured repeatedly |
| ☐ | ☒ | The statistical test(s) used AND whether they are one- or two-sided<br>*Only common tests should be described solely by name; describe more complex techniques in the Methods section.* |
| ☒ | ☐ | A description of all covariates tested |
| ☒ | ☐ | A description of any assumptions or corrections, such as tests of normality and adjustment for multiple comparisons |
| ☐ | ☒ | A full description of the statistical parameters including central tendency (e.g. means) or other basic estimates (e.g. regression coefficient) AND variation (e.g. standard deviation) or associated estimates of uncertainty (e.g. confidence intervals) |
| ☒ | ☐ | For null hypothesis testing, the test statistic (e.g. *F*, *t*, *r*) with confidence intervals, effect sizes, degrees of freedom and *P* value noted<br>*Give P values as exact values whenever suitable.* |
| ☒ | ☐ | For Bayesian analysis, information on the choice of priors and Markov chain Monte Carlo settings |
| ☒ | ☐ | For hierarchical and complex designs, identification of the appropriate level for tests and full reporting of outcomes |
| ☒ | ☐ | Estimates of effect sizes (e.g. Cohen's *d*, Pearson's *r*), indicating how they were calculated |

*Our web collection on statistics for biologists contains articles on many of the points above.*

## Software and code

Policy information about availability of computer code

| Data collection | For collection of the hydrogen and carbon isotopes of plant waxes, ISODAT software was used. |
|---|---|
| Data analysis | The age-depth models were performed using open source rbacon 2.5.7 package running in R 4.1.0 (https://CRAN.R-project.org/package=rbacon). The geospatial analyses and mapping were performed using open source Jupyterhub notebooks (5.7.8; https://jupyter.org/) running Python 3.7.3. |

For manuscripts utilizing custom algorithms or software that are central to the research but not yet described in published literature, software must be made available to editors and reviewers. We strongly encourage code deposition in a community repository (e.g. GitHub). See the Nature Portfolio guidelines for submitting code & software for further information.

## Data

Policy information about availability of data

All manuscripts must include a data availability statement. This statement should provide the following information, where applicable:
- Accession codes, unique identifiers, or web links for publicly available datasets
- A description of any restrictions on data availability
- For clinical datasets or third party data, please ensure that the statement adheres to our policy

Data that support the findings of this study are available in the PANGAEA repository: https://doi.pangaea.de/10.1594/PANGAEA.938019. Codes for age-depth models and for processing and analysis of geospatial data (climate spaces, tropical peatland distribution and precipitation reconstruction) are available in the IRD Dataverse repository: https://doi.org/10.23708/FO2HGM.

# Field-specific reporting

Please select the one below that is the best fit for your research. If you are not sure, read the appropriate sections before making your selection.

☐ Life sciences  ☐ Behavioural & social sciences  ☒ Ecological, evolutionary & environmental sciences

For a reference copy of the document with all sections, see nature.com/documents/nr-reporting-summary-flat.pdf

# Ecological, evolutionary & environmental sciences study design

All studies must disclose on these points even when the disclosure is negative.

| | |
|---|---|
| Study description | This study describes the palaeoenvironmental and palaeohydroclimatic history of the central Congo peatlands, the world's largest tropical peatland complex. |
| Research sample | This study primarily reports on the analyses of peat cores. Palaeoenvironmental proxies (organic matter properties), preserved pollen and palaeohydrological proxies (hydrogen isotopes of plant waxes) were analysed.<br><br>For the geospatial analyses, climatologies were derived from the CHELSA data Version 1.2 publicly available at https://chelsa-climate.org/downloads/ and the current geographical extents of the main peat-bearing tropical regions, including Africa, Southeast Asia/Oceania and America were derived from the PEATMAP dataset publicly available at https://archive.researchdata.leeds.ac.uk/251/. |
| Sampling strategy | Sampling strategy is described in the Methods section. |
| Data collection | All data collected were analysed and contributed to the final conclusions. |
| Timing and spatial scale | Fieldwork and initial data collection occurred during field seasons in the Republic of Congo and in the Democratic Republic of the Congo from 2014 to 2020. All presented analyses were conducted in an intermittent manner from 2017 onwards and were completed in 2021. |
| Data exclusions | No significant data have been excluded from this study. |
| Reproducibility | Accuracy and precision of the hydrogen isotopes of plant waxes were controlled by a lab internal n-alkane standard calibrated against the A4-Mix isotope standard (A. Schimmelmann, University of Indiana) every six measurements and by the daily determination of the H3+ factor. Measurement precision was determined by calculating the difference between the analysed values of each standard measurement and the long-term mean of standard measurements, which yielded a 1σ error of <3‰. H3+ factors varied between 4.9 and 5.2 (5.1 ±0.1). Accuracy and precision of the squalane internal standard were both 2‰ (n=238). Precision of the replicate analyses of the n-C27, n-C29 and n-C31 alkanes was 1‰ on average.<br>Accuracy and precision of the carbon isotopes of plant waxes were determined by measuring n-alkane standards calibrated against the A4-Mix isotope standard every six measurements. The difference between the long-term means and the measured standard values yielded a 1σ error of <0.3‰. Accuracy and precision of the squalane internal standard were both 0.2‰ (n = 264). Precision of the replicate analyses of the n-C27, n-C29 and n-C31 alkanes was 0.1‰ on average. |
| Randomization | This does not apply to this study as no experiment was conducted. |
| Blinding | This does not apply to this study as no experiment was conducted. |
| Did the study involve field work? | ☒ Yes  ☐ No |

## Field work, collection and transport

| | |
|---|---|
| Field conditions | The study sites are in the central Congo Basin. The climate and vegetation are described in the manuscript and it the Methods section. |
| Location | Cores CEN-17.4 (1° 11' 0.49" N, 17° 38' 23.7" E), EKGKM7-2019 (1° 10' 57.29" N, 17° 48' 18.4" E) and EKG03 (1° 11' 17.7" N, 17° 49' 53.29" E) were collected in a peat-filled interfluvial basin, informally named Ekolongouma, situated between the Likouala-aux-Herbes and Ubangui rivers, in the Likouala Department, Republic of the Congo. Core LOK5-5 (at Lokolama; 0° 19' 36.62" S, 18° 10' 24.37" E) was collected in an interfluvial basin close to the Congo River in the Democratic Republic of the Congo, 177 km from core CEN-17.4, and core BDM1-7 (at Bondamba; 0° 10' 24.38" S, 19° 41' 45.56" E), was collected in a river-influenced valley-floor peatland (not an interfluvial basin) close to the Ruki River, a tributary of the Congo River in Democratic Republic of the Congo, 274 km from core CEN-17.4. |
| Access & import/export | Fieldwork was permitted by the governments of the Republic of the Congo and of the Democratic Republic of the Congo. The samples were collected and exported with permission of the Republic of the Congo and of the Democratic Republic of the Congo, and imported in the EU under licence. |
| Disturbance | Minor disturbances occured for few peat sections, which were compacted during transportation, showing a depth reduction of up to 5 cm per 50 cm. The depth of each sample taken was restored on an undistorted depth scale. |

# Reporting for specific materials, systems and methods

We require information from authors about some types of materials, experimental systems and methods used in many studies. Here, indicate whether each material, system or method listed is relevant to your study. If you are not sure if a list item applies to your research, read the appropriate section before selecting a response.

## Materials & experimental systems

| n/a | Involved in the study |
|-----|----------------------|
| ☒ | ☐ Antibodies |
| ☒ | ☐ Eukaryotic cell lines |
| ☒ | ☐ Palaeontology and archaeology |
| ☒ | ☐ Animals and other organisms |
| ☒ | ☐ Human research participants |
| ☒ | ☐ Clinical data |
| ☒ | ☐ Dual use research of concern |

## Methods

| n/a | Involved in the study |
|-----|----------------------|
| ☒ | ☐ ChIP-seq |
| ☒ | ☐ Flow cytometry |
| ☒ | ☐ MRI-based neuroimaging |

