## [Peer Review File · Nature]

Manuscript Title: Hydroclimatic vulnerability of peat carbon in the central Congo Basin

Reviewer Comments & Author Rebuttals

Reviewer Reports on the Initial Version:

Referees' comments:

Referee #1 (Remarks to the Author):

This study provides a detailed analysis of palaeoenvironmental proxies from a 6.29m peat core from forested peatlands in the Republic of Congo. It finds a period of low rates of net peat accumulation, referred to as the "Ghost Interval", and presents evidence this is driven by a drier climate. The study does a good job of placing the changes in regional context through time. It also compares to climates of other peatlands around the world and discusses potential future implications of the changes.

Although this is the first core to be analyzed in detail from the Cuvette Centrale peatlands of the Congo, similar analyses have been conducted in other regions, including documentation of periods of hiatus in peat accumulation, changes in accumulation rates, and changes in vegetation community, linked to variation in climate over time in tropical peatlands in South America and Southeast Asia. Most of these detailed peat core analyses are published in journals relevant to the specialist audience, such as *Global Change Biology*; *Palaeogeography, Palaeoclimatology, Palaeoecology*; *Quaternary Science Reviews*; *Journal of Geophysical Research: Biogeosciences*, etc. Similarly, this manuscript is likely to primarily be of interest to a more specialist audience, and may benefit from a longer format to show more of the data and comparisons to supporting cores. I would therefore recommend publication in *Nature Climate Change*, *Nature Geoscience*, *Global Change Biology*, or another disciplinary journal.

Major comments/concerns:

Emphasis on peat loss and C source is not supported - There is frequent reference to "net peat loss" and switch to a C source during the Ghost Interval. For example, in the abstract: "...triggered the switch from net peat accumulation to net peat loss" and in the conclusions "likely causing a shift from a carbon sink ...to a carbon source..." (269-270). This emphasis seems sensationalized. The Ghost Interval is defined as the period from ~7,300 to ~2,000 cal yr BP, and there is cumulative peat accumulation of ~50cm to ~>1m across all dated cores over this time period (Fig 1, Fig ED3). There might have been a more limited period of peat loss within this time window, but over the entire Ghost Interval, there is still net peat accumulation. Relatedly, over each shorter individual dated interval in the main core (Fig 1), there is still net peat accumulation. Slower accumulation rates of highly decomposed peat, but still net accumulation. There is certainly a possibility that the peatland was a C source for a shorter fraction of the Ghost Interval period, but the authors should not imply a massive C source over this period.

Ghost Interval inconsistently presented - The Ghost interval is inconsistently referenced in figures,

emphasizing key transitions in each, but not in agreement. This makes comparisons across figures more confusing for the reader. The Ghost Interval is frequently defined as the period from ~7,300 to ~2,000 cal yr BP (line 111), as shown in Fig 2. However, in Fig 3, the Ghost Interval shading only includes the dry sub-interval of this period. In Fig 1 it is not consistent across cores. Also, inconsistent in ED Figures.

Extrapolation via secondary core data- The analysis of the primary core is very thorough and well-presented, and serves as the basis for a theory of regional change. However, the applicability of the theory beyond the single core is a crucial point that needs to be better supported. The data from secondary cores may support the proposed theory, but there are also inconsistencies. Additionally, it is hard to assess the relationship between the trends in these cores, due to the limited data presented, and the way it is displayed. More details below.

-Fig 1c and Fig ED3 – Substantial peat accumulation in secondary cores during Ghost Interval - Why did the peat accumulate so much during the Ghost interval at the other sites if it was a period of enhanced decomposition or peat loss?

It seems in some of these cores, the peat accumulated by as much as, or even over a meter, during the period from 7600 kyr BP to 2050kyrBP. For some of the cores, this is nearly half of the total peat accumulation at that site. Why would the peatland be expanding laterally or have higher rates of accumulation closer to the peat dome edge, during a dry period of peat loss in the interior? If the Ghost Interval is entirely driven by climate alone, then why would the peatland expand at all during this period? Could there also be some element of vegetation succession? Different hydrology across the dome? E.g. domed peatlands in Panama and SE Asia? Other explanation? Not clear if the secondary cores provide strong evidence for the Ghost Interval. This would benefit from further exploration in the manuscript, and bringing data from the secondary cores into the main manuscript.

-Misleading labeling - Labelling of the Ghost Interval on Fig 1 is misleading, suggesting it is more consistent than it is. Think about ways to plot the relevant data from all cores on the same time scale in some figure, to show what is happening over time in all cores. (see attached pdf for figure annotation)

-Number of cores – 5 supporting cores are mentioned (line 95), but only limited data is analyzed/presented, and not the same cores each time. Why are some cores (EKG4km, 7km and 9km) presented in Fig 1 with radiocarbon data, while cores EKG02 and 03 are presented in the extended data with RockEval data? Do you have radiocarbon and RockEval data from the same cores to show that the transition in carbon quality occurs over the same time period in all the cores? If not, could you add a few ¹⁴C dates to those cores?

Other comments/questions:

---For an interesting exploration of how tree falls in tropical peatlands can also cause major changes in apparent rates of peat accumulation, see Dommain et al, 2015 (JGR Biogeosciences), especially Fig 11. doi:10.1002/2014JG002796. Very relevant to detailed interpretation of peat cores in tropical peatlands. Can likely be ruled out with combination of wide range of datasets presented here, but may be relevant.

---Given variability in local conditions, trends need to be replicated across multiple cores – data

presentation needs strengthening.

---Although separated in space, all cores are from a single peat dome. Research in the inland peatlands of Pastaza-Maranon foreland basin in Peru with some similarities to the shows different peatlands can have different trajectories and responses, due to stages of vegetation succession, river meander, etc. in different parts of the peat complex

---This is likely a dynamic floodplain environment, which may have similarities to the peatland complex in the better studied Pastaza Maranon foreland basin. Any thoughts on the role of river meandering?

---Possible hiatus in peat accumulation and changes in vegetation community are also explored in Kelly et al (2017) <http://dx.doi.org/10.1016/j.palaeo.2016.11.039>

---Extended Data Table 2 – nice summary. Would be valuable in the main text.

--- Fig 1 – Relationship to Ghost Interval in other cores is not very clear. In Fig 1, the Ghost Interval is labeled as only ~50cm of the peat core, while in Fig ED3, the Ghost interval from the supporting cores is labeled as covering much of the analyzed core, perhaps over 1m of the peat core. What accounts for these differences?

---Fig 2: 2000-1000yrBP – Why did peat accumulation restart/continue in such extremely dry conditions?

---Fig ED3 – why did so much of the peat accumulate during the Ghost Interval at the site EKG03 and EKG02 if it was a drier period of enhanced decomposition? Why would the peatland be expanding to these areas of shallow peat during that period if peat was simultaneously being lost in the center of the peat dome? How do you reconcile the concurrent areal expansion and loss of peat depth?

---Were the climate reconstructions consistent across all cores? Show in SI?

---Ln 107: 4-6x slower accumulation is not net loss over the Ghost Interval. Avoid hyperbole.

---Ln 108-109: This statement and Ghost Interval labeling in Fig 1c is somewhat misleading.

---Ln 115-128: Nice job outlining possible scenarios and evidence for & against them.

---Ln 138: Two other peat cores? 5 supporting cores. What about other 3? Not analyzed? Or different trend?

---Very interesting ideas and comprehensive dataset. However, not yet convincingly supported at the basin scale by other cores.

---Ln 151-154: Interesting!

---Ln 170, 198, 225: What caused the sharp re-wetting ~2kyr BP from a climate perspective? Most discussion seems to focus on the drying trend, but the reversal is also of interest.

---Ln 199-203: Not enough explanation in the text here to follow what you did

---Ln 211-212: Also S. America (e.g. Kelly et al, 2017 PPP)

---Ln 215-217: Interesting point. Why would this be the case? Please explain and justify further.

---Ln 241: reduced area is a big claim. Why were peatlands accumulating substantially at other cores? Add more caveats?

---Ln 251: “may have been much larger in the past”. Or smaller? Likely varied through time, especially given wide variability in peat depth and basal ages.

---Ln 257: Not conclusive. “may” have been lost. “Lost” might not be the right word if it never formed due to enhanced decomposition/slower accumulation rates. Loss is a possibility, or hiatus in accumulation, as you also present.

---Ln 258: Would it be possible to date the other 2 supplemental cores? Could strengthen regional arguments.

---Ln 266-267: Abrupt massive C loss is not realistic or likely. Be careful not to accidentally imply it in the abstract – may confuse non-specialist readers.

---Ln 269-270: “most likely” – should this be “possibly”? Given net accumulation is still positive on average over every dated interval.

---Ln 286-288. Relevant work has addressed this issue but is not cited. e.g. Cobb et al. (2017). PNAS. See Fig 10.

[Editor note: please also see file attached]

Referee #2 (Remarks to the Author):

Summary of key results: In the forested swamps in central Congo, there may have been a drier period at 7000 (or 5000?) to 2000 cal yr BP, when the peat was more decomposed. This may have implications for future peat accumulation in the region.

Originality and significance: In principle, this is an original and valuable study, as it provides new evidence about past peat decomposition and accumulation dynamics and past climate changes in the forested swamps in central Congo basin.

Data and methodology: The biggest problem is that the study is based practically on one peat core, CEN-17.4. As the size of the forested peatland area is about 145 000 km², it should be obvious that no conclusions about the general trends in decomposition and peat accumulation rates can be made based on the results of a single core. The five secondary cores analyzed (earlier) with lower resolution do not really have any role in the paper (for example Figs. 2 and 3 show data only from CEN-17.4). We can compare this with the peat accumulation studies in the boreal and arctic peatlands, where regularly more than one core are analyzed from one bog or swamp, let alone from an area of 145 000 km².

The second main comment is about the classification of the pollen data. The authors divided the pollen taxa of the CEN-17.4 core into six vegetation types. But it is not explained how this was done. Pollen taxa are not the same as the plant taxa, and in most cases subjective decisions and major generalizations are needed for such classification of pollen taxa. It would thus be necessary to include a table, which would show how different pollen taxa were classified in this case.

Appropriate use of statistics: OK

Conclusions: Based on so scarce data, the conclusions are not feasible. While the topic is interesting, much more data based on peat cores needs to be generated before it is possible to make any general conclusions about the regional trends in peat decomposition and accumulation in the whole region.

Suggested improvements.: It would be good to be clearer with the timing of the period with high peat composition and possible dry period (termed “Ghost Interval” in the paper). It is stated in the abstract that this period dates to 7300-2000 cal yr BP and the same is indicated in Fig. 2. But in Fig. 3, which shows the pollen and δD data from the same core, the period is indicated at 5000-2000 cal yr BP. To some extent the authors state the factors which may explain the difference between the peat

decomposition data and the pollen/ δD data, but in any case the difference between Fig. 3 and the rest of the paper can be confusing for many potential readers.

Referee #3 (Remarks to the Author):

This manuscript presents the result of the first investigation into the Congo Basin peatlands Holocene history and the relationship between the peatland carbon accumulating function and hydroclimate. This is important because: 1) The Congo Basin represents the largest intact tropical peatland complex 2) There is little data on peatland development and resilience in the tropics 3) The fate of this peatland complex in the future very much matters in terms of carbon-cycle climate feedbacks.

The manuscript provides an original conclusion that merits publication. The manuscript and abstract are elegantly written, the introduction provides good context to the work.

The methodological approach is thorough and the manuscript presents an impressive amount of multi-proxy paleodata to pin down how climate is likely to have changed over time to explain variations in the peatland's carbon exchange with the atmosphere. I appreciate how the authors have moved away from attempting to quantify the change in the peatland's carbon sink - something that would need a modelling approach, meriting a separate publication - but have instead focussed the efforts in understanding through a multi-proxy approach how a dry period lead to contemporaneous and secondary decomposition of the peat carbon some 2000 years ago, leading to the eventual record of what they have called the Ghost Period. I particularly appreciated the description of the modern day climate space that shows how these peatlands are on the edge of the precipitation amount and seasonality distribution.

All of this impressive work is definitely appropriate for the aims of the investigation. The reporting of the data is also sufficiently detailed. Uncertainty in the data has been shown visually in figures or reported in the text.

In my view, these results will be of interest, not just to scientists in the discipline, but more widely to tropical ecologists (including those studying terra firme forests), climate scientists and soil scientists. The changes in climate would affect not just peatlands, but all other ecosystems in the Congo Basin.

The conclusions and data interpretation are robust, valid and reliable. The manuscript reference previous literature appropriately.

The quality of presentation of the data is very good. Here are some small suggestions for improvement of the figures:

Figure 1, panel B - the core labels are not legible, need to be changed to white (no border)? or another colour that allows reading.

Figure 3, Swamp forest data - visibility of the actual data points (at least in my computer) is poor - I suggest you either colour the 68% envelope in a lighter green (as you have done for the arboreal data) or make the data points a lighter colour (not black).

It would be good to have the data and code available publicly - it seems this will be done in time, but it would be good to make the links available when the manuscript is published.

Referee #4 (Remarks to the Author):

In this paper, Garcin, Schefuß, Dargie et al explore the vulnerability of a major wetland complex in the central Congo Basin to past changes in hydroclimate. To achieve this, they employ a multi-proxy approach (bulk geochemistry, organic geochemistry, palynology). The team find low accumulation rates between ca. 7000 to 2000 years ago. This interval (termed the “Ghost Interval”) is present in multiple cores and argued to result from enhanced peat decomposition (...and perhaps also post-depositional erosion). As this wetland holds ~30% of all tropical peat carbon (wow!), this has the potential to act as an important CO₂ source. Leaf wax hydrogen isotope ($\delta^2\text{H}$) values increase by ca. 30 per mil shift during this interval, suggesting that decomposition is associated with drier conditions. This implies that wetlands may release CO₂ under drier conditions, both in the past and future.

I really enjoyed reading this manuscript – it is well written and flowed logically. The figures are clear and concise. The findings have the potential to be genuinely important. The topic is also timely and has direct relevance to the future.

However, I did have quite a few questions. These are outlined below - I hope they help!

1) Estimating paleo-precipitation

To generate quantitative precipitation estimates, the authors must determine the relationship between $\delta^2\text{H}_{\text{wax}}$ and $\delta^2\text{H}_{\text{precip}}$. Here, the authors assume a positive linear relationship (i.e., slope = 1), between $\delta^2\text{H}_{\text{wax}}$ and mean annual $\delta^2\text{H}_{\text{precip}}$. However, various factors can affect the apparent fractionation between $\delta^2\text{H}_{\text{wax}}$ and $\delta^2\text{H}_{\text{precip}}$ (...and therefore, the resulting precipitation estimates). These factors include: i) soil evaporation, ii) leaf-water transpiration iii) wax biosynthesis.

I would argue that evaporative ²H-enrichment of soil water is relatively limited (especially in drylands). However, leaf water ²H-enrichment is far more variable and can depend on the species and the climatological setting (e.g., Daniels et al 2017, Feakins et al 2016a, Feakins & Sessions 2010, Kahmen et al 2013a). The apparent fractionation between $\delta^2\text{H}_{\text{wax}}$ and $\delta^2\text{H}_{\text{precip}}$ also varies widely (up to ~70 ‰) between plant life forms (i.e., trees, shrubs, forbs, graminoids) and physiological groups (i.e., C₃, C₄, CAM) (Sachse et al 2012). You observe a large decline in forest taxa from 50% to 20% during the Ghost interval. Can this explain changes in $\delta^2\text{H}$? One way to explore this is to use data from the same samples to calculate plant-specific fractionation factors (e.g., Feakins 2013; P3;

Inglis et al., 2020; P3). This is achievable - the authors have already generated pollen assemblage data.

Estimating the apparent fractionation between $\delta^2\text{H}_{\text{wax}}$ and $\delta^2\text{H}_{\text{precip}}$ must also account for any seasonal bias relative to annual rainfall – seasonal studies of modern plant ecohydrology reveal the seasonality of rainfall and plant uptake (Griepentrog et al 2019). As a “...reduction in total annual precipitation may have been accompanied by an increase in seasonality” (L207-208), this should also be considered by the authors.

2) Decoupling between hydrology and carbon cycling

The authors argue that drier conditions during the Ghost interval help to promote OC decomposition. This argument is based upon a shift towards high $\delta^2\text{H}$ values (-140 permil). However, there are intervals with similarly high values (e.g., 10-12 thousand years ago), but with NO evidence for enhanced OC decomposition. This is puzzling and suggest hydroclimate-carbon cycling decoupling in the early Holocene (...and hydroclimate-carbon cycling coupling in middle-to-late Holocene). This seems odd – why would this wetland complex only be susceptible to drying in the mid-to-late Holocene? (see also comment 4...). If this is true, what implications does this have for the future?

3) Non-hydrological controls

The authors argue that hydrology is the key control on OC oxidation – however, the authors should also acknowledge that temperature and respiration rates are closely coupled (e.g., <https://www.nature.com/articles/nature11205>) and may also play a role in our future warmer world.

4) Complex controls on peat decomposition

Various studies have already shown a link between hydroclimate and the carbon cycle in wetlands (especially in China). For example, Huang et al. (2018; Nature Comms.) use biomarkers to show that successive drying events have a big impact on the susceptibility of peat carbon stores to climate change. Swindles et al. (2018 Global Change Biology) also show that “droughts may lead to reduced C accumulation or even net loss of peat” in ombrotrophic domes (in Peru). However, peats are never simple and Swindles et al., 2018 show that “...future droughts may lead to phases of rapid carbon ACCUMULATION in some inundated tropical peat swamps”. In other words, drying does not necessarily equal decomposition.

Can the authors comment a bit more on the wetland architecture in their study site – are we looking at a raised ombrotrophic bog (i.e., drying = OC loss)? Or a flooded wetland (i.e., drying = OC accumulation)? A bit of both? And how might it evolve in the future?

5) More evidence needed for enhanced decomposition

The authors argue that there is enhanced decomposition in the Ghost Interval (scenarios 2/3) – this

is based on rock-eval pyrolysis, an increase in C/N ratios, a negative carbon isotope excursion in bulk OM, and higher TOC.

I have three questions here:

5.1. You associate higher TOC values with enhanced decomposition – however, I would expect to see LOWER TOC under enhanced decomposition. During the PETM we see very TOC values in continental sections, interpreted to represent enhanced soil OC oxidation (e.g. Denis et al., 2021; Baczynski et al., 2016;2019; Cotton et al., 2015). This also evident in biomarker distributions (e.g., changes in PAH distributions).

5.2. You associate higher C/N ratios with enhanced decomposition – however, shouldn't microbial consumption of C- and H-rich organic substances result in a decreased abundance of carbon relative to nitrogen (e.g., Hornibrook et al., 2000 *Geochemical Journal*). i.e, lower C/N ratios.

5.3. Changes in C/N could also be driven by changes in vegetation too (e.g., Hornibrook et al., 2000). Could this be a factor?

6) Quantifying CO₂ release

Assuming that this wetland complex is subject to widespread OC oxidation during the Ghost Interval, it has the potential to switch from a CO₂ sink to a CO₂ source. However, the authors do not provide any estimates for the potential CO₂ release. How much carbon could be released during the Ghost Interval? Without quantitative (or semi-quantitative) constraints, it is hard to assess if this is an important CO₂ source.

Minor comments:

L115-116: to refute this hypothesis you could calculate long-chain n-alkane mass accumulation rates. L221-222: and in China (Huang et al., 2018 *Nature Comms*) and Peru (Swindles et al., 2018; *Global Change Biology*).

L238-246: it would be nice to actually see this data in a figure to support your argument - the data from ref. 3 appears quite convincing.

Gordon Inglis

Author Rebuttals to Initial Comments:

NB: *The Reviewer's comments are printed in italic font (black). For reading ease our responses to the comments are printed in bold font (black). Passages in the manuscript (corrected or original) are printed in normal font (blue).*

We thank the reviewers for their comments, which have substantially improved the manuscript. There is one major and one minor addition of new data:

Major: We now include two additional ~6-m deep ¹⁴C dated peat cores from, Lokolama (LOK5-5) in Democratic Republic of the Congo, some 177 km to the southeast of the CEN-17.4 core, and Bondamba (BDM1-7) in Democratic Republic of the Congo, some 274 km to the east of the CEN-17.4 core (Fig. 1). These show that the Ghost Interval was a regional phenomenon. The addition of the two DRC cores means we have added new co-authors (Bart Crezee, Corneille Ewango, Joseph Kanyama, Pierre Bola and Ovide Emba), who obtained the cores and prepared samples from LOK5-5 and BDM1-7 in the Democratic Republic of the Congo. In response we have updated the radiocarbon calibration curve (using a mixed curve 50%:50% of north-south air-mass mixing derived from both the Northern Hemisphere curve (IntCal20) and the Southern Hemisphere curve (SHCal20)). This makes no appreciable difference to the dating, but is more robust given our sites are now just N and just S of the equator.

Minor: We also now report the same decomposition data from both CEN-17.4 and two secondary cores nearer the periphery of the Ekolongouma peatland, and have ¹⁴C dated the Ghost Interval for these cores (in Extended Data Fig. 3).

Referee #1 (Remarks to the Author):

*This study provides a detailed analysis of palaeoenvironmental proxies from a 6.29m peat core from forested peatlands in the Republic of Congo. It finds a period of low rates of net peat accumulation, referred to as the "Ghost Interval", and presents evidence this is driven by a drier climate. The study does a good job of placing the changes in regional context through time. It also compares to climates of other peatlands around the world and discusses potential future implications of the changes. Although this is the first core to be analyzed in detail from the Cuvette Centrale peatlands of the Congo, similar analyses have been conducted in other regions, including documentation of periods of hiatus in peat accumulation, changes in accumulation rates, and changes in vegetation community, linked to variation in climate over time in tropical peatlands in South America and Southeast Asia. Most of these detailed peat core analyses are published in journals relevant to the specialist audience, such as *Global Change Biology*; *Palaeogeography, Palaeoclimatology, Palaeoecology*; *Quaternary Science Reviews*; *Journal of Geophysical Research: Biogeosciences*, etc. Similarly, this manuscript is likely to primarily be of interest to a more specialist audience, and may benefit from a longer format to show more of the data and comparisons to supporting cores. I would therefore recommend publication in *Nature Climate Change*, *Nature Geoscience*, *Global Change Biology*, or another disciplinary journal.*

Thanks for the kind words that this study merits publication. As this is the first detailed palaeoenvironmental and palaeoclimate data from the world's largest tropical peatland, which we think is of interest to peatlands scientists, climate change scientists, tropical ecologists, paleoenvironmental scientists and Earth system modelers, we think it is suitable for a general science journal such as *Nature*.

Major comments/concerns: Emphasis on peat loss and C source is not supported - There is frequent reference to "net peat loss" and switch to a C source during the Ghost Interval. For example, in the abstract: "...triggered the switch from net peat accumulation to net peat loss" and in the conclusions "likely causing a shift from a carbon sink ..to a carbon source..." (269-270). This emphasis seems sensationalized. The Ghost Interval is defined as the period from ~7,300 to ~2,000 cal yr BP, and there is cumulative peat accumulation of ~50cm to ~>1m across all dated cores over this time period (Fig 1, Fig ED3). There might have been a more limited period of peat loss within this time window, but over the entire Ghost Interval, there is still net peat accumulation. Relatedly, over each shorter individual dated interval in the main core (Fig 1), there is still net peat accumulation. Slower accumulation rates of highly decomposed peat, but still net accumulation. There is certainly a possibility that the peatland was a C source for a shorter fraction of the Ghost Interval period, but the authors should not imply a massive C source over this period.

We define the Ghost Interval as a highly decomposed peat section (see our response below). In terms of interpretation of the Ghost Interval, we formally show, from the peat age-depth profile, a shallower profile at this, as shown in Fig. 1, and reported this in the main text. However, there is more than one interpretation of this period of low 'net peat accumulation', so we avoid saying this in the opening of the manuscript. Later, because we conclude that increased contemporaneous decomposition and secondary decomposition could explain the shallower age-depth profile and the highly decomposed peat of the Ghost Interval, we discuss two possible processes, contemporaneous decomposition (resulting in low net peat accumulation) and secondary decomposition and a large C release.

In addition to the data that supports contemporaneous decomposition, the key data that supports a role for secondary decomposition comes from the difference between the basal age of the decomposed material that forms the Ghost Interval (~7,500 cal. yr BP) and the age of the drying signal that likely triggered it (starting at ~5,000 cal. yr BP). This indicates that the drier conditions resulted in the loss of peat carbon that had accumulated before the drying event began, which is consistent with the expectation from secondary decomposition.

We now highlight this discrepancy in a revised Figure 3, adding a Fig. 3a to compare the Ghost Interval compared to the period of drying in Fig. 3b, and in the definition of the Ghost Interval timing, at line 240: "The basal age of the highly decomposed peat in the Ghost Interval at CEN-17.4 is ~7,500 cal. yr BP, which is older than the beginning of the climatic drying at ~5,000 cal. yr BP. However, the end of the decomposed Ghost Interval section and the end of the climatic drying are coincident at ~2,000 cal. yr BP. This pattern is consistent with secondary decomposition (scenario 3), likely resulting from drier conditions lowering the water table and exposing older peat layers, including peat deposited prior to aridification, to oxidation (i.e., decomposing peat older than 5,000 cal. yr BP); cf. 'Ghost Interval' and 'Drought' in Fig. 3a and b."

Ghost Interval inconsistently presented - The Ghost interval is inconsistently referenced in figures, emphasizing key transitions in each, but not in agreement. This makes comparisons across figures more confusing for the reader. The Ghost Interval is frequently defined as the period from ~7,300 to ~2,000 cal yr BP (line 111), as shown in Fig 2. However, in Fig 3, the Ghost Interval shading only includes the dry sub-interval of this period. In Fig 1 it is not consistent across cores. Also, inconsistent in ED Figures.

We apologize for not being clear enough about the definition of the Ghost Interval in the originally submitted manuscript, which caused confusion among the reviewers. The Ghost Interval represents a highly decomposed peat section. For the CEN-17.4 core, where we have the most information, we define the Ghost Interval as the depth interval showing a prominent break in the modelled age-depth profile, which also coincides with negative I-index values and a coeval increase in TOC values and C/N ratios. This dates the interval as ~7,500-2,000 cal. yr BP. For the secondary Ekolongouma cores (EKGKM7 and EKG03), the Ghost Interval is similarly defined by negative I-index values and a coeval increase in TOC values and C/N ratios, which we have now 14C dated the beginning and end of the Ghost Interval. For the new LOK5-5 and BDM1-7 cores, we define the Ghost Interval as a prominent break in the modelled age-depth profile. This is now implemented in the text and the figures harmonized, which has significantly improved the presentation and clarity of the manuscript.

Extrapolation via secondary core data- The analysis of the primary core is very thorough and well-presented, and serves as the basis for a theory of regional change. However, the applicability of the theory beyond the single core is a crucial point that needs to be better supported. The data from secondary cores may support the proposed theory, but there are also inconsistencies. Additionally, it is hard to assess the relationship between the trends in these cores, due to the limited data presented, and the way it is displayed. More details below -Fig 1c and Fig ED3 – Substantial peat accumulation in secondary cores during Ghost Interval - Why did the peat accumulate so much during the Ghost interval at the other sites if it was a period of enhanced decomposition or peat loss? It seems in some of these cores, the peat accumulated by as much as, or even over a meter, during the period from 7600 kyr BP to 2050kyrBP. For some of the cores, this is nearly half of the total peat accumulation at that site. Why would the peatland be expanding laterally or have higher rates of accumulation closer to the peat dome edge, during a dry period of peat loss in the interior? If the Ghost Interval is entirely driven by climate alone, then why would the peatland expand at all during this period? Could there also be some element of vegetation succession? Different hydrology across the dome? E.g. domed peatlands in Panama and SE Asia? Other explanation? Not clear if the secondary

cores provide strong evidence for the Ghost Interval. This would benefit from further exploration in the manuscript, and bringing data from the secondary cores into the main manuscript. Misleading labeling - Labelling of the Ghost Interval on Fig 1 is misleading, suggesting it is more consistent than it is. Think about ways to plot the relevant data from all cores on the same time scale in some figure, to show what is happening over time in all cores. (see attached pdf for figure annotation) Number of cores – 5 supporting cores are mentioned (line 95), but only limited data is analyzed/presented, and not the same cores each time. Why are some cores (EKG4km, 7km and 9km) presented in Fig 1 with radiocarbon data, while cores EKG02 and 03 are presented in the extended data with RockEval data? Do you have radiocarbon and RockEval data from the same cores to show that the transition in carbon quality occurs over the same time period in all the cores? If not, could you add a few ^{14}C dates to those cores?

We have now added two ~6 m cores to better support the CEN-17.4 core. The Ghost Interval is clear in these cores which are both located far away (>170 km) from CEN-17.4. This is now in Figure 1. All three cores are reported on the same scale in Figure 1.

We agree that we should have presented more consistent data across the cores within the Ekolongouma basin. The reason was that this manuscript has no dedicated funding, as we did not expect to find a Ghost Interval, so the secondary cores come from re-purposing data from two different PhD projects on different research questions, hence the different data. We have done new laboratory analyses to now present cores EKGKM7, and EKG03 with the same decomposition data (I-Index, total organic carbon, C/N ratio) as we have for the CEN-17.4 data, and include new radiocarbon dates to assist in defining the timing of the Ghost Interval, as suggested. This data is now in Extended Data Fig. 3, and shows the same data from all three cores on the same scale, as requested. Our new dataset indicates that the thickness of the Ghost Interval is consistent (i.e. of 40-50 cm) across ~20 km of the Ekolongouma interfluvial basin.

Other comments/questions:

---For an interesting exploration of how tree falls in tropical peatlands can also cause major changes in apparent rates of peat accumulation, see Dommain et al, 2015 (JGR Biogeosciences), especially Fig 11. doi:10.1002/2014JG002796. Very relevant to detailed interpretation of peat cores in tropical peatlands. Can likely be ruled out with combination of wide range of datasets presented here, but may be relevant.

Thank you for indicating this valuable reference, which we are now citing.

---Given variability in local conditions, trends need to be replicated across multiple cores – data presentation needs strengthening.

The Ghost Interval is now presented in cores from across the region, from Lokolama and Bondamba, in Fig. 1.

---Although separated in space, all cores are from a single peat dome. Research in the inland peatlands of Pastaza-Maranon foreland basin in Peru with some similarities to the shows different peatlands can have different trajectories and responses, due to stages of vegetation succession, river meander, etc. in different parts of the peat complex

We now include two cores from other peatland areas, Lokolama and Bondamba, both >170 km from the Ekolongouma peatland, showing the Ghost Interval indeed is a regional phenomenon.

We are alert to the similarities and differences with Peruvian peatlands, which are notably younger due to river meanders, which do not seem to be a feature of the central Congo peatlands, given the ages of the deep cores in Fig. 1 (some authors of our manuscript have worked in the field in both systems).

---This is likely a dynamic floodplain environment, which may have similarities to the peatland complex in the better studied Pastaza Maranon foreland basin. Any thoughts on the role of river meandering?

This is not a dynamic floodplain environment. This is why we chose this CEN-17.4 core to work on, from an interfluvial basin far from river influence, far from direct human influence, and is a rain-fed ombrotrophic-like interfluvial basin, ~40 km from the nearest river (detailed in Dargie et al. 2017, *Nature*).

---Possible hiatus in peat accumulation and changes in vegetation community are also explored in Kelly et al (2017) <http://dx.doi.org/10.1016/j.palaeo.2016.11.039>

We now cite this, thanks, at line 239.

---Extended Data Table 2 – nice summary. Would be valuable in the main text.

We are currently at the limit of Figures, Tables and text, so leave this in the Extended Data, but could change this on the editor's advice.

--- Fig 1 – Relationship to Ghost Interval in other cores is not very clear. In Fig 1, the Ghost Interval is labeled as only ~50cm of the peat core, while in Fig ED3, the Ghost interval from the supporting cores is labeled as covering much of the analyzed core, perhaps over 1m of the peat core. What accounts for these differences?

We have now amended Fig. 1 and Extended Data Fig. 3, with more sites and more ¹⁴C dates. The Ghost Interval varies from 39 cm at CEN-17.4, 40 cm at LOK5-5, and 69 cm at BDM1-7, which is very consistent for these three widely spaced and deep cores, particularly given the lower dating resolution of LOK5-5 and BDM1-7. The Ghost Interval is 48 cm at EKGKM7 and EKG03, which are towards the periphery of the Ekolongouma peatland, but are still 8 km and 6 km from the peatland margin respectively, and also have lower dating resolution than the CEN-17.4 core, again broadly consistent.

---Fig 2: 2000-1000yrBP – Why did peat accumulation restart/continue in such extremely dry conditions?

We now explain this more clearly in main text, line 248: “The gradual recovery of hydrophilic swamp forest taxa and renewed high rates of net peat accumulation (‘Recovery’ in Fig. 3) following the Ghost Interval contrasts with the high and stable δD_{precip} values lasting until ~900 cal. yr BP. This may indicate that the peatland ecosystem was more sensitive to the change in the precipitation regime, than to the absolute level of precipitation. Models³¹ and simulations^{32,33} show that net peat accumulation can recover, even at permanently lower levels of water inputs, via negative feedback mechanisms, such as changes in peat permeability in response to drying, that lead to shallower water tables^{18,34}. Alternatively, the recovery may have resulted from an increase in the amount of precipitation and a decrease in seasonality, as this would result in invariant δD_{precip} values (Fig. 4).”

---Fig ED3 – why did so much of the peat accumulate during the Ghost Interval at the site EKG03 and EKG02 if it was a drier period of enhanced decomposition? Why would the peatland be expanding to these areas of shallow peat during that period if peat was simultaneously being lost in the center of the peat dome? How do you reconcile the concurrent areal expansion and loss of peat depth?

We now have better sampling resolution and additional decomposition proxies for the Ekolongouma Basin secondary cores (EKGKM7 and EKG03) and they show a similarly thick Ghost Interval compared to core CEN-17.4 (see response above). Therefore, we do not need to reconcile the loss of peat in one location and accumulation in another, as all three sites are showing similar highly decomposed peat over a similar time period, now shown with new ¹⁴C dates on the EKGKM7 and EKG03 sites (shown in Extended Data Fig. 3).

---Were the climate reconstructions consistent across all cores? Show in SI?

The climate reconstruction was made only for core CEN-17.4, it is shown in Extended Data Fig 6.

---Ln 107: 4-6x slower accumulation is not net loss over the Ghost Interval. Avoid hyperbole.

This statement does not mention ‘net loss’. The full sentence is:

“In the depth interval 190 to 151 cm, dated from 7,520 to 2,150 cal. yr BP, the gradient of the modelled age-depth profile is five to eight times shallower than in the peat immediately below and above.”

---Ln 108-109: This statement and Ghost Interval labeling in Fig 1c is somewhat misleading.

We have updated Fig. 1 (see responses above). We have updated this statement to read: “We assess if this Ghost Interval is a widespread feature of the central Congo peatlands, by ¹⁴C dating two other ~6-m-long cores from (i) LOK5-5, from a basin close to the Congo River in the Democratic Republic of the Congo, 177 km from CEN-17.4, and (ii) BDM1-7, from a river-influenced valley-floor peatland (not an interfluvial basin) close to the Ruki River, a tributary of the Congo River in Democratic Republic of the Congo, 274 km from CEN-17.4 (Fig. 1, see Methods). Both cores LOK5-5 and BDM1-7 show similar basal ages to CEN-17.4, and similar overall age-depth profile patterns, including much shallower Mid- to Late Holocene age-depth profiles compared to the peat immediately above or below, despite fewer ¹⁴C dates (Fig. 1c,d). The Ghost Interval therefore appears to be a regional phenomenon.”

---Ln 115-128: Nice job outlining possible scenarios and evidence for & against them.

Thank you.

---Ln 138: Two other peat cores? 5 supporting cores. What about other 3? Not analyzed? Or different trend?

As discussed above, we have clarified this section to include the same analysis of the cores from the periphery of the Ekolongouma basin, including the Rock-Eval I-index, the C/N ratios and the TOC % for cores EKGKM7 and EKG03 (see Extended Data Fig. 3). Thus, in the revised manuscript there are three cores from the Ekolongouma basin, all with the same datasets analysed and presented on the same axis scales. They all show similar trends.

---Very interesting ideas and comprehensive dataset. However, not yet convincingly supported at the basin scale by other cores.

Thank you. We have dated two new 6-m-long cores (LOK5-5 and BDM1-7) and we show that the Ghost Interval appears to have occurred at the basin scale (see new Fig. 1).

---Ln 151-154: Interesting!

Thank you.

---Ln 170, 198, 225: What caused the sharp re-wetting ~2kyr BP from a climate perspective? Most discussion seems to focus on the drying trend, but the reversal is also of interest.

In our update of Fig. 3, we have more clearly labeled the Recovery, as this is of interest. The cause of this change is not well known and we can only speculate about its origin (e.g., ENSO variability). We preferred not to attribute a specific driver for it, to avoid speculation.

---Ln 199-203: Not enough explanation in the text here to follow what you did

We now provide a new figure presenting the workflow used (see Extended Data Fig. 6), and more clearly direct the reader to this and the methods where the detail is, at line 226: “We empirically compared the relative changes in peat δD_{n-C29} values at CEN-17.4 with δD_{precip} values computed using modern climate data (see Methods and Extended Data Fig. 6) to provide a climate space that includes all

climate solutions consistent with peat δD_{n-C29} values for both the pre-drought period, i.e., at 5,000 cal. yr BP (dashed lines in Fig. 4) and the most severe peat decomposition period, i.e., at 2,000 cal. yr BP (solid lines in Fig. 4). These reconstructions of past precipitation regimes suggest that the precipitation at ~2,000 cal. yr BP was at least 800 mm^{-1} lower than that at ~5,000 cal. yr BP, and possibility as high as $1,500 \text{ mm yr}^{-1}$ lower (Fig. 4).”

---Ln 211-212: Also S. America (e.g. Kelly et al, 2017 PPP)

Thank you. Reference added.

---Ln 215-217: Interesting point. Why would this be the case? Please explain and justify further.

Thank you. We updated the text adding: “Models³¹ and simulations^{32,33} show that net peat accumulation can recover, even at permanently lower levels of water inputs, via negative feedback mechanisms, such as changes in peat permeability in response to drying, that lead to shallower water tables^{18,34}. Alternatively, the recovery may have resulted from an increase in the amount of precipitation and a decrease in seasonality, as this would result in invariant δD_{precip} values (Fig. 4).”

---Ln 241: reduced area is a big claim. Why were peatlands accumulating substantially at other cores? Add more caveats?

We say ‘may have reduced the area’, based on the arrival of pre-aged carbon in the Congo Fan. We are reporting one line of evidence here rather than making a major claim. Additionally, the new cores at Lokolama and Bondamba show that other peatlands within the central Congo Basin complex show a similar Ghost Interval to the Ekolongouma peatland. We also now show more clearly, that EKGKM7, and EKG03 did not accumulate substantial peat in the peatland margin (see response above). For clarity, we now note that EKGKM7 and EKG03 are 8 km, and 6 km from the peatland margin respectively, at line 169.

---Ln 251: “may have been much larger in the past”. Or smaller? Likely varied through time, especially given wide variability in peat depth and basal ages.

We have now clarified the time period to make this sentence clearer and more accurate, and removed the word ‘much’, at line 286: “The surface area occupied by central Congo peatlands may thus have been larger than today’s $145,000 \text{ km}^2$ (Ref. ¹) prior to the period of drying beginning ~5,000 cal. yr BP.”

---Ln 257: Not conclusive. “may” have been lost. “Lost” might not be the right word if it never formed due to enhanced decomposition/slower accumulation rates. Loss is a possibility, or hiatus in accumulation, as you also present.

We have inserted the word ‘may’ in this sentence. We keep the word ‘lost’, as this is our interpretation, on balance, of the evidence. We present all the possibilities, appraise the evidence, and this is our conclusion.

---Ln 258: Would it be possible to date the other 2 supplemental cores? Could strengthen regional arguments.

We have now dated the EKGKM7 and EKG03 supplemental cores from the same peat complex of core CEN-17.4, in Extended Data Fig 3. We have also dated two new cores (LOK5-5 and BDM1-7) from other peat complexes, in Fig. 1. They do strengthen the regional arguments.

---Ln 266-267: Abrupt massive C loss is not realistic or likely. Be careful not to accidentally imply it in the abstract – may confuse non-specialist readers.

We did not use the words ‘abrupt’ or ‘massive’. We wrote: “However, the lack of a sharp increase in atmospheric CO₂ from high-resolution ice core records at ~2,000 cal. yr BP implies that carbon release was over a period of at least decades, if not hundreds of years⁴⁵”, which suggests that it was not abrupt.

---Ln 269-270: “most likely” – should this be “possibly”? Given net accumulation is still positive on average over every dated interval.

We report all the possibilities in the paragraphs prior to this, and this ‘most likely’ statement is our conclusion, based on our interpretation of the data.

---Ln 286-288. Relevant work has addressed this issue but is not cited. e.g. Cobb et al. (2017). PNAS. See Fig 10.

Thank you. We have now added this reference.

Referee #2 (Remarks to the Author):

Summary of key results: In the forested swamps in central Congo, there may have been a drier period at 7000 (or 5000?) to 2000 cal yr BP, when the peat was more decomposed. This may have implications for future peat accumulation in the region.

We have clarified the 7000/5000 confusion. The decomposition-affected peat is dated from ~7,500 to ~2,000 cal. yr BP, which we call the Ghost Interval (Fig. 1 b, Fig. 3a). The drier conditions, from δD_{n-C29} , a proxy of precipitation, began ~5,000 cal. yr BP and strengthened until ~2,000 cal. yr BP (Fig 3b-e). We now note this at line 240 in the manuscript, and in Fig. 3 more clearly, by adding a new Fig. 3a to show the Ghost Interval.

Originality and significance: In principle, this is an original and valuable study, as it provides new evidence about past peat decomposition and accumulation dynamics and past climate changes in the forested swamps in central Congo basin.

Thank you.

Data and methodology: The biggest problem is that the study is based practically on one peat core, CEN-17.4. As the size of the forested peatland area is about 145 000 km², it should be obvious that no conclusions about the general trends in decomposition and peat accumulation rates can be made based on the results of a single core. The five secondary cores analyzed (earlier) with lower resolution do not really have any role in the paper (for example Figs. 2 and 3 show data only from CEN-17.4). We can compare this with the peat accumulation studies in the boreal and arctic peatlands, where regularly more than one core are analyzed from one bog or swamp, let alone from an area of 145 000 km².

We agree that more data is better, but we must start somewhere, as this is the first ever detailed palaeoenvironmental study of the world’s largest tropical peatland.

We have added two new ~6-m cores, to match the ~6-m CEN-17.4, LOK5-5 and BDM1-7, collected 177 and 274 km to the southeast and east of core CEN-17.4 respectively, and both 14C dated, now plotted in Fig. 1. These are all the similarly deep cores from the region that we have. They show the same Ghost interval. Thus, we have three cores, from three areas far from one-another showing strikingly similar patterns, which reinforces our conclusions.

The second main comment is about the classification of the pollen data. The authors divided the pollen taxa of the CEN-17.4 core into six vegetation types. But it is not explained how this was done. Pollen taxa are not the same as the plant taxa, and in most cases subjective decisions and major generalizations are needed for such classification of pollen taxa. It would thus be necessary to include a table, which would show how different pollen taxa were classified in this case.

We also agree that the classification used may be subjective. To avoid this problem, instead of grouping taxa into 'swamp forest', we plot the two commonest taxa, in terms of percentage pollen following swamp forest establishment, *Pycnanthus* and *Panandus*, which both show sharp declines at ~2,000 cal. yr BP (Fig. 3). In addition, we also plot two common light demanding and pioneer taxa that also show changes coincident with the Ghost Interval: *Lophira* and *Alchornea* in the Extended Data Fig. 4.

Appropriate use of statistics: OK

Conclusions: Based on so scarce data, the conclusions are not feasible. While the topic is interesting, much more data based on peat cores needs to be generated before it is possible to make any general conclusions about the regional trends in peat decomposition and accumulation in the whole region.

Our two new additional ~6-m cores from across the region allow us to be much more confident about our conclusions, particularly as these are in very different (interfluvial and river-valley) settings. But, of course, this is the first such study of the region, so conclusions will necessarily be subject to potential revision as more data come available. Regarding the central Congo peatlands we are at a stage similar to a century ago in temperate peatlands, the first studies providing an initial description and understanding of the system, which will necessarily be more data-sparse than regions with a long history of study.

Suggested improvements.: It would be good to be clearer with the timing of the period with high peat composition and possible dry period (termed "Ghost Interval" in the paper). It is stated in the abstract that this period dates to 7300-2000 cal yr BP and the same is indicated in Fig. 2. But in Fig. 3, which shows the pollen and δD data from the same core, the period is indicated at 5000-2000 cal yr BP. To some extent the authors state the factors which may explain the difference between the peat decomposition data and the pollen/ δD data, but in any case the difference between Fig. 3 and the rest of the paper can be confusing for many potential readers.

Apologies, this is now fixed. We are now clearer in the abstract as to why this difference occurs and what it means: "The peat appears to have lost much of its original carbon mass during a phase of severe decomposition, including the loss of carbon accumulated during prior wetter conditions, as deeper water tables probably triggered a switch from net peat accumulation to net peat loss."

Also, we have modified Fig. 3 where the Ghost Interval is indicated from ~7,500 to ~2,000 cal. yr BP (Fig. 3a) to clearly distinguish it from the panels showing the drier conditions from ~5,000 cal. yr BP to ~2,000 cal. yr BP (Fig 3b-e). We also address this in the text, and line 240: "The basal age of the highly decomposed peat in the Ghost Interval at CEN-17.4 is ~7,500 cal. yr BP, which is older than the beginning of the climatic drying at ~5,000 cal. yr BP. However, the end of the decomposed Ghost Interval section and the end of the climatic drying are coincident at ~2,000 cal. yr BP. This pattern is consistent with secondary decomposition (scenario 3), likely resulting from drier conditions lowering the water table and exposing older peat layers, including peat deposited prior to aridification, to oxidation (i.e., decomposing peat older than 5,000 cal. yr BP); cf. 'Ghost Interval' and 'Drought' in Fig. 3a and b."

Thanks, these suggestions have really improved the paper.

Referee #3 (Remarks to the Author):

This manuscript presents the result of the first investigation into the Congo Basin peatlands Holocene history and the relationship between the peatland carbon accumulating function and hydroclimate. This is important because: 1) The Congo Basin represents the largest intact tropical peatland complex 2) There is little data on peatland development and resilience in the tropics 3) The fate of this peatland complex in the future very much matters in terms of carbon-cycle climate feedbacks.

The manuscript provides an original conclusion that merits publication. The manuscript and abstract are elegantly written, the introduction provides good context to the work.

The methodological approach is thorough and the manuscript presents an impressive amount of multi-proxy paleodata to pin down how climate is likely to have changed over time to explain variations in the peatland's carbon exchange with the atmosphere. I appreciate how the authors have moved away from attempting to quantify the change in the peatland's carbon sink - something that would need a modelling approach, meriting a separate publication - but have instead focussed the efforts in understanding through a multi-proxy approach how a dry period lead to contemporaneous and secondary decomposition of the peat carbon some 2000 years ago, leading to the eventual record of what they have called the Ghost Period. I particularly appreciated the description of the modern day climate space that shows how these peatlands are on the edge of the precipitation amount and seasonality distribution.

All of this impressive work is definitely appropriate for the aims of the investigation. The reporting of the data is also sufficiently detailed. Uncertainty in the data has been shown visually in figures or reported in the text.

In my view, these results will be of interest, not just to scientists in the discipline, but more widely to tropical ecologists (including those studying terra firme forests), climate scientists and soil scientists. The changes in climate would affect not just peatlands, but all other ecosystems in the Congo Basin.

The conclusions and data interpretation are robust, valid and reliable. The manuscript reference previous literature appropriately.

The quality of presentation of the data is very good.

We thank you for all your positive comments.

Here are some small suggestions for improvement of the figures:

Figure 1, panel B - the core labels are not legible, need to be changed to white (no border)? or another colour that allows reading.

We have made these changes to improve clarity.

Figure 3, Swamp forest data - visibility of the actual data points (at least in my computer) is poor - I suggest you either colour the 68% envelope in a lighter green (as you have done for the arboreal data) or make the data points a lighter colour (not black).

We modified Fig. 3 accordingly.

It would be good to have the data and code available publicly - it seems this will be done in time, but it would be good to make the links available when the manuscript is published.

Data (<https://doi.pangaea.de/10.1594/PANGAEA.938019>) and codes (<https://doi.org/10.23708/FO2HGM>) will be made available publicly once accepted for publication.

Here is the confidential and anonymous access link to view the codes before publication:
<https://dataverse.ird.fr/privateurl.xhtml?token=604113fc-2e65-468f-a438-667c2166a8ee>

Referee #4 (Remarks to the Author):

In this paper, Garcin, Schefuß, Dargie et al explore the vulnerability of a major wetland complex in the central Congo Basin to past changes in hydroclimate. To achieve this, they employ a multi-proxy approach (bulk geochemistry, organic geochemistry, palynology). The team find low accumulation rates between ca. 7000 to 2000 years ago. This interval (termed the "Ghost Interval") is present in multiple cores and argued to result from enhanced peat decomposition (...and perhaps also post-depositional erosion). As this wetland holds ~30% of all tropical peat carbon (wow!), this has the potential to act as an important CO₂ source. Leaf wax hydrogen isotope ($\delta^{2}H$) values increase by ca. 30 per mil shift during this interval, suggesting that decomposition is associated with drier conditions. This implies that wetlands may release CO₂ under drier conditions, both in the past and future.

I really enjoyed reading this manuscript – it is well written and flowed logically. The figures are clear and

concise. The findings have the potential to be genuinely important. The topic is also timely and has direct relevance to the future.

We thank you for these constructive comments.

However, I did have quite a few questions. These are outlined below - I hope they help!

1) Estimating paleo-precipitation

To generate quantitative precipitation estimates, the authors must determine the relationship between δ^2H_{wax} and δ^2H_{precip} . Here, the authors assume a positive linear relationship (i.e., slope = 1), between δ^2H_{wax} and mean annual δ^2H_{precip} . However, various factors can affect the apparent fractionation between δ^2H_{wax} and δ^2H_{precip} (...and therefore, the resulting precipitation estimates). These factors include: i) soil evaporation, ii) leaf-water transpiration iii) wax biosynthesis.

I would argue that evaporative 2H -enrichment of soil water is relatively limited (especially in drylands). However, leaf water 2H -enrichment is far more variable and can depend on the species and the climatological setting (e.g., Daniels et al 2017, Feakins et al 2016a, Feakins & Sessions 2010, Kahmen et al 2013a). The apparent fractionation between δ^2H_{wax} and δ^2H_{precip} also varies widely (up to ~70 ‰) between plant life forms (i.e., trees, shrubs, forbs, graminoids) and physiological groups (i.e., C3, C4, CAM) (Sachse et al 2012). You observe a large decline in forest taxa from 50% to 20% during the Ghost interval. Can this explain changes in δ^2H ? One way to explore this is to use data from the same samples to calculate plant-specific fractionation factors (e.g., Feakins 2013; P3; Inglis et al., 2020; P3). This is achievable - the authors have already generated pollen assemblage data.

We did observe a decline in swamp forest taxa during the Ghost Interval, but this cannot explain the δD_{n-C29} change, because there was no change in the abundance of forest taxa overall, just a change in composition to less swamp forest associated taxa and more terra firme associated taxa. There was neither change in plant life forms, nor in C3, C4, CAM physiological groups, immediately before, during or post the Ghost Interval. Over the whole of this period there was a closed-canopy tropical forest at CEN-17.4. We now note in the Methods at line 749: “Different vegetation types (e.g., C₃ vs. C₄) can cause offsets in δD values of plant wax due to different hydrogen isotope fractionation factors²³. However, as the measured n -alkane $\delta^{13}C$ values are stable during the Holocene (Extended Data Fig. 2c), no correction for vegetation type changes was applied.”

We translated peat δD_{n-C29} values into mean annual δD_{precip} values using a global surface soil calibration relationship (Ladd et al. 2021), which include peat samples, and which has a slope of 0.98 ($R^2 = 0.67, p < 0.001, n = 474$), as detailed in the Methods at line 791.

Estimating the apparent fractionation between δ^2H_{wax} and δ^2H_{precip} must also account for any seasonal bias relative to annual rainfall – seasonal studies of modern plant ecohydrology reveal the seasonality of rainfall and plant uptake (Griepentrog et al 2019). As a “...reduction in total annual precipitation may have been accompanied by an increase in seasonality” (L207-208), this should also be considered by the authors.

We have not included this point because rainwater isotopic compositions from GNIP stations bordering the central Congo basin show a clear relationship between precipitation amount and δD_{precip} and no important seasonally different contributions from different moisture sources. We added in the Methods at line 799: “The Congo Basin, which has a concentric structure with a large central depression (the “Cuvette Centrale”) surrounded by topographic highs, is one of the most convective regions on Earth⁷⁷. Local recycling processes are very important and more than 80% of the total moisture contribution to precipitation over the basin originates from land sources, with ~60% of the total originating from the Congo Basin itself, and with relatively stable contributions throughout the climatological year⁷⁸.”

The Griepentrog et al. paper, though highly valuable to understand fractionation processes that drive δD_{wax} composition, is from Lake Challa in Kenya/Tanzania where air mass trajectories (NE versus SE monsoon) strongly control seasonal patterns of rainwater isotopic compositions, and where it is expected that plant growing at different seasons will record distinct δD_{precip} composition. This is a different system compared to the central Congo basin, which is wetter and much less seasonal.

2) Decoupling between hydrology and carbon cycling

The authors argue that drier conditions during the Ghost interval help to promote OC decomposition. This argument is based upon a shift towards high $\delta^{13}C_{org}$ values (-140 permil). However, there are intervals with similarly high values (e.g., 10-12 thousand years ago), but with NO evidence for enhanced OC decomposition. This is puzzling and suggest hydroclimate-carbon cycling decoupling in the early Holocene (...and hydroclimate-carbon cycling coupling in middle-to-late Holocene). This seems odd – why would this wetland complex only be susceptible to drying in the mid-to-late Holocene? (see also comment 4...). If this is true, what implications does this have for the future?

At 10-12 kyr ago the climate was transitioning from a cooler and drier climate to a warmer and wetter climate. Thus, while the $\delta^{13}C_{org}$ values 10-12 kyr ago are high, they are decreasing, as the climate gets wetter. Thus we see slow peat accumulation in the Pleistocene, then as we enter the Holocene we see greater peat accumulation as wetting occurs (Fig. 1b). The lack of OC decomposition related to the wetting of a drier climate is likely because the drier and cooler climate was already peat forming, but dominated by a different community of C4 grasses and sedges at that time (Fig. 2f).

We have not included this explanation due to limited space and it is not a central part of the analysis, but we could if the reviewer and editor require it.

3) Non-hydrological controls

The authors argue that hydrology is the key control on OC oxidation – however, the authors should also acknowledge that temperature and respiration rates are closely coupled (e.g., <https://www.nature.com/articles/nature11205>) and may also play a role in our future warmer world.

We agree, but we were trying to avoid too much speculation in the manuscript. We have added at line 324, and added recent references: “Rising air temperatures may amplify this feedback by either decreasing forest productivity and therefore litter inputs⁴⁸ and/or increasing microbially-mediated soil organic matter decomposition rates⁴⁹.”

4) Complex controls on peat decomposition

Various studies have already shown a link between hydroclimate and the carbon cycle in wetlands (especially in China). For example, Huang et al. (2018; Nature Comms.) use biomarkers to show that successive drying events have a big impact on the susceptibility of peat carbon stores to climate change. Swindles et al. (2018 Global Change Biology) also show that “droughts may lead to reduced C accumulation or even net loss of peat” in ombrotrophic domes (in Peru). However, peats are never simple and Swindles et al., 2018 show that “..future droughts may lead to phases of rapid carbon ACCUMULATION in some inundated tropical peat swamps”. In other words, drying does not necessarily equal decomposition. Can the authors comment a bit more on the wetland architecture in their study site – are we looking at a raised ombrotrophic bog (i.e., drying = OC loss)? Or a flooded wetland (i.e., drying = OC accumulation)? A bit of both? And how might it evolve in the future?

On wetland architecture, we have made clearer the introduction that CEN-17.4 site is a raised ombrotrophic-like raised bog, at line 97: “We provide a detailed analysis of a 6.29-m-long core from the centre of a ~40-km-wide interfluvial peat-filled basin⁸ which is a raised ombrotrophic-like peatland covered by swamp forest^{1,8}, informally named Ekolongouma, in the Likouala Department, Republic of the Congo, containing the deepest peat yet observed in the region¹.”

Also, when we first mention the new cores at Lokolama and Bondamba, we give the same information (note one is a river-influenced site): “We assess if this Ghost Interval is a widespread feature of the central Congo peatlands, by ¹⁴C dating two other ~6-m-long cores from (i) LOK-5.5, from a basin close to the Congo River in the Democratic Republic of Congo, 177 km from CEN-17.4, and (ii) BDM-1.7, from a river-influenced valley-floor peatland (not an interfluvial basin) close to the Ruki River, a tributary of the Congo River in Democratic Republic of Congo, 274 km from CEN-17.4 (Fig. 1, see Methods).”

We now report three widely spaced 6 m cores, which show similar age-depth profiles, including a similar Ghost Interval, which suggests that the Mid-to-Late Holocene drought did drive decomposition in these three different settings (updated Fig. 1).

On the controls on decomposition, we agree that drought-peat accumulation/decomposition is

complex, and hoped to convey some of that, in that the evidence suggests that drier conditions led to peat loss, including the loss of some peat older than the drier conditions that drove the loss, and then there was a recovery phase, in the new drier climate. We hope our revised Fig. 3 now conveys this clearly.

5) More evidence needed for enhanced decomposition

The authors argue that there is enhanced decomposition in the Ghost Interval (scenarios 2/3) – this is based on rock-eval pyrolysis, an increase in C/N ratios, a negative carbon isotope excursion in bulk OM, and higher TOC.

I have three questions here:

5.1. You associate higher TOC values with enhanced decomposition – however, I would expect to see LOWER TOC under enhanced decomposition. During the PETM we see very TOC values in continental sections, interpreted to represent enhanced soil OC oxidation (e.g. Denis et al., 2021; Baczynski et al., 2016;2019; Cotton et al., 2015). This also evident in biomarker distributions (e.g., changes in PAH distributions).

This statement is true for sediments that include a significant fraction of mineral matter and where the decay of organic matter would increase the relative contribution of mineral matter. However, the studied peat is made almost entirely of organic matter (~97%): peat decomposition over time results in a continuous increase in C content due to preferential mineralisation of the more labile organic fractions, notably the H-bond rich compounds (hydrocarbon compounds), leading to a relative enrichment in more refractory forms of carbon, such as C-bond rich compounds. We added in the Methods section at line 637: “Higher TOC_{EA} values in cores CEN-17.4, EKGKM7 and EKG03 are associated with enhanced decomposition because the peat from this study area is made almost entirely of organic matter (~97%, Ref. ⁴⁸), and peat decomposition over time results in a continuous increase in C content due to preferential mineralisation of the more labile organic fractions, notably the H-bond rich compounds (hydrocarbon compounds), leading to a relative enrichment in more refractory forms of carbon, such as C-bond rich compounds (Fig. 2 and Extended Data Figs. 2 and 3).”

5.2. You associate higher C/N ratios with enhanced decomposition – however, shouldn't microbial consumption of C- and H-rich organic substances result in a decreased abundance of carbon relative to nitrogen (e.g., Hornibrook et al., 2000 *Geochemical Journal*). i.e, lower C/N ratios.

A decrease of C/N ratios with ongoing decomposition would be observed if microbes fix the released nitrogen and bound it to the mineral matrix in sediments. Here, we are dealing with peat without a mineral matrix and the released nitrogen is lost from the system leading to an increase of C/N upon decomposition, i.e. preservation of condensed carbon. In order to explain this trend, we write the following text in the Methods section at line 637: “Higher C/N ratios in cores CEN-17.4, EKGKM7 and EKG03 are also associated with enhanced decomposition since there are no obvious changes in the source of organic matter and/or vegetation cover in the studied peat cores. The observed topmost increase in the C/N ratios with depth is consistent with a preferential loss of nitrogen in the surface peat due to aerobic respiration of organic matter (Fig. 2 and Extended Data Figs. 2 and 3). The C/N ratios stabilise at ~40-50 and do not show any further downward decrease as typically observed in boreal *Sphagnum* peat where the loss of more labile organic C is related to anaerobic decomposition of organic matter in the deep waterlogged peat (Kuhry and Witt 1996). The organic matter composition of this tropical peat is different from that of boreal *Sphagnum* peat, being dominated by roots and woody material from swamp forest tree species. Anaerobic decomposition of this type of peat is typically very slow under constant high water tables, which is attributed to the recalcitrance of the organic matter (Chimner and Ewel 2005)”.

5.3. Changes in C/N could also be driven by changes in vegetation too (e.g., Hornibrook et al., 2000). Could this be a factor?

Yes, we agree with this statement and added in the main text the following: “This may imply a preferential loss of nitrogen through aerobic decomposition, probably triggered by a lowering of the water

table in the past, although other processes including vegetation changes (Cools et al. 2014) might also influence C/N ratios.”

6) Quantifying CO₂ release

Assuming that this wetland complex is subject to widespread OC oxidation during the Ghost Interval, it has the potential to switch from a CO₂ sink to a CO₂ source. However, the authors do not provide any estimates for the potential CO₂ release. How much carbon could be released during the Ghost Interval? Without quantitative (or semi-quantitative) constraints, it is hard to assess if this is an important CO₂ source.

We have not made an assessment of CO₂ release, as we are cautious about speculating using a very simple calculation, which is all that is open to us at present. More data will be available in the future (such as a map of peat depth estimates), and combined with model simulations of peatland development, these will allow a first assessment. If the editor agrees that this is critical to the manuscript we could add a very simple back-of-envelope calculation.

Minor comments:

L115-116: to refute this hypothesis you could calculate long-chain n-alkane mass accumulation rates.

We refrain using accumulation rates because of secondary decomposition processes. Instead, we present the pollen concentration data, which show a marked increase across the Ghost Interval that is inconsistent with reduced litter inputs: “The increased pollen concentration (Extended Data Fig. 4f) across the Ghost Interval is consistent with contemporaneous and secondary decomposition (scenarios 2 and 3)”.

L221-222: and in China (Huang et al., 2018 Nature Comms) and Peru (Swindles et al., 2018; Global Change Biology).

We added the Peru reference but not the China reference as the latter study site (Dajiuhu peatland) is not a tropical lowland peat swamp forest.

L238-246: it would be nice to actually see this data in a figure to support your argument - the data from ref. 3 appears quite convincing.

The data in Ref. 3 is already published, and we do not think there is space to re-publish this, as we are at the limit of our Extended Data display items. Additionally, in fairness, we ought to then also re-publish data from Ref. 2, as we have no argument to choose one above the other for comparison with our data. Since re-publishing these does not help the reader that much, we prefer to leave it as our written statement.

Reviewer Reports on the First Revision:

Referees' comments:

Referee #2 (Remarks to the Author):

This is the second time I am reviewing this manuscript.

As compared to the first version, the major change in the current version is that the peat accumulation data from two other peat cores have been added (LOK5-5 and BDMI-7). They show the same main accumulation trend with a period of slow accumulation at 7.5-2 ka, which lends support for the argument of the "Ghost period". This arguably makes the case stronger.

I still partially disagree with the interpretation of the palaeoclimate data in Fig. 3. The humidity reconstruction is based on δD_n-C29 values. When one looks at these values in Fig. 3b (not Fig. 3a as stated on line 218), we can see that the driest peak dates to 9.0-8.5 ka. This period should therefore also be marked as "drought" in the figure and should be taken into account when the impacts of drought on peat accumulation are discussed.

Instead of the summary pollen curves of the first version, pollen curves of only two taxa, *Pandanus* and *Pycnanthus*, are shown in Fig. 3. A reference to their ecological requirements would make the interpretation stronger. I note that *Pandanus* has a peak at 9 ka, coeval with the drought peak

Referee #3 (Remarks to the Author):

My initial review was overall positive:

"This manuscript presents the result of the first investigation into the Congo Basin peatlands Holocene history and the relationship between the peatland carbon accumulating function and hydroclimate. This is important because: 1) The Congo Basin represents the largest intact tropical peatland complex 2) There is little data on peatland development and resilience in the tropics 3) The fate of this peatland complex in the future very much matters in terms of carbon-cycle climate feedbacks.

The manuscript provides an original conclusion that merits publication. The manuscript and abstract are elegantly written, the introduction provides good context to the work.

The methodological approach is thorough and the manuscript presents an impressive amount of multi-proxy paleodata to pin down how climate is likely to have changed over time to explain variations in the peatland's carbon exchange with the atmosphere. I appreciate how the authors have moved away from attempting to quantify the change in the peatland's carbon sink - something that would need a modelling approach, meriting a separate publication - but have instead focussed the efforts in understanding through a multi-proxy approach how a dry period lead to contemporaneous and secondary decomposition of the peat carbon some 2000 years ago, leading to the eventual record of what they have called the Ghost Period. I particularly appreciated the description of the modern day climate space that shows how these peatlands are on the edge of the

precipitation amount and seasonality distribution.

All of this impressive work is definitely appropriate for the aims of the investigation. The reporting of the data is also sufficiently detailed. Uncertainty in the data has been shown visually in figures or reported in the text.

In my view, these results will be of interest, not just to scientists in the discipline, but more widely to tropical ecologists (including those studying terra firme forests), climate scientists and soil scientists. The changes in climate would affect not just peatlands, but all other ecosystems in the Congo Basin. The conclusions and data interpretation are robust, valid and reliable. The manuscript reference previous literature appropriately.

The quality of presentation of the data is very good."

I made some comments on quality of some of the graphics (which have been fixed). I am also satisfied that the data will be made publicly available.

The authors have now addressed most of the reviewers comments. Where they do not fully address these, they have explained why not and provided acceptable reasons. They have importantly added additional data (data from two new cores from the area) that support the Ghost Interval highlighted by the work that was one of the main problems with the previous version of the manuscript. They have additionally shown that using a different calibration curve does not change the conclusions of the study. The issues with the complications of reconstruction of paleo-precipitation have been addressed. As I already wrote in my first review. This work merits publication in Nature, the work is novel, it is the first of its kind for the region - the main tropical peatland in Africa and importantly, a vast intact tropical peatland region that has never been studied before. The edits and additions to the manuscript have improved its quality. I agree with the authors on the conclusions (that the Ghost Interval likely represents peat loss) - and I am satisfied that the wording represents well the uncertainty in the results and does not over-claim conclusions.

Referee #4 (Remarks to the Author):

Review of Garcin et al.,

A major concern was the reliance of a single peat core. The authors have now included two additional peat cores from the wetland complex, both of which exhibit similar (although not identical) bulk geochemical features during the middle Holocene. This is a welcome addition. However, visual inspection of the two new cores suggest that the Ghost Interval terminates at different times in the peat complex (e.g. ~ 4 ka in LOK5-5). In addition, the new data does not help to refine the authors data interpretation and only one of the four potential hypothesis is excluded (i.e., reduced litter input; L283-286). As such, the papers remains speculative.

I also have various questions about the data interpretation (see below):

1) Leaf wax hydrogen isotope - vegetation effects

The authors use leaf wax δ^2H values to reconstruct precipitation δ^2H during the "ghost interval". However, the study shows a change in vegetation type within the Ghost Interval (e.g., decline in

swamp forest taxa accompanied by an increase in light demanding and pioneer taxa; L184-189). As the apparent fractionation between $\delta^2\text{H}_{\text{wax}}$ and $\delta^2\text{H}_{\text{precip}}$ varies widely (up to $\sim 70\text{‰}$) between plant life forms (i.e., trees, shrubs, forbs, graminoids) and physiological groups (i.e., C3, C4, CAM), such changes could bias your results.

The authors acknowledge that "...different vegetation types (e.g., C3 vs. C4) can cause offsets in δD values of plant wax due to different hydrogen isotope fractionation factors". However, they argue that changes in C3/C4 plants are not important at this site. I agree - there is no obvious change in C3/C4 plants (this would be reflected in the leaf wax $\delta^{13}\text{C}$ values...). However, you do find large changes in the composition of C3 plants. Changes in the plant community can modify leaf wax hydrogen isotope values significantly, especially in peats. For example, Balascio et al. (2018) analysed leaf waxes in a Norwegian bog and found that that $\delta^2\text{H}$ values varied from -120 to -220‰ across a range of plant species.

2) Leaf wax hydrogen isotopes - inferring $\delta^2\text{H}$ precip

The authors used "a global surface soil calibration relationship (Ladd et al. 2021), which include peat samples" to infer $\delta^2\text{H}$ precip values. However, I am unclear how many peat samples are actually included in this calibration. I had a quick look into two major compilations included in Ladd et al., 2011 (McFarlin et al., 2019 and Liu and Ann, 2019; these should be cited). The McFarlin et al., 2019 paper appears to be lakes only, and Liu and Ann 2019 doesn't appear to include any peat samples (e.g., Huang 2018, 2016, Balascio 2018 ad Zaho et al., 2018, Seki 2009). Can the authors please clarify how many peat samples are used in this compilation?

This is crucial because the source of the pore water used for leaf wax biosynthesis in peatlands is not well constrained and is likely variable across the peatland. For example, , Huang et al. (2016) demonstrated that $\delta^2\text{H}$ values of long-chain n-alkanes within a single peatland is highly variable. This uncertainty needs to be considered.

3) C/N ratios

The authors argue that higher C/N ratios are an indicator for enhanced decomposition. However, evidence suggests we would expect the opposite - microbial consumption of C- and H-rich organic substances is known to result in a decreased abundance of carbon relative to nitrogen (i.e., lower C/N ratios; e.g. Malmer and Holm, 1984). In the response, the authors disagree and suggest this argument is not valid in peats. However, the evidence I cited was based on peats. In fact, the C/N ratio has been commonly used as indicator for the degree of peat decomposition, based on the relatively higher loss of C compared to N during decomposition and thus indicating peat mass loss (e.g. Hornibrook et al., 2000; Kuhry and Vitt, 1996; Broder et al., 2012). Thus I struggle to understand how higher C/N ratios = enhanced decomposition.

4) The ghost interval ends at different times...

Leaf wax $\delta^2\text{H}$ values suggest drying (5.0-2.0 kyr) occurs 2,000 years after the Ghost Interval begins. The authors argue that this pattern is consistent with secondary decomposition (scenario 3) around

2,000 years ago (L133-135 in revised MS). However, visual inspection of the two new cores suggest that the Ghost Interval terminates closer to 4 ka (LOK5-5) and 2.5 ka (BDM1-7). This suggests that secondary decomposition scenario is occurring at different times in the peat complex - it may also be inconsistent with the secondary decomposition scenario argued above.

5) Leaf wax hydrogen isotopes and carbon cycling decoupling

The authors argue that drier conditions during the Ghost interval help to promote OC decomposition. This argument is based upon a shift towards high δ^{2H} values (-140 permil). However, there are intervals with similarly high values (e.g., 10-12 thousand years ago), but with NO evidence for enhanced OC decomposition. The authors have attempted to address this in the rebuttal, but I don't quite understand their explanation. Can the authors please expand and include some relevant citations to support their argument?

Minor comments:

The paper states that "Our results indicate a newly identified positive feedback in the global carbon cycle, whereby climate-induced drought leads to the release of further carbon from peat to the atmosphere". However, I don't think this is a new idea (e.g., Limpens, J. et al. *Biogeosciences* (2008)).

Author Rebuttals to First Revision:

NB: *The Reviewer's comments are printed in italic font (black). For reading ease our responses to the comments are printed in bold font (black). Passages in the manuscript (corrected or original) are printed in normal font (blue).*

We thank the reviewers for their second round of comments, which helped to clarify some points of the manuscript. At the request of the editor we have reduced the length of the abstract to 228 words, and have reduced the number of references in the main text to 49.

Referee #2 (Remarks to the Author):

This is the second time I am reviewing this manuscript.

As compared to the first version, the major change in the current version is that the peat accumulation data from two other peat cores have been added (LOK5-5 and BDMI-7). They show the same main accumulation trend with a period of slow accumulation at 7.5-2 ka, which lends support for the argument of the "Ghost period". This arguably makes the case stronger.

I still partially disagree with the interpretation of the palaeoclimate data in Fig. 3. The humidity reconstruction is based on δD_{n-C29} values. When one looks at these values in Fig. 3b (not Fig. 3a as stated on line 218), we can see that the driest peak dates to 9.0-8.5 ka. This period should therefore also be marked as "drought" in the figure and should be taken into account when the impacts of drought on peat accumulation are discussed.

Instead of the summary pollen curves of the first version, pollen curves of only two taxa, Pandanus and Pycnanthus, are shown in Fig. 3. A reference to their ecological requirements would make the interpretation stronger. I note that Pandanus has a peak at 9 ka, coeval with the drought peak

Thank you. Referee #2 is right, there is a marked decrease in δD_{n-C29} values at 9.0-8.5 ka, which is coeval with a peak in *Pandanus* pollen and which supports our paleo-hydrological reconstruction. However, this decrease in δD_{n-C29} values corresponds to a peak of wetness and not to a drying phase as mentioned by Referee #2. We apologize for not being clear enough about the definition of the wet/dry periods in Fig. 3 of the revised manuscript, which caused Referee #2's confusion. We have now amended Fig. 3 (see below) to better show the drying conditions that resulted in the Ghost Interval.

Furthermore have also clarified the interpretation of the δD_{n-C29} values in the main text (lines 208 to 217): “Over the last 12,000 years, δD_{n-C29} values (Fig. 3b) range from -170 to -137‰. From ~12,000 to ~9,000 cal. yr BP, decreasing δD_{n-C29} values indicate a wetting trend, followed by generally wet conditions until ~5,000 cal. yr BP. From ~5,000 to ~2,000 cal. yr BP, increasingly D-enriched δD_{n-C29} values indicate a gradual drying. The overall 29‰ increase in δD_{n-C29} values observed from the midpoint of the Ghost Interval (at ~5,000 cal. yr BP) towards its top (at ~2,000 cal. yr BP) reflects a drying, which strengthened through time and coincides with the drying trend detected in an offshore marine archive²². This drying trend has been explained by the increasing meridional South Atlantic sea-surface temperature gradient from the Mid- to Late Holocene causing intensified trade-winds which reduced moisture transport from the Atlantic Ocean onto central Africa²².”

Regarding the ecological requirements of *Pandanus* and *Pycnanthus*, we have now added the following text with supporting references in the Methods’ section: “*Pandanus* and *Pycnanthus* pollen were selected to describe indicative changes in swamp forest vegetation. Both taxa are well known swamp forest taxa, occurring across central Africa. More specifically *Pandanus candelabrum*⁶⁷, which we find at CEN-17.4 site today, is a relatively heliophilic species often found on hydromorphic soils and can tolerate standing water up to ~1.2 m deep⁶⁸. It is seen in the wider peat swamp forests of the Ekolongouma region (G.C.D. and S.L.L. pers. obs.) and has been documented in swamp forests around Lake Télé, Lake Djéké, Lake Déké, Lake Manmagoye, in the flooded forests and on the banks of the watercourses in the Cuvette Centrale⁶⁹. Two species of *Pycnanthus* are documented in the Cuvette Centrale, *Pycnanthus angolensis* and *Pycnanthus marchalianus* Ghesq. They are both relatively heliophilic species and are often found on hydromorphic or clay-sandy soils. *Pycnanthus angolensis* is common in swamp forests as well as riverine and valley forests, open woodland and secondary bushland⁷⁰. *Pycnanthus marchalianus* Ghesq. is a more obligate swamp specialist, documented in the inundated forests of the Cuvette Centrale on marshy soils⁶⁹.”

Referee #3 (Remarks to the Author):

My initial review was overall positive:

"This manuscript presents the result of the first investigation into the Congo Basin peatlands Holocene history and the relationship between the peatland carbon accumulating function and hydroclimate. This is important because: 1) The Congo Basin represents the largest intact tropical peatland complex 2) There is little data on peatland development and resilience in the tropics 3) The fate of this peatland complex in the future very much matters in terms of carbon-cycle climate feedbacks.

The manuscript provides an original conclusion that merits publication. The manuscript and abstract are elegantly written, the introduction provides good context to the work.

The methodological approach is thorough and the manuscript presents an impressive amount of multi-proxy paleodata to pin down how climate is likely to have changed over time to explain variations in the peatland's carbon exchange with the atmosphere. I appreciate how the authors have moved away from attempting to quantify the change in the peatland's carbon sink - something that would need a modelling approach, meriting a separate publication - but have instead focussed the efforts in understanding through a multi-proxy approach how a dry period lead to contemporaneous and secondary decomposition of the peat carbon some 2000 years ago, leading to the eventual record of what they have called the Ghost Period. I particularly appreciated the description of the modern day climate space that shows how these peatlands are on the edge of the precipitation amount and seasonality distribution.

All of this impressive work is definitely appropriate for the aims of the investigation. The reporting of the data is also sufficiently detailed. Uncertainty in the data has been shown visually in figures or reported in the text.

In my view, these results will be of interest, not just to scientists in the discipline, but more widely to tropical ecologists (including those studying terra firme forests), climate scientists and soil scientists. The changes in climate would affect not just peatlands, but all other ecosystems in the Congo Basin.

The conclusions and data interpretation are robust, valid and reliable. The manuscript reference previous literature appropriately.

The quality of presentation of the data is very good."

I made some comments on quality of some of the graphics (which have been fixed). I am also satisfied that the

data will be made publicly available.

The authors have now addressed most of the reviewers comments. Where they do not fully address these, they have explained why not and provided acceptable reasons. They have importantly added additional data (data from two new cores from the area) that support the Ghost Interval highlighted by the work that was one of the main problems with the previous version of the manuscript. They have additionally shown that using a different calibration curve does not change the conclusions of the study. The issues with the complications of reconstruction of paleo-precipitation have been addressed. As I already wrote in my first review. This works merits publication in *Nature*, the work is novel, it is the first of its kind for the region - the main tropical peatland in Africa and importantly, a vast intact tropical peatland region that has never been studied before. The edits and additions to the manuscript have improved its quality. I agree with the authors on the conclusions (that the Ghost Interval likely represents peat loss) - and I am satisfied that the wording represents well the uncertainty in the results and does not over-claim conclusions.

Many thanks for the kind words on our study.

Referee #4 (Remarks to the Author):

A major concern was the reliance of a single peat core. The authors have now included two additional peat cores from the wetland complex, both of which exhibit similar (although not identical) bulk geochemical features during the middle Holocene. This is a welcome addition. However, visual inspection of the two new cores suggest that the Ghost Interval terminates at different times in the peat complex (e.g. ~ 4 ka in LOK5-5). In addition, the new data does not help to refine the authors data interpretation and only one of the four potential hypothesis is excluded (i.e., reduced litter input; L283-286). As such, the papers remains speculative.

We answer the question about visual inspection of different times under point 4 below. On the speculative point: as Referee #4 said in the initial review: “I really enjoyed reading this manuscript – it is well written and flowed logically. The figures are clear and concise. The findings have the potential to be genuinely important. The topic is also timely and has direct relevance to the future.” We also think the manuscript is genuinely important, but it is also far from the last word on this system. We can only go where the data takes us, and can only exclude one possibility at present. We set the hypotheses out to aid clear thinking rather than in the expectation that only one process would occur across the central Congo peatlands.

I also have various questions about the data interpretation (see below):

1) Leaf wax hydrogen isotope - vegetation effects

The authors use leaf wax $\delta^{2}H$ values to reconstruct precipitation $\delta^{2}H$ during the "ghost interval". However, the study shows a change in vegetation type within the Ghost Interval (e.g., decline in swamp forest taxa accompanied by an increase in light demanding and pioneer taxa; L184-189). As the apparent fractionation between $\delta^{2}H_{wax}$ and $\delta^{2}H_{precip}$ varies widely (up to ~70 ‰) between plant life forms (i.e., trees, shrubs, forbs, graminoids) and physiological groups (i.e., C3, C4, CAM), such changes could bias your results.

The authors acknowledge that “...different vegetation types (e.g., C3 vs. C4) can cause offsets in δD values of plant wax due to different hydrogen isotope fractionation factors”. However, they argue that changes in C3/C4 plants are not important at this site. I agree - there is no obvious change in C3/C4 plants (this would be reflected in the leaf wax $\delta^{13}C$ values...). However, you do find large changes in the composition of C3 plants. Changes in the plant community can modify leaf wax hydrogen isotope values significantly, especially in peats. For example, Balascio et al. (2018) analysed leaf waxes in a Norwegian bog and found that that $\delta^{2}H$ values varied from -120 to -220‰ across a range of plant species.

Following the last review we are happy to agree with Referee #4 that, given the continued dominance of C3 trees in our system, we do not expect any important changes in the apparent fractionation between $\delta^{2}H_{wax}$ and $\delta^{2}H_{precip}$ at our CEN-17.4 site due to changes in plant life forms. Now the question is whether a change in a forest composed of C3 tree vegetation could change fractionation between $\delta^{2}H_{wax}$ and $\delta^{2}H_{precip}$. We make three points.

First, the Norwegian peat bog study actually found variation due to different plant life forms, and not C3 trees. Their abstract states: “We identified 14 different plant types growing on the bog surface, including mosses, graminoids and other herbs, sub-shrubs and a tree.” Thus, this study does not provide assistance with the question about variation within C3 trees.

Second, we know of no study that describes the apparent fractionation between $\delta^2\text{H}_{\text{wax}}$ and $\delta^2\text{H}_{\text{precip}}$ in modern plants of any type from the Central Congo peatlands. The system was only recently identified and mapped so no study on this matter has been conducted yet.

Third, and more positively, we have independent evidence that our assumption that changes in C3 tree species composition is not a major driver of changes in hydrogen isotope fractionation, which we highlight in this new section in the Methods, writing (lines 750 to 758): “Plant wax δD is a robust tracer for $\delta\text{D}_{\text{precip}}$ and is extensively used in tropical Africa to reconstruct palaeohydrology^{22,30,31,73,74}. Since there are no significant changes in both plant life forms and photosynthetic pathways in the CEN-17.4 record during the Holocene (see above) we assume an invariant hydrogen isotope fractionation between $\delta\text{D}_{n\text{-C}29}$ and $\delta\text{D}_{\text{precip}}$. Furthermore, a marine record off the Congo River (core GeoB 6518-1) going back to 20,000 cal. yr BP shows a striking correlation between $\delta\text{D}_{n\text{-C}29}$ and the $\delta^{18}\text{O}$ of planktonic foraminifera²². As the latter is determined by discharge amount of the Congo River, this suggests that both, notably independent, climatic proxies record large-scale central African precipitation changes²².”

If a strong correlation between $\delta^2\text{H}_{\text{wax}}$ and rainfall amounts, and thus rainfall isotopes, exists on the catchment scale, $\delta^2\text{H}_{\text{wax}}$ on a local scale must also be controlled by rainfall amounts and not differential fractionation, in the majority of the catchment. It is thus highly unlikely that it would be different in the Congo peatlands, centrally located in the basin.

Overall, our assumption that changes in the composition of C3 trees that form a tropical forest is not affecting fractionation between $\delta^2\text{H}_{\text{wax}}$ and $\delta^2\text{H}_{\text{precip}}$ is the most parsimonious based on the available evidence.

2) Leaf wax hydrogen isotopes - inferring $\delta^2\text{H}_{\text{precip}}$

The authors used "a global surface soil calibration relationship (Ladd et al. 2021), which include peat samples" to infer $\delta^2\text{H}_{\text{precip}}$ values. However, I am unclear how many peat samples are actually included in this calibration. I had a quick look into two major compilations included in Ladd et al., 2011 (McFarlin et al., 2019 and Liu and Ann, 2019; these should be cited). The McFarlin et al., 2019 paper appears to be lakes only, and Liu and Ann 2019 doesn't appear to include any peat samples (e.g., Huang 2018, 2016, Balascio 2018 ad Zaho et al., 2018, Seki 2009). Can the authors please clarify how many peat samples are used in this compilation?

This is crucial because the source of the pore water used for leaf wax biosynthesis in peatlands is not well constrained and is likely variable across the peatland. For example, , Huang et al. (2016) demonstrated that $\delta^2\text{H}$ values of long-chain n-alkanes within a single peatland is highly variable. This uncertainty needs to be considered.

Eleven peat samples were included. We have added this to the Methods.

3) C/N ratios

The authors argue that higher C/N ratios are an indicator for enhanced decomposition. However, evidence suggests we would expect the opposite - microbial consumption of C- and H-rich organic substances is known to result in a decreased abundance of carbon relative to nitrogen (i.e., lower C/N ratios; e.g. Malmer and Holm, 1984). In the response, the authors disagree and suggest this argument is not valid in peats. However, the evidence I cited was based on peats. In fact, the C/N ratio has been commonly used as indicator for the degree of peat decomposition, based on the relatively higher loss of C compared to N during decomposition and thus indicating peat mass loss (e.g. Hornibrook et al., 2000; Kuhry and Vitt, 1996; Broder et al., 2012). Thus I struggle to understand how higher C/N ratios = enhanced decomposition.

Carbon-to-nitrogen ratios behave differently according to specific peat type and setting. We explained this in the Method section, indicating that the C/N changes with depth are different in tropical peat swamp forests from boreal *Sphagnum* peat (lines 593 to 595). We have now also added a recent reference to the Methods section, which shows that the response of C/N ratios to degradation depends on peat nutrient status (Reuter et al., 2020) with low nutrient peats being characterized by preferential protein loss. Furthermore, the increase of the C/N ratios that we relate to decomposition is supported by our Rock-Eval data on the same samples (Fig. 2 and Extended Data Figs 1 & 3).

To avoid generalization about C/N in peat we have amended the sentence in the main text to be specific to our study site peatland (lines 148 to 152): “Third, the carbon-to-nitrogen ratio (C/N) indicates increased

decomposition (Fig. 2d, Methods and Extended Data Fig. 2), as we observe a prominent increase in C/N to ~80 in this part of the record. At our low-nutrient site¹³ this may imply a preferential loss of nitrogen through aerobic decomposition, triggered by a lowering of the water table, although other processes including vegetation changes¹⁴ might also influence C/N.”

Furthermore, we cited Hodgkins et al. (Nat. Commun., 2018) in the Methods section (lines 600 to 605), who demonstrated that boreal and tropical peats have a different chemistry, with the tropical ones being more recalcitrant. “Peat and plant chemistry are different in boreal and tropical regions, resulting in higher recalcitrance for tropical peat⁵⁷. Roots and woody material from swamp forest tree species dominate the organic matter composition of the Ekolongouma peat. Anaerobic decomposition of this type of peat is typically very slow under constant high water tables, which is attributed to the recalcitrance of the organic matter⁵⁸”.

4) The ghost interval ends at different times...

Leaf wax $\delta^{13}C$ values suggest drying (5.0-2.0 kyr) occurs 2,000 years after the Ghost Interval begins. The authors argue that this pattern is consistent with secondary decomposition (scenario 3) around 2,000 years ago (L133-135 in revised MS). However, visual inspection of the two new cores suggest that the Ghost Interval terminates closer to 4 ka (LOK5-5) and 2.5 ka (BDMI-7). This suggests that secondary decomposition scenario is occurring at different times in the peat complex - it may also be inconsistent with the secondary decomposition scenario argued above.

We did not mean to inadvertently imply that every location in the peatlands would terminate the Ghost Interval at exactly 2,000 cal yr BP. We thank the reviewer for pointing this out. Our dating resolution at the additional locations is poorer (only one-third of the sampling intensity), and site specific differences may also influence the timing of the ending of the Ghost Interval. We clarify this with an additional sentence in the main text (lines 112 to 115): “The three cores have a comparable age-depth profile pattern, indicating a common large-scale driver of the Ghost Interval. However, the precise timing of the onset and termination of the Ghost Interval at each location will be impacted by differing AMS dating resolution and site-specific differences.”

Of course, any dating or site-specific differences do not undermine the large-scale occurrence of a peat decomposition period shown in our study.

5) Leaf wax hydrogen isotopes and carbon cycling decoupling

The authors argue that drier conditions during the Ghost interval help to promote OC decomposition. This argument is based upon a shift towards high $\delta^{13}C$ values (-140 permil). However, there are intervals with similarly high values (e.g., 10-12 thousand years ago), but with NO evidence for enhanced OC decomposition. The authors have attempted to address this in the rebuttal, but I don't quite understand their explanation. Can the authors please expand and include some relevant citations to support their argument?

We clarify the explanation of limited OC decomposition: from 12 to 9 ka BP, the $\delta^{13}C_{org}$ data of core CEN-17.4 indicate a wetting trend, which corresponds to the development of a rainfed peat swamp forest with C3 vegetation, hence the lack of OC decomposition. We clarify this with better labeling of Fig. 3 – please see our response to Referee#2.

Minor comments:

The paper states that "Our results indicate a newly identified positive feedback in the global carbon cycle, whereby climate-induced drought leads to the release of further carbon from peat to the atmosphere". However, I don't think this is a new idea (e.g., Limpens, J. et al. Biogeosciences (2008).

We have modified this sentence to be clear that by ‘newly identified’ we are referring to the central Congo peatlands, and not a peat carbon feedback generally: “Our results indicate a newly identified positive feedback in the global carbon cycle, whereby climate-induced drying in the central Congo Basin leads to the release of further carbon from peat to the atmosphere”.

Reviewer Reports on the Second Revision:

Referees' comments:

Referee #1 (Remarks to the Author):

This is the second time I am reviewing this manuscript. I very much appreciate the new analyses the authors have added since the initial submission. The new data strengthens the manuscript, in particular the better sampling resolution and additional decomposition proxies for the Ekolongouma Basin secondary cores (EKGKM7 and EKG03); as well as the addition of two new cores in Lokolama (LOK5-5) and Bondamba (BDM1-7). The figures and text are also much clearer.

I have the following additional comments to clarify the presentation of the results:

-----The abstract conveys much more certainty than is expressed by the figures and text. The text already does an excellent job of considering possible scenarios that could explain the data and weighing their relative likelihood. This nuance needs to be expressed in the abstract as well, to avoid confusion for casual readers. For example, please update the phrasing of the abstract Ln 63-66 to indicate some uncertainty (e.g. "likely explanation", "may have triggered") and mention both the contemporaneous and secondary decomposition that are frequently referenced throughout the text. The contemporaneous decomposition seems to be missing from the abstract.

For example, I really appreciated the clear summary (Ln 270-273): "Collectively, the evidence suggest that the hydroclimate change from the Mid- to Late Holocene resulted in a regional drop in the water table and led to both contemporaneous (scenario 2) and secondary (scenario 3) decomposition of peat, coupled with secondary removal via peat erosion at sites close to river courses (scenario 4)."

Please make sure that this summary is reflected to a greater degree in the abstract. This sentence in particular would benefit from clarification to avoid oversimplification of the manuscript's interesting results -- Ln 63-66 "The peat appears to have lost a substantial amount of its original carbon mass during a phase of severe decomposition, including the loss of carbon accumulated during prior wetter conditions, as drying conditions triggered a switch from net peat accumulation to net peat loss."

-----It is very interesting that the peatland recovered even as conditions stayed dry from 2000 to 900 cal yr BP. Making sure this is clear would motivate future research.

---Please update the phrasing throughout the manuscript to clarify that the driest period lasted from approximately ~2000 to 900 cal yr BP (or whatever the specific date range is). Although it is not the intent, the current phrasing often leads the reader to think that drying peaked in 2000 cal yr BP and immediately started getting wetter afterwards, and requires frequent cross-referencing with the figures to avoid confusion. Some places with opportunity to clarify, and possible suggestions, include:

-Ln 61-62: "indicate a shift to drier conditions, culminating from 2000 to 1000 cal yr BP" (rather than "culminating at")

- Ln 186: "Immediately following the Ghost Interval, swamp forest taxa gradually increased although the climate remained dry, reaching their maximum abundance..."

-Ln 238-246- Excellent paragraph, but please explicitly state that it stayed dry until 900 cal yr BP at the beginning of the paragraph to clarify for readers. Possible phrasing: "renewed high rates of new peat addition following the Ghost Interval contrasts with the high and stable dD values, indicating dry conditions lasted from ~2000 cal yr BP until 900 cal yr BP."

-Ln 258: "culminated at 2000 to 900 cal yr BP."

---Could you add a supplemental figure with age-depth curves for all the peat cores together? (Similar to the earlier fig 1 & current fig 1 combined) Or at least all the EKG age-depth curves together? It looks like they were removed during the revisions as the data is now separately presented in other places. However, it would still be helpful to show them together since the central message of the paper relates to peat accumulation.

--- Curiosity - Does the peat of the Ghost Interval look different in color or texture? Would it be visible in a core in the field? Fig2a cartoon suggests not, but if the peat characteristics are so different, perhaps so? You could mention somewhere if not already included (e.g. SI or related figure caption), as it could inform other researchers in the future, especially for fieldwork if any visible differences.

Referee #4 (Remarks to the Author):

This is the third time I have reviewed this paper and I appreciate the authors response to my early reviews. However, I still have a few comments:

1) Leaf wax $\delta^{2}H$ record - the authors argue that drier conditions during the Ghost interval help to promote OC decomposition. This argument is based upon a shift towards ^{2}H -enriched leaf wax values (-140 permil). However, there are intervals with similarly high values (e.g., 10-12 thousand years ago), but with NO evidence for enhanced OC decomposition.

I appreciate there is a 'wetting' trend between 12 to 9 kyr. However, its hard to ignore that the leaf wax values in the early Holocene and Ghost Interval are the same (-140 per mil). Yet only the latter is associated with enhanced decomposition. Why is this?

Related to this, if you apply the same workflow to compute estimates of past precipitation at CEN-17.4 using the relationship between $\delta^{2}H$ precip and precipitation amount (i.e., Extended Figure 6), does the early Holocene exhibit similar MAP values to the Ghost Interval. Are the authors able to carry out this additional analysis?

If the early Holocene was dry, it implies the Congo peatland is only vulnerable to drying when it happens quickly (i.e. late Holocene). This is still a nice finding. And perhaps makes the findings even more important in the face of (rapid) anthropogenic warming.

2) this paper attributed an increase in C/N ratios to reflect enhanced decomposition, which is an uncommon finding. Existing literature from peat (see my previous review) suggest that C/N ratios typically decrease in response to decomposition. The authors add one reference (Reuter et al, 2020) to support their claim, but that study focuses on high-latitude Sphagnum dominated peatlands (which is not analogous to this site). Perhaps looking at other decomposition metrics (e.g. carbon preference index) may be more convincing and avoid an overreliance on a relatively simple metric.

Minor comments:

L198: one way to test this would be to look for the presence of absence of coprostanol and other faecal biomarkers in your samples (see Sear et al., 2020 in PNAS)

Fig 3. leaf wax $\delta^{13}C$ data – axis should be flipped. More enriched values towards top, more depleted values towards the bottom

L47-249: would be useful to actually see these records in SI – otherwise its hard to assess this claim

Fig. 3 I am not sure if your assignment of the drying and recovery in figure 3 matches the data – pollen and leaf waxes suggest recovery occurs after 1kyr. Whereas its currently labelled at 2kyr...but that looks like the peak dry interval

Author Rebuttals to Second Revision:

NB: *The Reviewer's comments are printed in italic font (black). For reading ease our responses to the comments are printed in bold font (black). Passages in the manuscript (corrected or original) are printed in normal font (blue).*

We thank Referees #1 and #4 for their insightful and constructive comments on the third version of our manuscript.

Referee #1 (Remarks to the Author):

This is the second time I am reviewing this manuscript. I very much appreciate the new analyses the authors have added since the initial submission. The new data strengthens the manuscript, in particular the better sampling resolution and additional decomposition proxies for the Ekolongouma Basin secondary cores (EKGKM7 and EKG03); as well as the addition of two new cores in Lokolama (LOK5-5) and Bondamba (BDM1-7). The figures and text are also much clearer.

We appreciate the positive evaluation of Referee #1.

I have the following additional comments to clarify the presentation of the results:

-----The abstract conveys much more certainty than is expressed by the figures and text. The text already does an excellent job of considering possible scenarios that could explain the data and weighing their relative likelihood. This nuance needs to be expressed in the abstract as well, to avoid confusion for casual readers. For example, please update the phrasing of the abstract Ln 63-66 to indicate some uncertainty (e.g. "likely explanation", "may have triggered") and mention both the contemporaneous and secondary decomposition that are frequently referenced throughout the text. The contemporaneous decomposition seems to be missing from the abstract.

For example, I really appreciated the clear summary (Ln 270-273): "Collectively, the evidence suggest that the hydroclimate change from the Mid- to Late Holocene resulted in a regional drop in the water table and led to both contemporaneous (scenario 2) and secondary (scenario 3) decomposition of peat, coupled with secondary removal via peat erosion at sites close to river courses (scenario 4)."

Please make sure that this summary is reflected to a greater degree in the abstract. This sentence in particular would benefit from clarification to avoid oversimplification of the manuscript's interesting results -- Ln 63-66 "The peat appears to have lost a substantial amount of its original carbon mass during a phase of severe decomposition, including the loss of carbon accumulated during prior wetter conditions, as drying conditions triggered a switch from net peat accumulation to net peat loss."

We agree with your suggestion and within the word limits, and avoiding the terms contemporaneous and secondary decomposition, as we do not have space to explain, we have updated lines 68-70 in the abstract accordingly: "This drying probably resulted in a regional drop in the water table, which triggered peat decomposition including the loss of peat carbon accumulated prior to the onset of the drier conditions."

-----It is very interesting that the peatland recovered even as conditions stayed dry from 2000 to 900 cal yr BP. Making sure this is clear would motivate future research.

Thank you, we agree.

---Please update the phrasing throughout the manuscript to clarify that the driest period lasted from approximately ~2000 to 900 cal yr BP (or whatever the specific date range is). Although it is not the intent, the current phrasing often leads the reader to think that drying peaked in 2000 cal yr BP and immediately started getting wetter afterwards, and requires frequent cross-referencing with the figures to avoid confusion. Some places with opportunity to clarify, and possible suggestions, include:

-Ln 61-62: "indicate a shift to drier conditions, culminating from 2000 to 1000 cal yr BP" (rather than "culminating at")

We have re-phrased this (lines 70-71) to read: "After ~2,000 cal. yr BP, the drying trend ceased, hydrologic conditions stabilized, and peat accumulation resumed."

- Ln 186: “Immediately following the Ghost Interval, swamp forest taxa gradually increased although the climate remained dry, reaching their maximum abundance...”

Line 186 belongs to the “Vegetation changes” section where the climate reconstruction is not yet introduced. We refrained from discussing the climate status here as this is done in the subsequent section of the main text.

-Ln 238-246- Excellent paragraph, but please explicitly state that it stayed dry until 900 cal yr BP at the beginning of the paragraph to clarify for readers. Possible phrasing: “renewed high rates of new peat addition following the Ghost Interval contrasts with the high and stable dD values, indicating dry conditions lasted from ~2000 cal yr BP until 900 cal yr BP.”

Thank you, we have updated the text accordingly (lines 253 to 256): “The recovery of hydrophilic swamp forest taxa and renewed high rates of new peat addition following the Ghost Interval (‘Recovery’ in Fig. 3) contrasts with the high and stable δD_{precip} , suggesting that the drier conditions were relatively stable and lasted from ~2,000 until ~900 cal. yr BP.”

-Ln 258: “culminated at 2000 to 900 cal yr BP.”

Here, we keep “culminated at ~2,000 cal. yr BP” since this is the drying trend that is discussed, not the dry conditions *sensu stricto*.

---Could you add a supplemental figure with age-depth curves for all the peat cores together? (Similar to the earlier fig 1 & current fig 1 combined) Or at least all the EKG age-depth curves together? It looks like they were removed during the revisions as the data is now separately presented in other places. However, it would still be helpful to show them together since the central message of the paper relates to peat accumulation.

The reviewer is correct, that in the re-submitted version we removed the previously published (Dargie et al. 2017 Nature) EKG age-depth curves, and replaced them with age-depth cores from far away (BDM, LOK, in this version Fig. 1) as we were asked by the reviewers to show that the Ghost Interval was not occurring at only one site. These age-depth curves are in Fig. 1. Then, again in response to reviewer concerns, we focused on obtaining identical geochemical and radiocarbon data for each of the EKG cores, to ensure transparent comparability across the EKG and main CEN cores. However, some of these EKG cores did not have down-core dates (as they were from a different PhD project), thus for this manuscript we additionally only dated the top and bottom boundaries of the Ghost Interval (based on geochemical evidence) for the EKG cores in Extended Data Fig. 3. Therefore, we cannot provide age-depth models for these EKG cores as we do not have them, and the ones we do have are already published.

--- Curiosity - Does the peat of the Ghost Interval look different in color or texture? Would it be visible in a core in the field? Fig2a cartoon suggests not, but if the peat characteristics are so different, perhaps so? You could mention somewhere if not already included (e.g. SI or related figure caption), as it could inform other researchers in the future, especially for fieldwork if any visible differences.

Thank you, this is a very interesting point. We now added the following sentence in the caption of Fig. 2: “Note that peat colour and texture in the Ghost Interval is similar to that of the peat immediately below and above.”

Referee #4 (Remarks to the Author):

This is the third time I have reviewed this paper and I appreciate the authors response to my early reviews.

Thank you.

However, I still have a few comments:

1) Leaf wax $\delta^{2}H$ record - the authors argue that drier conditions during the Ghost interval help to promote OC decomposition. This argument is based upon a shift towards ^{2}H -enriched leaf wax values (-140 permil). However, there are intervals with similarly high values (e.g., 10-12 thousand years ago), but with NO evidence for enhanced OC decomposition. I appreciate there is a ‘wetting’ trend between 12 to 9 kyr. However, its hard to ignore that the leaf wax values in the early Holocene and Ghost Interval are the same (-140 per mil). Yet only the latter is associated with enhanced decomposition. Why is this? Related to this, if you apply the same workflow to compute estimates of past precipitation at CEN-17.4 using the relationship between $\delta^{2}H$ precip

and precipitation amount (i.e., Extended Figure 6), does the early Holocene exhibit similar MAP values to the Ghost Interval. Are the authors able to carry out this additional analysis? If the early Holocene was dry, it implies the Congo peatland is only vulnerable to drying when it happens quickly (i.e. late Holocene). This is still a nice finding. And perhaps makes the findings even more important in the face of (rapid) anthropogenic warming.

This point was previously risen by Referee #4. We tried to clarify even more. We have amended the text, lines 245-252: “Our hydroclimate record also reveals drier conditions at ~12,000 cal. yr BP (Fig. 3b), but this is not associated with as shallow an age-depth profile as we find in the Ghost Interval (Fig. 1b). This likely relates to the differing environment at the time, being dominated by C4 grasses which are more typical of wetter floodplain and marshy habitats²⁸, rather than the forest we find in the Ghost Interval (inferred from the shift from higher to lower $\delta^{13}\text{C}_{\text{n-C}_{29}}$ values at ~12,000 cal. yr BP; Fig. 2f). Furthermore, the peat was thinner at ~12,000 cal. yr BP, and so likely had a lower elevational gradient between its centre and margins (slower surface-water flow), potentially lessening the impact of the drying²⁹.”

²⁸Moutsamboté, J. M. Ecological, Phytogeographic and Phytosociological Study of Northern Congo (Plateaus, Bowls, Likouala and Sangha) (University of Marien Ngouabi, Brazzaville, Republic of the Congo, 2012).

²⁹Dingman, S. L. Fluvial Hydrology (ed. Freeman, W. H.) (New York, 1984).

2) this paper attributed an increase in C/N ratios to reflect enhanced decomposition, which is an uncommon finding. Existing literature from peat (see my previous review) suggest that C/N ratios typically decrease in response to decomposition. The authors add one reference (Reuter et al, 2020) to support their claim, but that study focuses on high-latitude Sphagnum dominated peatlands (which is not analogous to this site). Perhaps looking at other decomposition metrics (e.g. carbon preference index) may be more convincing and avoid an overreliance on a relatively simple metric.

We have added the Carbon Preference Index in Extended Data Fig. 2 (see below). The marked drop in CPI values during the Ghost Interval also supports enhanced decomposition (as do the Rock-Eval, $\delta^{13}\text{C}_{\text{bulk}}$ and pollen data). We have also added, the following on the C/N ratio, with more references, at lines 620-621: “Fluctuating wet and dry conditions, varying N deposition and changing vegetation during peat formation can overprint the expected trends of C/N with depth⁵⁹⁻⁶².”

⁵⁹Broder, T., Blodau, C., Biester, H. & Knorr, K. H. Peat decomposition records in three pristine ombrotrophic bogs in southern Patagonia. *Biogeosciences* 9, 1479-1491 (2012).

⁶⁰Hornibrook, E. R. C., Longstaffe, F. J. & Fyfe, W. S. Evolution of stable carbon isotope compositions for methane and carbon dioxide in freshwater wetlands and other anaerobic environments. *Geochim. Cosmochim. Acta* 64, 1013-1027 (2000).

⁶¹Biester, H., Knorr, K. H., Schellekens, J., Basler, A. & Hermanns, Y. M. Comparison of different methods to determine the degree of peat decomposition in peat bogs. *Biogeosciences* 11, 2691-2707 (2014).

⁶²Leifeld, J., Klein, K. & Wüst-Galley, C. Soil organic matter stoichiometry as indicator for peatland degradation. *Sci. Rep.* 10, 7634 (2020).

Minor comments:

L198: one way to test this would be to look for the presence of absence of coprostanol and other faecal biomarkers in your samples (see Sear et al., 2020 in PNAS)

Thank you for your suggestion, but we cannot do this. Non-human primates and other omnivorous mammals are producing faecal biomarkers such as coprostanol, but a database of faecal signatures for Central Africa would be necessary before attributing faecal peat biomarkers to humans, and this does not, to our knowledge, exist.

Fig 3. leaf wax δD_{wax} data – axis should be flipped. More enriched values towards top, more depleted values towards the bottom

We keep Fig. 3 as it is, as the palaeoclimate community often plots the δD_{wax} on a reversed y-axis to better reflect wetter (upwards) and drier (downwards) conditions.

L247-249: would be useful to actually see these records in SI – otherwise its hard to assess this claim

We agree. We now present a compilation of hydroclimate records in Western Central Africa in a new figure: Extended Data Fig. 7 (see below).

Fig. 3 I am not sure if your assignment of the drying and recovery in figure 3 matches the data – pollen and leaf waxes suggest recovery occurs after 1kyr. Whereas its currently labelled at 2kyr...but that looks like the peak dry interval

We agree. Referee #1 had a similar comment and we updated the main text accordingly.